# Scalable Single-Cell Gene Expression Generation with Latent Diffusion Models

Giovanni Palla [1 2]   Sudarshan Babu [3]   Payam Dibaeinia [3]   James D. Pearce [1]   Donghui Li [1]   Aly A. Khan [3 4]
Theofanis Karaletsos [5 6 7]   Jakub M. Tomczak [1 7]

## Abstract

Computational modeling of single-cell gene expression is crucial for understanding cellular processes, but generating realistic expression profiles remains a major challenge. This difficulty arises from the count nature of gene expression data and complex latent dependencies among genes. Existing generative models often impose artificial gene orderings or rely on shallow neural network architectures. We introduce a scalable latent diffusion model for single-cell gene expression data, which we refer to as scLDM, that respects the fundamental exchangeability property of the data. Our VAE uses fixed-size latent variables leveraging a unified Multi-head Cross-Attention Block (MCAB) architecture, which serves dual roles: permutation-invariant pooling in the encoder and permutation-equivariant unpooling in the decoder. We enhance this framework by replacing the Gaussian prior with a latent diffusion model using Diffusion Transformers and linear interpolants, enabling high-quality generation with multi-conditional classifier-free guidance. We show its superior performance in a variety of experiments for both observational and perturbational single-cell data, as well as downstream tasks like cell-level classification.

## 1. Introduction

Single-cell transcriptomics has revolutionized our understanding of cellular heterogeneity and biological processes at unprecedented resolution (Rozenblatt-Rosen et al., 2017), enabling high-throughput gene expression profiling across millions of cells (Virshup et al., 2023), and providing insights into cellular differentiation (Gulati et al., 2020), disease progression (Zeng & Dai, 2019), responses to drug perturbations (Adduri et al., 2025; Bereket & Karaletsos, 2023; Zhang et al., 2025). However, modeling the complex, high-dimensional gene expression data from single cells presents significant computational and methodological challenges (Lähnemann et al., 2020; Luecken et al., 2022; Neu et al., 2017).

Deep generative modeling (Tomczak, 2024) offers a powerful framework to formulate expressive probability distributions. In the context of single-cell data, multiple methods have been proposed. In particular, Variational Auto-Encoders (VAEs) have been extensively utilized for representation learning (single-cell Variational Inference; scVI) (Lopez et al., 2018), perturbation modeling (Lotfollahi et al., 2023b; Palma et al., 2025b), trajectory inference (Gayoso et al., 2024), among others (Gayoso et al., 2022). Additionally, Generative Adversarial Networks (GANs) have also been proposed, both for generating realistic cell populations (scGAN; (Marouf et al., 2020b)) and for inferring cellular trajectories (Reiman et al., 2021). Recently, diffusion-based models have also been adopted for single-cell gene expression (Luo et al., 2024). An interesting research line was proposed in (Palma et al., 2025a) that combines scVI with a flow matching model in the latent space (CFGen).

However, two key challenges limit existing methods. First, they often require a fixed ordering of genes or operate on a restricted subset of highly variable genes (HGVs). This assumption directly clashes with the biological reality that gene expression profiles are **exchangeable** sets, where the order of genes carries no meaning. Beyond exchangeability itself, fixed-order architectures tie each input dimension to a specific gene, so the gene vocabulary cannot vary across tissues or species without retraining or surgical weight permutation. A permutation-invariant design that identifies genes through embedding indices rather than input positions removes this restriction by construction. Second, approaches based on GANs inherit well-known training instabilities and risks of mode collapse. These limitations make current models inflexible, difficult to scale, and unable to properly handle the unordered nature of single-cell data.

[1]Biohub, Redwood City, CA, USA [2]current affiliation: Lila Sciences [3]Biohub, Chicago, IL, USA [4]University of Chicago, Chicago, IL, USA [5]Achira, Inc, San Francisco, CA, USA [6]work conducted at Chan Zuckerberg Initative, Redwood City, CA, USA [7]co-last authors. Correspondence to: Giovanni Palla <giov.pll@gmail.com>, Jakub M. Tomczak <jmk.tomczak@gmail.com>.

*Proceedings of the $43^{rd}$ International Conference on Machine Learning*, Seoul, South Korea. PMLR 306, 2026. Copyright 2026 by the author(s).

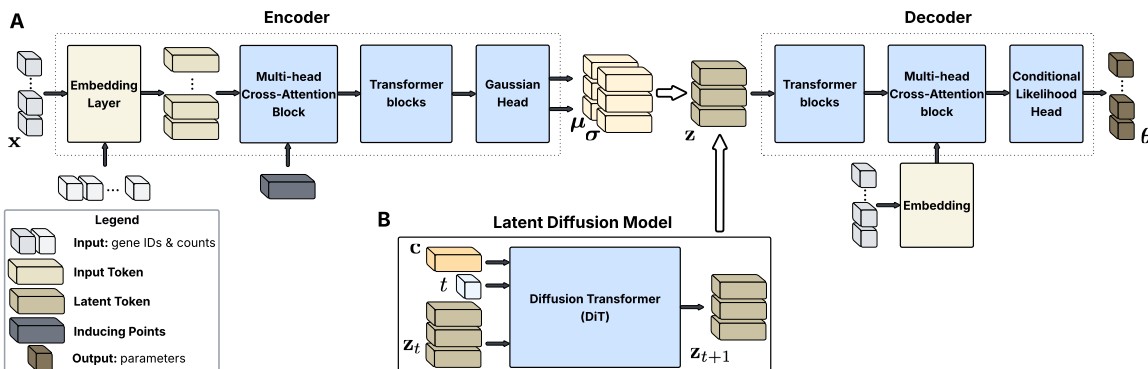

*Figure 1.* Our deep generative model, scLDM, for single-cell gene expression data. **A**: A fully transformer-based architecture for processing gene expressions. The encoder network results in permutation-invariant latent variables represented as tokens. The decoder network returns permutation-equivariant counts for given gene IDs. **B**: At the second stage, a vanilla prior is replaced by a latent diffusion model. We model latent tokens using Diffusion Transformers (DiT), and train the resulting LDM using linear interpolants and the flow matching loss. Sampling is carried out by applying the Scalable Interpolant Transformers (SiT) library (Ma et al., 2024).

This paper introduces a novel approach that combines the flexibility of VAEs with the power of latent diffusion models (see Figure 1), specifically designed to handle the exchangeable nature of gene expression data. The key insight is that careful architectural choices, particularly in the parameterization of permutation-invariant and permutation-equivariant components, result in a scalable, deep, and exchangeable generative model. The paper's contributions are as follows:

- We propose a novel fully transformer-based VAE architecture for exchangeable data that uses a single set of fixed-size, permutation-invariant latent variables. The model utilizes a Multi-head Cross-Attention Block (MCAB) that serves two purposes: It acts as a permutation-invariant pooling operator in the encoder, and functions as a permutation-equivariant unpooling operator in the decoder. This unified approach eliminates the need for separate architectural components for ensuring parameterizing exchangeable probability distributions.

- We replace the standard Gaussian prior with a latent diffusion model trained with the flow matching loss and linear interpolants using the Scalable Interpolant Transformers formulation (SiT) (Ma et al., 2024), and a denoiser parameterized by Diffusion Transformers (DiT) (Peebles & Xie, 2023). This allows for better modeling of the complex distribution of cellular states and enables controlled generation through classifier-free guidance.

- The proposed framework, which we refer to as scLDM, supports generation conditioned on multiple attributes simultaneously through a joint classifier-free guidance formulation over attribute combinations, which we show outperforms additive multi-attribute guidance (as used in CFGen) on perturbational benchmarks. Moreover, we indicate the strengths of our fully transformer-based auto-encoder in terms of reconstruction metrics and on a downstream prediction task.

Code is available at: https://github.com/czi-ai/scldm.

**Conflict of Interest Disclosure** None to disclose.

## 2. Background

### Variational Auto-Encoders

Another approach is Variational Auto-Encoders (Kingma & Welling, 2022; Rezende et al., 2014), which offer flexible modeling capabilities. (Kim et al., 2021) proposed SetVAE with two latent variables for varying set sizes: $\mathbf{z}_\mathcal{I}$ matching $\mathbf{x}_\mathcal{I}$'s dimensionality (where $\mathbf{z}_i$ corresponds to $\mathbf{x}_i$, $i \in \mathcal{I}$) and constant-size $\mathbf{c} \in \mathbb{R}^{d_1}$. They used hierarchical VAE with multiple $\mathbf{z}_\mathcal{I}$ and $\mathbf{c}$ layers and replaced conditional likelihood with Chamfer Distance. While we appreciate VAE's flexibility, we find two distinct latents and hierarchical structure unnecessary, arguing that careful parameterization is *crucial* for high performance.

**Permutation-equivariant/invariant Parameterizations** Geometric deep neural networks typically compose permutation-invariant and/or permutation-equivariant layers with nonlinearity activations (Bronstein et al., 2021). DeepSets (Zaheer et al., 2017) exemplifies this blueprint by processing elements consistently regardless of position, then applying symmetric aggregation (averaging or pooling (Kimura et al., 2024; Ilse et al., 2018; Xie & Tong, 2025)) to ensure permutation invariance. However, processing elements separately before aggregation with non-learnable pooling is limiting. Learned attention mechanisms in trans-

former architectures offer a solution, enabling joint element transformation. SetTransformer (Lee et al., 2019) introduces multi-head attention blocks and Pooling by Multi-head Attention for permutation invariance. We propose an alternative parameterization using a single multi-head attention layer for fixed-size output, followed by transformer blocks. While the encoder side of MCAB is analogous to a Perceiver IO block (Jaegle et al., 2022), our decoder reuses the same module with a key modification: the queries are gene-specific embeddings $E\mathcal{I}$ from the shared embedding matrix, indexed by the requested gene IDs. This makes the decoder permutation-equivariant with respect to gene ordering (Property 4) and removes the need for the separate pooling and unpooling architectures used by SetTransformer (PMA + ISAB) and SetVAE.

**Latent Diffusion Models** Latent Diffusion Models (LDMs) perform diffusion processes in learned latent spaces rather than directly in high-dimensional data spaces. Stable Diffusion (Rombach et al., 2022) pioneered this approach for text-to-image synthesis by training diffusion models in the latent space of a pre-trained VAE, dramatically reducing computational costs while maintaining generation quality. This paradigm has proven effective across diverse scientific domains: all-atom diffusion transformers (Joshi et al., 2025) generate molecules and materials with atomic-level precision, similarly LaM-SLidE (Sestak et al., 2025) utilizes transformer-based LDM for molecular dynamics (among others), while La-proteina (Geffner et al., 2025) employs transformer-based partially latent flow matching for atomistic protein generation. These advances demonstrate the versatility of latent diffusion approaches for complex, high-dimensional scientific data across multiple modalities. Here, we extend this framework by proposing a transformer-based LDM for single-cell transcriptomics.

**Generative Models for scRNA-seq** In the context of single-cell genomics, numerous generative models have been developed for (conditional) sampling of gene expression profiles. scVI (Lopez et al., 2018) represents an early VAE-based generative model, while more recent approaches include transformer-based VAEs (Connell et al., 2022), as well as GAN-based and diffusion-based architectures such as sc-GAN (Marouf et al., 2020a) and scDiffusion (Luo et al., 2024). The latter two classes of models operate in continuous space and therefore transform discrete gene expression data into log-normalized counts. Recently, latent diffusion frameworks have emerged with models like SCLD (Wang et al., 2023), scVAEDer (Sadria & Layton, 2025), and CF-Gen (Palma et al., 2025a), which leverage latent diffusion frameworks on top of an MLP-based autoencoders. Additionally, application-specific generative models have been developed for perturbational single-cell genomics, including CPA (Lotfollahi et al., 2023a), SquiDiff (He et al., 2024), CellFlow (Klein et al., 2025), and CellOT (Bunne

et al., 2023), which are tailored to capture the effects of genetic and chemical perturbations on cellular states. Our approach is similar in vein to CFGen and SCLD, but leverages transformer-based architectures for both our newly proposed VAE as well as the latent diffusion model.

## 3. Methodology

### 3.1. Problem formulation

Let us consider $M$ random variables, $\mathbf{x}$, where each $\mathbf{x}_i \in \mathbb{X}^D$, e.g., $\mathbb{X} = \mathbb{N}$. A set of indices of $M$ random variables is denoted as $\mathcal{I}$, namely, $\mathcal{I} = \pi(\{1, 2, \ldots, M\})$, where $\pi(\cdot)$ is a permutation[1]. Further, we denote a specific order of variables in $\mathbf{x}$ determined by $\mathcal{I}$ as $\mathbf{x}_{\mathcal{I}}$. We assume that for a given $\mathcal{I}$, an object $\mathbf{x}_{\mathcal{I}}$ is equivalent to an object defined by $\pi(\mathcal{I})$, namely, $\mathbf{x}_{\mathcal{I}} = \mathbf{x}_{\pi(\mathcal{I})}$. An example of such a setting is gene expression data where $\{1, 2, \ldots, M\}$ corresponds to gene IDs and the order of gene IDs does not change the state of a cell. Further, we assume a *true* conditional distribution model $p(\mathbf{x}_{\mathcal{I}}|\mathcal{I})$ that for a given order of indices $\mathcal{I}$ allows sampling $\mathbf{x}_{\mathcal{I}}$. We access this *true* distribution through observed *iid* data $\mathcal{D} = \{(\mathbf{x}_{\mathcal{I}_n}, \mathcal{I}_n)\}_{n=1}^{N}$. We look for a model $p(\mathbf{x}_{\mathcal{I}}|\theta, \mathcal{I})$ with parameters $\theta$ that optimizes the log-likelihood function for the empirical distribution with data $\mathcal{D}$, $\ell(\theta; \mathcal{D}) = \sum_{n=1}^{N} \ln p(\mathbf{x}_{\mathcal{I}_n}|\theta, \mathcal{I}_n)$. Moreover, we are interested in finding a single model that for given indices $\mathcal{I}$ generates corresponding $\mathbf{x}_{\mathcal{I}}$. Formally, we require the model to be *exchangeable*, namely, $p(\mathbf{x}_{\mathcal{I}}|\mathcal{I}) = p(\mathbf{x}_{\pi(\mathcal{I})}|\pi(\mathcal{I}))$. For instance, a model generates the same gene expression profile for given different orders of gene IDs.

To model an exchangeable probabilistic model $p(\mathbf{x}_{\mathcal{I}}|\theta, \mathcal{I})$, we introduce $m$ latent variables (i.e., the number of latents is fixed for all subsets $\mathcal{I}$), $\mathbf{Z} \in \mathbb{R}^{m \times D}$. By using the family of variational posteriors of the form $q(\mathbf{Z}|\phi, \mathbf{x}_{\mathcal{I}})$, the Evidence Lower BOund (ELBO) is the following:

$$\ln p(\mathbf{x}_{\mathcal{I}}|\theta, \mathcal{I}) \geq \mathbb{E}_{\mathbf{Z} \sim q(\mathbf{Z}|\phi, \mathbf{x}_{\mathcal{I}})} [\ln p(\mathbf{x}_{\mathcal{I}}|\eta, \mathbf{Z}, \mathcal{I}) + \ln p(\mathbf{Z}|\psi) - \ln q(\mathbf{Z}|\phi, \mathbf{x}_{\mathcal{I}})], \quad (1)$$

where $\theta = \{\eta, \psi, \phi\}$ are the parameters of the model. We propose to model these parameters using neural networks, namely: $\phi(\mathbf{x}_{\mathcal{I}}) = \text{NN}_{enc}(\mathbf{x}_{\mathcal{I}})$, $\eta(\mathbf{Z}, \mathcal{I}) = \text{NN}_{dec}(\mathbf{Z}, \mathcal{I})$, and $\psi$ are weights of a parameterization of the prior. Since our assumption is that the model must be exchangable, we propose to parameterize the distributions in a way that: (i) $\mathbf{Z}$ is permutation-invariant, namely, we aim for defining variational posteriors as Gaussian distributions with permutation-invariant neural networks $\text{NN}_{enc}$, (ii) the conditional likelihood is defined as $p(\mathbf{x}_{\mathcal{I}}|\eta(\mathbf{Z}, \mathcal{I})) = \prod_{i \in I} p(\mathbf{x}_i|\eta_i(\mathbf{Z}, \mathcal{I}))$, hence, we must ensure that: $\mathbf{P}\eta(\mathbf{Z}, \pi(\mathcal{I})) = \text{NN}(\mathbf{Z}, \pi(\mathcal{I}))$.

---

[1]We denote a permutation either as a function $\pi(\cdot)$ or, equivalently, as a matrix $\mathbf{P}$.

## 3.2. scLDM: A Transformer-based VAE with Latent Diffusion

**Permutation-invariant/equivariant Cross-Attention** Our VAE is parameterized by a novel transformer-based architecture that leverages multi-head cross-attention block (MCAB), enabling pooling/unpooling operations to avoid processing tens of thousands of tokens at the same time:

$$\text{MCAB}_{\mathbf{S}}(\mathbf{X}) = F(\mathbf{X}, \mathbf{S}) + \text{MLP}(\text{LN}_F(F(\mathbf{X}, \mathbf{S})) \quad (2)$$
$$F(\mathbf{X}, \mathbf{S}) = \mathbf{Q} + \text{Att}_K(\text{LN}_Q(\mathbf{Q}), \mathbf{K}, \mathbf{V})) \quad (3)$$

where $\mathbf{Q} = \text{Linear}_Q(\mathbf{S})$, $\mathbf{K} = \text{Linear}_K(\text{LN}_K(\mathbf{X}))$, $\mathbf{V} = \text{Linear}_V(\text{LN}_V(\mathbf{X}))$, Linear is a linear layer, $\text{LN}(\cdot)$ denotes a layer norm, and $\text{MLP}(\cdot)$ is a fully-connected neural network $\text{MLP}(\mathbf{X}) = (\text{Linear} \circ (\text{Linear} \odot (\text{silu} \circ \text{Linear})))(\mathbf{X})$.[2] $\mathbf{S}$ are learnable pseudoinputs. $\text{MCAB}_{\mathbf{S}}$ is defined similarly to a block in Perceiver (Jaegle et al., 2022; 2021).

MCAB is either permutation-invariant or permutation-equivariant. Since it relies on the attention mechanism, if we permute $\mathbf{X}$ but do not permute $\mathbf{S}$, then MCAB is permutation-invariant (see Property 3). However, if we process $\mathbf{Z}$ by a permutation-invariant function and we permute $\mathbf{S}$ accordingly to the permuted indices, then MCAB becomes permutation-equivariant (see Property 4). Resultantly, we use MCAB as a permutation-invariant pooling operator in the encoder, and as a permutation-equivariant unpooling operator in the decoder.

**Input processing** One of the main challenges for transformer-based architectures is to deal with long context windows. To circumvent the computational complexity following from the potentially very large number of tokenes (e.g., genes), we propose a method for processing sparse data that focuses computational resources on relevant signals. Since we focus on gene expression data, in the following we reger to biological terms instead of generic tokens. Given a set of $D$ genes with their corresponding expression counts, our approach addresses the inherent sparsity in single-cell RNA sequencing data, where typically 70% or more of gene-cell entries are zero.

Let $\mathcal{I} = \{1, 2, \ldots, D\}$ denote the complete set of gene IDs represented as integers, and let $\mathbf{x} = (x_1, x_2, \ldots, x_D)$ represent the corresponding gene expression counts for a given cell, where $x_i \in \mathbb{N}_0$ is the count for gene $g_i$, then an $n$-th single cell is defined as a tuple $(\mathbf{x}_{\mathcal{I}_n}, \mathcal{I}_n)$. Our method, $G(\mathbf{x}, \mathcal{I}) = (\bar{\mathbf{x}}_{\mathcal{J}}, \mathcal{J})$, processes inputs as follows:

1. **Context length constraint**: We define a maximum context length $d < D$ to limit the computational complexity of downstream processing.

---

[2] We use the following notation for function compositions: $(f \circ g)(x) \stackrel{df}{=} f(g(x))$, $(f \cdot g)(x) \stackrel{df}{=} f(x)g(x)$, $(f \oplus g)(x) \stackrel{df}{=} f(x) + g(x)$, and $(f \boxplus g)(x) \stackrel{df}{=} \text{concatenate}(f(x), g(x))$.

2. **Expression-based filtering**: For each cell, we identify the set of expressed genes:

$$\mathcal{J} = \{i \in \mathcal{I} : x_i > 0\}. \quad (4)$$

3. **Context construction**: We construct a fixed-length input representation of dimension $d$. When $|\mathcal{J}| < d$ (which is typically the case due to high sparsity), we pad the input with artificial tokens to maintain consistent dimensionality (here we use $\text{Out} \stackrel{df}{=} (\bar{\mathbf{x}}_{\mathcal{J}}, \mathcal{J})$):

$$\text{Out} = \begin{cases} \{(x_i, i)\}_{i \in \mathcal{J}} \text{ if } |\mathcal{J}| = d \\ \{(x_i, i)\}_{i \in \mathcal{J}} \cup \{(0, \text{PAD})\}^{d-|\mathcal{J}|} \text{ otherwise} \end{cases} \quad (5)$$

where PAD is a special token for zero count.

This approach offers both computational and biological advantages. By excluding zero-expression genes (dropouts) from the input representation, we enable the model to focus exclusively on expressed genes, which carry the meaningful biological signal. The padding tokens serve purely as placeholders for implementation consistency and do not introduce spurious biological information, as they are explicitly marked with zero counts. This design choice aligns with the biological understanding that in single-cell data, the absence of detected expression often represents technical dropouts rather than meaningful biological zeros, making it advantageous to direct the model's attention solely to the detected expression events (Lähnemann et al., 2020). We emphasize that input filtering does not restrict the decoder's expressive capacity over zero-valued genes. The decoder is conditioned on the permutation-invariant latent $\mathbf{Z}$ rather than on the masked input directly, and parameterizes the full conditional likelihood $p(\mathbf{x}_{\mathcal{I}} \mid \eta(\mathbf{Z}, \mathcal{I}))$ over all queried gene IDs, including those with zero counts. Because the Negative Binomial distribution places substantial probability mass at zero, the family of decoder distributions $\{p(x_i \mid \eta_i(\mathbf{Z}, \mathcal{I}))\}_{i \in \mathcal{I}}$ remains capable of representing structural zeros for any gene, irrespective of whether that gene appeared in the encoder context. Filtering at the encoder is therefore a reduction of context length rather than a restriction on the model class. Table 15 and the $R^2$ Zeros scores in Table 17 confirm this empirically: the zero-filtering variant matches or exceeds the full-context baseline on reconstruction while accurately recovering gene-level sparsity patterns.

**Encoder (Variational Posterior)** We define the family of variational posteriors as Gaussians, $q(\mathbf{Z}|\phi(\mathbf{x}_{\mathcal{I}})) = \mathcal{N}(\mathbf{Z}|\mu(\mathbf{x}_{\mathcal{I}}), \sigma(\mathbf{x}_{\mathcal{I}}))$, $\phi(\mathbf{x}_{\mathcal{I}}) \stackrel{df}{=} \{\mu(\mathbf{x}_{\mathcal{I}}), \sigma^2(\mathbf{x}_{\mathcal{I}})\}$. We need $\mathbf{Z}$ to be of fixed size and invariant to permutations of $\mathbf{x}_{\mathcal{I}}$, we propose the following architecture of the encoder network: $\text{Enc}(\mathbf{x}_{\mathcal{I}}, \mathcal{I}) = (\text{T} \circ \text{MCAB}_{\mathbf{S}} \circ \text{Emb} \circ G)(\mathbf{x}_{\mathcal{I}}, \mathcal{I})$, where $\text{T} = \text{T}_L \circ \ldots \circ \text{T}_1$ is a transformer, $\text{T}_l(\cdot)$ denotes a

transformer block, e.g., $T_l(\mathbf{X}) = ((\mathrm{Id} \oplus (\mathrm{MLP} \circ \mathrm{LN}_2)) \circ (\mathrm{Id} \oplus (\mathrm{Att}_K \circ \mathrm{LN}_1)))(\mathbf{X})$, and $\mathrm{Emb}(\cdot, \cdot)$ is an embedding layer. Since inputs $\mathbf{x}_\mathcal{I}$ form a (column) vector of counts, and $\mathcal{I}$ are IDs, we propose to use the following embedding layer after applying the presented input processing: $\mathrm{Emb}(\bar{\mathbf{x}}_\mathcal{J}, \mathcal{J}) = \mathrm{Linear} \circ (\mathrm{repeat}_d(\bar{\mathbf{x}}_\mathcal{J}) \boxplus \mathbf{E}_\mathcal{J})$, where $\mathrm{repeat}_d$ repeats the counts $d$-times resulting in a matrix $M \times d$, Linear projects the concatenated $2d$-dimensional space to the $d$-dimensional space, and $\mathbf{E} \in \mathbb{R}^{M \times d}$ is the embedding matrix. The rationale behind this way of embedding both counts and indices is to mix the information and be able to learn the mixing through a projection layer.

The last transformer block duplicates the embedding dimension such that both the means $\mu$ and the variances $\sigma^2$ of a Gaussian are modeled. Alternatively, we can output means only to have an auto-encoder architecture, which is typically used in Latent Diffusion Models (Rombach et al., 2022). Note that all transformer blocks are permutation-equivariant, but our $\mathrm{MCAB_S}$ is permutation-invariant. As a result, the proposed parameterization $\mathrm{NN}_{enc}$ results in permutation-invariant variational posteriors.

**Decoder (Conditional Likelihood)** The decoder network parameterizes the conditional likelihood function $p(\mathbf{x}_\mathcal{I} | \eta(\mathbf{Z}, \mathcal{I}))$ for given latents $\mathbf{Z}$ and indices $\mathcal{I}$. The conditional likelihood could be a Gaussian if $\mathbf{x}$'s are continuous, or Poisson or Negative Binomial for counts. To fulfill the requirement on modeling exchangeable distributions, we need to ensure the conditional likelihood is exchangeable. In other words, for a given permutation $\pi$, the following holds true: $p(\mathbf{x}_\mathcal{I} | \eta(\mathbf{Z}, \mathcal{I})) = p(\mathbf{x}_{\pi(\mathcal{I})} | \eta(\mathbf{Z}, \pi(\mathcal{I})))$. First, we assume that for given $\mathbf{Z}$, the conditional likelihood is fully factorized: $p(\mathbf{x}_\mathcal{I} | \eta(\mathbf{Z}, \mathcal{I})) = \prod_{i \in I} p(\mathbf{x}_i | \eta_i(\mathbf{Z}, \mathcal{I}))$. Next, we make the parameterization of $p(\mathbf{x}_\mathcal{I} | \eta(\mathbf{Z}, \mathcal{I}))$ permutation equivariant, because, otherwise, transforming $\mathbf{Z}$ would result in incorrect parameters for each component $p(\mathbf{x}_i | \eta_i(\mathbf{Z}, \mathcal{I}))$. Keeping in mind that $\mathbf{Z}$ is permutation-invariant to permutations of $\mathbf{x}_\mathcal{I}$, we propose the following decoder network: $\mathrm{Dec}(\mathbf{Z}, \mathcal{I}) = (\mathrm{MCAB}_{\mathbf{E}_\mathcal{I}} \circ \mathrm{T})(\mathbf{Z}, \mathcal{I})$, and then use the outcomes of $\mathrm{Dec}(\mathbf{Z}, \mathcal{I})$ to parameterize an appropriate distribution, e.g., Negative Binomial (NB; see Appendix E.1 for further details) or Gaussian (GAUSS).

In our decoder network, we use $\mathrm{MCAB}_{\mathbf{E}_\mathcal{I}}$ as our final block that outputs the parameters of the conditional likelihood. To make sure the model is permutation-equivariant, we define pseudoinputs in the multi-head cross-attention block selecting embedding vectors specified by $\mathcal{I}$, $\mathbf{S} = \mathbf{E}_\mathcal{I}$, where $\mathbf{E}$ is the embedding used in the encoder network. This way, we ensure permutation-equivariance since permuting indices is equivalent to permuting embedding vectors, $\mathbf{E}_{\pi(\mathcal{I})} = \mathbf{E}_\mathcal{I}$, see Property 4 in Appendix. Eventually, we obtain a family of exchangeable conditional likelihood functions.

**Prior (Marginal over Latents)** The final component of the proposed VAE is the *prior* of latent variables. Formulating permutation-equivariant priors is challenging (Kuzina et al., 2022); fortunately, our latents $\mathbf{Z}$ are permutation-invariant and length-invariant. As a result, we can use any prior distribution, including standard Gaussian, $p(\mathbf{Z}) = \mathcal{N}(\mathbf{Z}|\mathbf{0}, \mathbf{I})$.

In this paper, we advocate to use a Latent Diffusion Model (LDM) (Rombach et al., 2022), namely, for a pre-trained VAE, we fit a diffusion-based model in the latent space to replace a *simpler* prior like $\mathcal{N}(\mathbf{Z}|\mathbf{0}, \mathbf{I})$. Using LDMs not only results in a better match with the aggregated posterior (Tomczak & Welling, 2018; Tomczak, 2024), but allows the application of controlled sampling using techniques such as classifier-free guidance (Ho & Salimans, 2022). In particular, we focus on linear interpolants and the flow matching (FM) loss (Lipman et al., 2022; Tong et al., 2024), and the following version of the classifier-free guidance for FM:

$$\tilde{v}_{t,\epsilon}(\mathbf{Z}, y) = v_{t,\epsilon}(\mathbf{Z}; \mathrm{Null}) + \omega \left[ v_{t,\epsilon}(\mathbf{Z}; y) - v_{t,\epsilon}(\mathbf{Z}; \mathrm{Null}) \right], \tag{6}$$

where $v_{t,\epsilon}(\mathbf{Z}; \cdot)$ is a parameterized vector field, and $\omega$ is the guidance strenght for attributes $\mathbf{y} \in \{0, 1\}^J$, where any combination of attributes is possible (we refer to it as *joint conditioning*); the $\mathrm{Null}$ attribute corresponds to no conditioning. In CFGen (Palma et al., 2025a), a different classifier-free guidance was used, namely, $\tilde{v}_{t,\epsilon}(\mathbf{Z}, y) = v_{t,\epsilon}(\mathbf{Z}; \mathrm{Null}) + \sum_{j=1}^J \omega_j \left[ v_{t,\epsilon}(\mathbf{Z}; y_j) - v_{t,\epsilon}(\mathbf{Z}; \mathrm{Null}) \right]$, that assumes *additive conditioning* s.t. $\sum_j y_j = 1$.

We parameterize the vector field (score) model using Diffusion Transformer (DiT) blocks (Peebles & Xie, 2023). The network is a composition of DiT and perfectly fits our modeling scenario since latents $\mathbf{Z}$ are tokens.

### 3.3. Training & Sampling

**Training** We train our model (scLDM) using the two-stage approach: (1) A VAE is trained to learn a permutation-invariant latent space by reconstructing gene expression; and (2) An LDM is trained to generate new samples from this latent space which can be controlled by classifier-free guidance (Ho & Salimans, 2022) with multiple conditions (Palma et al., 2025a).

*Stage 1: VAE* We train our VAE with a standard Gaussian prior by optimizing the ELBO in (1). However, to encourage better reconstruction capabilities, we introduce $\beta$-weighting of the KL-term like in (Higgins et al., 2017). In the most extreme case, for $\beta = 0$ the encoder returns only $\mu(\mathbf{x}_\mathcal{I})$.

*Stage 2: LDM* In the second stage, we freeze the VAE and replace the standard Gaussian prior with a score-based (diffusion) model parameterized by a DiT network trained with linear interpolants and the flow matching loss. To encourage controlled sampling, for each element of a mini-batch, we sample from the Bernoulli distribution with probability $\rho$ to determine the conditioning status.

**Sampling** In our model, sampling $\mathbf{x}$'s determined by the indices $\mathcal{I}$ is defined by the following generative process: (i) $\mathbf{Z} \sim p(\mathbf{Z})$, (ii) $\mathbf{x}_{\mathcal{I}} \sim p(\mathbf{x}_{\mathcal{I}}|\eta(\mathbf{Z},\mathcal{I}))$. We can also sample *conditionally* by applying the classifier-free guided sampling technique, following the vector field defined in (6).

## 4. Experiments

**Settings** We provide more details on the experiments in the Appendix, namely, the datasets in Appendix F, the baselines in Appendix G, the hyperparams of our scLDM in Appendix H, the evaluation pipeline with metrics in Appendix I, and additional results in Appendix K: (i) in K.1, we present a few ablation studies assessing our input process vs. no processing, presenting the performance for various hyperparameters of the VAE architecture, comparing MCAB to other aggregation operators (incl. SetTransformer pooling), (ii) some interpretability results on cross-attention scores., (iii) UMAP-based visualizations, (iv) a comparison between additive and joint conditioning in classifier-free guidance, (v) a comparison using perturbation prediction metrics. In the following, we present superior capabilities of our scLDM: (i) the powerful reconstructive performance of the fully transformer-based VAE, (ii) the unconditional and conditional generative performance on observational and perturbational datasets, (iii) the usefulness of the embeddings provided by our auto-encoder on classification downstream tasks.

### 4.1. (Un)conditional Generation on Observational Data

**Details** For the first experiment, we used single-cell RNA-sequencing data from the benchmark datasets used in (Palma et al., 2025a). Here, we are interested in evaluating the reconstructive and generative capabilities of our scLDM. For generations, we train our scLDM to synthesize gene expression profiles conditioned on a single attribute. At inference time, we query the model with specific labels to generate new synthetic cells that match the desired cellular identity. In the case of unconditional generation, we sample from the vector field without conditioning on the cell type label ($y = \text{Null}$). We compare our approach to scVI (Lopez et al., 2018) for reconstruction, and for generation with scDiffusion (Luo et al., 2024) and the current SOTA generative model CFGen (Palma et al., 2025a).

**Results and discussion** Our proposed scLDM model demonstrates substantial improvements over existing approaches across all evaluated datasets and metrics, see Table 1. scLDM consistently achieves the lowest reconstruction error values, with particularly notable improvements on Tabula Muris (4993.6 vs. 5547.6 for CFGen) and HLCA (4898.9 vs. 5428.7 for CFGen) datasets. The Pearson correlation coefficients show dramatic improvements, with scLDM achieving 0.376 on Tabula Muris compared

*Table 1.* Model performance comparison on cell reconstruction task. Reported values are mean and standard error. Results for CFGen are calculated based on a checkpoint provided in the official repository of (Palma et al., 2025a).

| Dataset | Model | RE ↓ | PCC ↑ | MSE ↓ |
|---|---|---|---|---|
| Dentate Gyrus | scVI | $5193.2 \pm 0.1$ | $0.058 \pm 0.000$ | $0.378 \pm 0.000$ |
| | CFGen | $5468.8 \pm \text{N/A}$ | $0.076 \pm \text{N/A}$ | $0.253 \pm \text{N/A}$ |
| | scLDM (NB) | $\mathbf{4571.6} \pm 26.5$ | $\mathbf{0.273} \pm 0.005$ | $\mathbf{0.206} \pm 0.002$ |
| Tabula Muris | scVI | $5588.2 \pm 1.0$ | $0.221 \pm 0.000$ | $0.132 \pm 0.000$ |
| | CFGen | $5547.6 \pm \text{N/A}$ | $0.136 \pm \text{N/A}$ | $0.127 \pm \text{N/A}$ |
| | scLDM (NB) | $\mathbf{4993.6} \pm 25.1$ | $\mathbf{0.376} \pm 0.006$ | $\mathbf{0.106} \pm 0.001$ |
| HLCA | scVI | $5659.2 \pm 0.3$ | $0.125 \pm 0.000$ | $0.238 \pm 0.000$ |
| | CFGen | $5428.7 \pm \text{N/A}$ | $0.146 \pm \text{N/A}$ | $0.117 \pm \text{N/A}$ |
| | scLDM (NB) | $\mathbf{4898.9} \pm 12.4$ | $\mathbf{0.310} \pm 0.003$ | $\mathbf{0.095} \pm 0.001$ |

to 0.221 for scVI and 0.136 for CFGen, nearly doubling the correlation with ground truth. Similarly, MSE is consistently reduced, with scLDM achieving 0.095 on HLCA compared to 0.117 for CFGen and 0.238 for scVI. These results suggest that our fully transformer-based VAE captures the complex structure of single-cell gene expression data more effectively than traditional VAE-based methods (scVI, CFGen). The consistent improvements across diverse tissue types (brain, whole organism, and lung) indicate the generalizability of our approach, namely, parameterizing the VAE with the proposed transformer architectures.

*Table 2.* Model performance comparison on (un)conditional cell generation benchmarks on highly variable genes.

| Setting | Model | W2 ↓ | MMD² RBF ↓ | 1-NN → 0.5 | Prec ↑ | Rec ↑ |
|---|---|---|---|---|---|---|
| | | | Dentate Gyrus | | | |
| Uncond | scDiffusion | $17.443 \pm 0.019$ | $0.258 \pm 0.001$ | $0.989 \pm 0.000$ | $0.367 \pm 0.005$ | $0.007 \pm 0.005$ |
| | CFGen | $12.617 \pm 0.024$ | $0.022 \pm 0.000$ | $0.856 \pm 0.002$ | $0.278 \pm 0.002$ | $\mathbf{0.385} \pm 0.011$ |
| | scLDM (NB) | $\mathbf{10.710} \pm 0.034$ | $\mathbf{0.017} \pm 0.000$ | $\mathbf{0.709} \pm 0.002$ | $\mathbf{0.664} \pm 0.001$ | $0.291 \pm 0.004$ |
| | scLDM (Gauss) | $17.678 \pm 0.022$ | $0.185 \pm 0.001$ | $0.991 \pm 0.000$ | $0.212 \pm 0.004$ | $0.000 \pm 0.000$ |
| Cond | scDiffusion | $17.321 \pm 0.236$ | $0.689 \pm 0.015$ | $0.983 \pm 0.002$ | $0.434 \pm 0.032$ | $0.000 \pm 0.000$ |
| | CFGen | $11.608 \pm 0.283$ | $\mathbf{0.075} \pm 0.008$ | $0.741 \pm 0.007$ | $0.340 \pm 0.016$ | $\mathbf{0.496} \pm 0.011$ |
| | scLDM (NB) | $\mathbf{10.485} \pm 0.271$ | $0.084 \pm 0.008$ | $\mathbf{0.689} \pm 0.007$ | $\mathbf{0.643} \pm 0.016$ | $0.276 \pm 0.015$ |
| | scLDM (Gauss) | $18.034 \pm 0.242$ | $0.570 \pm 0.014$ | $0.990 \pm 0.001$ | $0.225 \pm 0.018$ | $0.000 \pm 0.000$ |
| | | | Tabula Muris | | | |
| Uncond | scDiffusion | $14.143 \pm 0.009$ | $0.144 \pm 0.001$ | $0.982 \pm 0.000$ | $0.297 \pm 0.003$ | $0.007 \pm 0.001$ |
| | CFGen | $11.658 \pm 0.156$ | $0.008 \pm 0.000$ | $0.773 \pm 0.003$ | $0.255 \pm 0.002$ | $0.591 \pm 0.010$ |
| | scLDM (NB) | $\mathbf{7.267} \pm 0.108$ | $\mathbf{0.002} \pm 0.000$ | $\mathbf{0.596} \pm 0.004$ | $\mathbf{0.539} \pm 0.003$ | $\mathbf{0.608} \pm 0.005$ |
| | scLDM (Gauss) | $14.670 \pm 0.036$ | $0.099 \pm 0.001$ | $0.952 \pm 0.000$ | $0.083 \pm 0.003$ | $0.146 \pm 0.005$ |
| Cond | scDiffusion | $14.890 \pm 2.349$ | $0.452 \pm 0.231$ | $0.989 \pm 0.016$ | $0.359 \pm 0.207$ | $0.008 \pm 0.011$ |
| | CFGen | $8.921 \pm 2.018$ | $0.026 \pm 0.016$ | $0.754 \pm 0.045$ | $0.228 \pm 0.066$ | $\mathbf{0.644} \pm 0.029$ |
| | scLDM (NB) | $\mathbf{6.789} \pm 1.804$ | $\mathbf{0.010} \pm 0.005$ | $\mathbf{0.615} \pm 0.026$ | $\mathbf{0.475} \pm 0.037$ | $0.569 \pm 0.034$ |
| | scLDM (Gauss) | $14.871 \pm 2.676$ | $0.330 \pm 0.203$ | $0.955 \pm 0.081$ | $0.097 \pm 0.088$ | $0.116 \pm 0.227$ |
| | | | HLCA | | | |
| Uncond | scDiffusion | $15.886 \pm 0.047$ | $0.163 \pm 0.002$ | $0.946 \pm 0.001$ | $\mathbf{0.788} \pm 0.003$ | $0.008 \pm 0.000$ |
| | CFGen | $12.433 \pm 0.056$ | $0.007 \pm 0.000$ | $0.760 \pm 0.001$ | $0.272 \pm 0.004$ | $0.583 \pm 0.004$ |
| | scLDM (NB) | $\mathbf{9.272} \pm 0.020$ | $\mathbf{0.004} \pm 0.000$ | $\mathbf{0.605} \pm 0.003$ | $0.540 \pm 0.005$ | $\mathbf{0.622} \pm 0.005$ |
| | scLDM (Gauss) | $16.487 \pm 0.024$ | $0.121 \pm 0.001$ | $0.942 \pm 0.001$ | $0.458 \pm 0.005$ | $0.067 \pm 0.001$ |
| Cond | scDiffusion | $14.056 \pm 3.792$ | $0.519 \pm 0.227$ | $0.948 \pm 0.039$ | $\mathbf{0.691} \pm 0.232$ | $0.014 \pm 0.033$ |
| | CFGen | $9.758 \pm 2.438$ | $0.090 \pm 0.136$ | $0.731 \pm 0.060$ | $0.254 \pm 0.080$ | $\mathbf{0.612} \pm 0.107$ |
| | scLDM (NB) | $\mathbf{8.303} \pm 1.842$ | $\mathbf{0.066} \pm 0.112$ | $\mathbf{0.617} \pm 0.047$ | $0.474 \pm 0.105$ | $0.602 \pm 0.093$ |
| | scLDM (Gauss) | $14.493 \pm 3.521$ | $0.408 \pm 0.217$ | $0.906 \pm 0.117$ | $0.318 \pm 0.200$ | $0.149 \pm 0.251$ |

Table 2 summarizes unconditional and conditional generation results across three datasets using Wasserstein-2, MMD² (RBF), 1-NN accuracy, and precision–recall metrics. In the unconditional setting, scLDM achieves the lowest or Wasserstein-2 distance and consistently smaller MMD² values than scDIFFUSION and CFGEN, indicating improved distributional matching. Its 1-NN accuracy is closer to the ideal value of 0.5, reflecting better overlap between generated and real samples. In terms of sample quality and coverage, scLDM exhibits a more balanced precision–recall trade-off, whereas CFGEN tends toward high recall but low precision. In the conditional setting, scLDM maintains this advantage, outperforming competing methods. We report evaluation for all genes in Table 16,

and various ablations in Appendix K.1.

In Figure 2, we report qualitative evaluations of conditional generation results for the HLCA datasets for all three models. Our model shows qualitatively a better coverage of the cell state variation on UMAP coordinates, showcasing how it is able to recapitulate high resolution cell states in highly heterogenous tissues like the human lung. Additionally, in Appendinx K.2 we provide an interpretability analysis on the cross-attention scores of the encoder-decoder model of scLDM, showing how the latent tokens map to specific marker gene set patterns. These results demonstrate that our latent diffusion approach not only generates more realistic single-cell expression profiles but also captures relevant biological information determining cellular diversity.

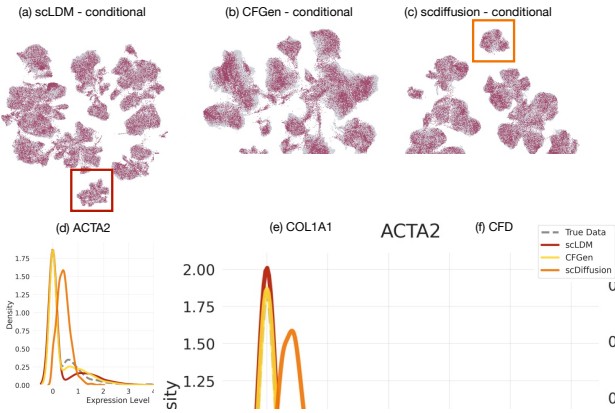

*Figure 2.* Conditional generation for the HLCA dataset for: (a) scLDM, (b) CFGen and (c) scdiffusion. Expression levels for 3 marker genes: (d) ACTA2, (e) COL1A1 and (f) CFD, markers of "alveolar type 2 fibroblast cell", corresponding to cell populations in the insets.

### 4.2. Conditional Generation on Perturbational Data

**Details** In the second experiment, we train our model for conditional gene expression generation based on multiple attributes: a cell context (cell lines and cell types) and a perturbation type (gene knockouts and cytokines). The VAE baseline is trained without attribute conditioning, focusing solely on the reconstruction objective, while the flow matching component incorporates multi-attribute conditioning. By training across diverse contexts, the model learns to capture joint structure spanning different axes of variation. At inference time, the flow matching model is queried with specific combinations of cell type and perturbation to generate new gene expression profiles.

We leverage two datasets: (1) Parse 1M, containing perturbational single-cell RNA-sequencing data from human peripheral blood mononuclear cells (PBMCs) generated by Parse Biosciences (par) with 1,267,690 single cells across 18 annotated cell types, each subjected to one of 90 cytokine

perturbations or a control condition, and to test generalization capabilities, we hold out 27 cytokine perturbations in CD4 Naive cells; and (2) Replogle, a benchmark genetic perturbation dataset (Nadig et al., 2025) consisting of 2,024 gene knockouts across four cell lines after filtering perturbations with low on-target efficacy Adduri et al. (2025), holding out 372 genetic perturbations in HepG2 cells to evaluate generalization to unseen cell context–perturbation pairs. For both datasets, we restricted analysis to the top 2,000 highly variable genes (HVGs) following Adduri et al. (2025), under which several baselines were originally evaluated. We emphasize that this restriction is driven by the evaluation protocol rather than by any architectural constraint of scLDM. We compare our model against established baselines: CPA (Lotfollahi et al., 2023a), scVI (Lopez et al., 2018), scGPT (Cui et al., 2024), STATE-Tx (Adduri et al., 2025) and CellFlow (Klein et al., 2025).

*Table 3.* Model performance comparison on conditional cell generation on Parse1M and Replogle.

| Dataset | Model | W2 ↓ | MMD² RBF ↓ | FD ↓ | 1-NN → 0.5 | Precision ↑ | Recall ↑ |
|---|---|---|---|---|---|---|---|
| Parse 1M | scVI | 35.508 ± 0.182 | 1.372 ± 0.016 | 1233.109 ± 12.694 | 0.995 ± 0.000 | 0.059 ± 0.006 | 0.000 ± 0.000 |
| | CPA | 13.534 ± 0.036 | 1.117 ± 0.014 | 181.324 ± 0.985 | 0.988 ± 0.001 | **0.960** ± 0.013 | 0.000 ± 0.000 |
| | Cellflow | **11.836** ± 0.063 | 0.015 ± 0.002 | 9.443 ± 1.238 | 0.579 ± 0.008 | 0.678 ± 0.009 | 0.490 ± 0.003 |
| | scGPT | 22.870 ± 0.152 | 2.203 ± 0.013 | 523.932 ± 7.043 | 1.000 ± 0.000 | 0.582 ± 0.055 | 0.000 ± 0.000 |
| | STATE-Tx | 19.111 ± 0.137 | 0.714 ± 0.009 | 312.344 ± 5.743 | 0.993 ± 0.001 | 0.654 ± 0.031 | 0.000 ± 0.000 |
| | scLDM (NB, ω=1) | 12.346 ± 0.046 | 0.020 ± 0.002 | 13.163 ± 0.922 | 0.581 ± 0.005 | 0.577 ± 0.006 | 0.626 ± 0.002 |
| | scLDM (Gauss, ω=1) | 12.374 ± 0.052 | **0.008** ± 0.001 | **4.695** ± 0.637 | **0.535** ± 0.004 | 0.459 ± 0.005 | **0.708** ± 0.002 |
| Replogle | scVI | 17.359 ± 0.051 | 0.453 ± 0.003 | 284.474 ± 1.825 | 0.916 ± 0.001 | 0.117 ± 0.004 | 0.001 ± 0.000 |
| | CPA | 11.510 ± 0.029 | 0.532 ± 0.043 | 126.805 ± 0.693 | 0.873 ± 0.002 | **0.939** ± 0.004 | 0.000 ± 0.000 |
| | Cellflow | **10.684** ± 0.046 | 0.289 ± 0.003 | 73.358 ± 0.977 | 0.715 ± 0.002 | 0.943 ± 0.003 | 0.067 ± 0.002 |
| | scGPT | 34.166 ± 0.272 | 3.087 ± 0.010 | 1247.678 ± 20.245 | 1.000 ± 0.000 | 0.025 ± 0.005 | 0.000 ± 0.000 |
| | STATE-Tx | 20.821 ± 0.040 | 0.731 ± 0.003 | 366.642 ± 1.547 | 0.990 ± 0.000 | 0.275 ± 0.008 | 0.003 ± 0.000 |
| | scLDM (NB, ω=1) | 11.554 ± 0.034 | 0.192 ± 0.002 | 54.414 ± 0.673 | 0.653 ± 0.002 | 0.667 ± 0.003 | 0.580 ± 0.003 |
| | scLDM (Gauss, ω=1) | 11.612 ± 0.035 | **0.168** ± 0.002 | **47.027** ± 0.732 | **0.591** ± 0.002 | 0.559 ± 0.004 | **0.733** ± 0.003 |

**Results and Discussion** Table 3 summarizes conditional generation performance on perturbational datasets (Parse 1M and Replogle) using Wasserstein-2, MMD² (RBF), Frechet Distance, 1-NN accuracy, and precision–recall metrics. Across both datasets, sCLDM is competitive to baselines in terms of distributional similarity, achieving the lowest or near-lowest averages across the metrics considered. Moreover, sCLDM exhibits a more balanced precision–recall trade-off than competing methods, as well as lower 1-NN accuracy, avoiding mode collapse while maintaining high sample fidelity. Table 22 reports the same metrics restricted to differentially expressed genes, where sCLDM remains competitive compared to baselines. Appendix K.6 presents perturbation prediction evaluations from (Adduri et al., 2025); here, sCLDM is the top-performing method in the Parse 1M dataset, and best or second-best in the Replogle dataset, while models like CELLFLOW and CPA, despite strong generative performance, fall short on these perturbation-specific measures.

In Figure 3, we report a qualitative evaluation of our model generative performances for the Parse 1M dataset for unseen combinations of CD4-Naive cells with various cytokine perturbations like IL-9 and LT-alpha1-beta2. Furthermore, we show the same for Replogle dataset for unseen combinations of HepG2 cells with PPP6c and ZDHHC7 gene edits.

In Table 18 (Appendix), we report reconstruction results

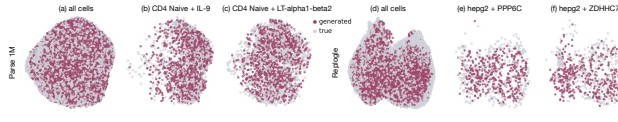

*Figure 3.* Conditional generation across multiple attributes: cell type and perturbation. (a) Generated vs. true cells across all cell types in the Parse 1M dataset show close alignment. (b–c) For CD4 Naive cells, conditioning on cytokine perturbations (IL-9, LT-alpha1-beta2) produces perturbation-specific shifts consistent with the true test distributions. (d) Generated vs. true cells across all cell types in the Replogle dataset. (e–f) For HepG2 cells, conditioning on genetic perturbations (PPP6C, ZDHHC7) yields realistic perturbation-dependent distributions that closely follow the experimental data.

*Table 4.* COVID-19 model performance comparison (averaged across all donors). Since all standard errors are below 0.003 (see Appendix K.8), they are omitted in this table.

| Model | F1 Score | Recall | Precision |
| --- | --- | --- | --- |
| TranscriptFormer | 0.814 | 0.829 | 0.801 |
| UCE | 0.775 | 0.781 | 0.771 |
| scGPT | 0.779 | 0.793 | 0.766 |
| Geneformer | 0.768 | 0.781 | 0.757 |
| AIDO.Cell | 0.717 | 0.729 | 0.708 |
| scVI | 0.675 | 0.680 | 0.680 |
| scLDM (20M) | 0.811 | 0.827 | 0.797 |
| scLDM (70M) | 0.815 | 0.830 | 0.801 |
| scLDM (270M) | **0.820** | **0.836** | **0.806** |

between scLDM and scVI for both datasets, showing how our improved transformer-based VAE significantly outperforms MLP-based scVI on the reconstruction task. We also tested how the additive conditioning for the classifier-free guidance proposed in (Palma et al., 2025a) compares to the standard classifier-free guidance approach (Ho & Salimans, 2022). In Table 22 (Appendix), we report that the standard approach is superior to the additive approach in multi-attribute conditional settings for perturbational data.

### 4.3. scLDM-VAE embedding evaluations on classification tasks

**Details** For the third experiment, we leveraged two datasets: the first dataset is a human lung single-cell RNA-sequencing data from healthy donors and patients affected by COVID-19 (Wu et al., 2024), the second dataset consists of 6 tissues from the Tabula Sapiens 2.0 (Tabula Sapiens Consortium & Quake, 2025). The goal of this experiment is to verify the quality of embeddings provided by the auto-encoder on a downstream task (here: classification). We compare our approach to embeddings provided by TranscriptFormer (Pearce et al., 2025), scVI (Lopez et al., 2018), AIDO.Cell (Ho et al., 2024), Geneformer (Theodoris et al., 2023), scGPT (Cui et al., 2024), UCE (Rosen et al., 2023). We used Human Census data (CellxGene)[3] to train three versions of scLDM-VAE, namely, with around 20M parameters, 70M parameters, and 270M parameters. For our scLDM-VAE and benchmark models, both datasets represent out-of-distribution data that were unseen during training.

To evaluate the quality of the learned representations, we process each of the four COVID-19 donors through the models to generate cell embeddings. For scLDM variants, we use the mean of the latent distribution, $\mu(\mathbf{x})$, which is flattened to a 4096-dimensional vector. To ensure fair comparison across models with different embedding dimensions, we apply principal component analysis (PCA) to all embeddings, retaining the top 128 principal components.

---

[3] https://cellxgene.cziscience.com/

For each donor independently, we train an unregularized logistic regression classifier to distinguish infected from uninfected cells using 5-fold cross-validation. The final metrics are computed as equally weighted averages across the four donors ($n = 4$), with uncertainties propagated using standard error addition in quadrature: $\sigma_{\text{combined}} = \frac{1}{n}\sqrt{\sum_{i=1}^{n}\sigma_i^2}$.

For the Tabula Sapiens 2.0 dataset, we evaluated cell type classification across 6 tissues: blood, spleen, lymph node, small intestine, thymus, and liver. Following the same protocol as the COVID-19 analysis, we stratified samples by tissue instead of donor and filtered out cell types with fewer than 100 cells to ensure robust classification. We employed multinomial logistic regression for the multi-class cell type prediction task. Final metrics are averaged over tissues with propagated uncertainties (see Appendix K.8).

*Table 5.* Tabula Sapiens 2.0 model performance comparison (averaged across all tissues). Since all standard errors are below 0.003 (see Appendix K.8), they are omitted in this table.

| Model | F1 Score | Recall | Precision |
| --- | --- | --- | --- |
| scGPT | 0.8 | 0.802 | 0.806 |
| scVI | 0.799 | 0.794 | 0.814 |
| TranscriptFormer | 0.799 | 0.8 | 0.802 |
| UCE | 0.796 | 0.797 | 0.801 |
| Geneformer | 0.777 | 0.776 | 0.786 |
| AIDO.Cell | 0.724 | 0.715 | 0.748 |
| scLDM (20M) | **0.804** | **0.805** | **0.812** |
| scLDM (70M) | 0.802 | 0.802 | 0.810 |
| scLDM (270M) | 0.802 | 0.803 | 0.811 |

**Results and discussion** As shown in Table 4, our 270M and 70M models achieve superior performance across all evaluated metrics for COVID-19 infection detection. The performance differences between scLDM (270M) and TranscriptFormer—the strongest baseline—represent meaningful differences given the measurement uncertainty, with our model achieving F1 score of $0.820 \pm 0.001$ compared to TranscriptFormer's $0.814 \pm 0.002$. The strong discriminative performance demonstrates that our transformer-based

VAE learns biologically meaningful representations that capture infection-related transcriptional signatures. We observe substantial improvements over the VAE-based scVI model (F1: $0.675 \pm 0.001$), highlighting the advantages of our architectural innovations and model scale.

For the Tabula Sapiens 2.0 classification results shown in Table 5, the differences in F1 scores between the scLDM model variants are within measurement uncertainty and may not be significant. Moreover, all top-performing models—scLDM variants, scGPT, scVI, and Transcript-Former—achieve F1 scores within each other's uncertainties (ranging from $0.799$ to $0.804$ with standard errors of $0.002$), indicating comparable performance for multi-class cell type classification. The consistent performance across both binary (COVID-19 infection) and multi-class (cell type) classification tasks validates the biological utility of our learned embeddings, making them valuable for biological discovery applications beyond generation.

## 5. Limitations

While scLDM achieves strong performance across observational, perturbational, and downstream classification benchmarks, several limitations remain. First, the relative performance of the Gaussian and Negative Binomial likelihoods is task-dependent, and the choice of likelihood currently requires per-task selection rather than being learned end-to-end. Second, our framework targets the transcriptomic modality and does not currently integrate complementary single-cell measurements such as chromatin accessibility or surface protein abundance; extending the MCAB design to multi-modal exchangeable data is a natural direction.

## 6. Conclusion

In this paper, we introduced a scalable architecture that combines a permutation-invariant encoder and a permutation-equivariant decoder within a fully transformer-based VAE with a latent diffusion model parameterized using DiTs, achieving state-of-the-art performance on cell generation benchmarks, both observational and perturbational data, as well as downstream classification tasks. Our work extends beyond imposing artificial structure on gene expression data, instead providing a principled framework for learning exchangable models. Importantly, the proposed design naturally supports flexible conditioning and generalization across unseen contexts, enabling faithful modeling of complex cellular responses. This approach is not limited to transcriptomics and lays the groundwork for developing foundational models for other exchangeable biological data, such as multi-omics and multi-modal data.

## Impact Statement

This paper presents work whose goal is to advance the field of Machine Learning, with applications to single-cell genomics. Generative models for single-cell data have the potential to accelerate therapeutic development by enabling in silico perturbation experiments, reducing reliance on expensive and time-consuming wet-lab screens. While we do not foresee direct negative societal consequences, we acknowledge that, as with any predictive model in biology, outputs should be validated experimentally before informing clinical decisions.

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

## A. LLM Usage Disclaimer

We used large language models (ChatGPT and Claude) to assist with manuscript polishing, including grammar and clarity improvements, and to verify technical definitions and terminology. All scientific content, analysis, and conclusions are the original work of the authors.

## B. Definitions

**Self-attention**   Attention mechanism is defined as $\mathrm{Att}(\mathbf{Q}, \mathbf{K}, \mathbf{V}) \stackrel{df}{=} \mathrm{softmax}(\mathbf{Q}\mathbf{K}^\top)\mathbf{V}$, where $\mathbf{Q} \in \mathbb{R}^{m \times D}, \mathbf{K} \in \mathbb{R}^{M \times D}, \mathbf{V} \in \mathbb{R}^{M \times D}$, and $\mathrm{softmax}(\cdot)$ is the row-wise softmax function.[4] Typically, $\mathbf{Q}$, $\mathbf{K}$ and $\mathbf{V}$ are results of linear transformations of given inputs. If all matrices are calculated based on the same input $\mathbf{X}$, we refer to it as *self-attention*.

**Cross-attention**   However, if $\mathbf{Q}$ is calculated using $\mathbf{X}$, and another input $\mathbf{Y}$ is used to obtain $\mathbf{K}$ and $\mathbf{V}$, then we refer to it as *cross-attention*.

**Multi-head attention**   $\mathrm{Att}_K$ denotes a multi-head attention with $K$ heads, i.e., a concatenation of $K$ attention layers.

**Multi-head Attention Block (MAB)**   A multi-head attention block (MAB) is defined as follows (Lee et al., 2019; Zhang et al., 2022):

$$\mathrm{MAB}(\mathbf{X}, \mathbf{Y}) \stackrel{df}{=} f(\mathbf{X}, \mathbf{Y}) + \mathrm{relu}(f(\mathbf{X}, \mathbf{Y})\mathbf{W}), \tag{7}$$

$$f(\mathbf{X}, \mathbf{Y}) \stackrel{df}{=} \mathbf{X}\mathbf{W}_f + \mathrm{Att}_K(\mathbf{Q}(\mathbf{X}), \mathbf{K}(\mathbf{Y}), \mathbf{V}(\mathbf{Y})),$$

where $f(\mathbf{X}, \mathbf{Y}) \stackrel{df}{=} \mathbf{X}\mathbf{W}_f + \mathrm{Att}_K(\mathbf{Q}(\mathbf{X}), \mathbf{K}(\mathbf{Y}), \mathbf{V}(\mathbf{Y}))$, $\mathbf{W}$ and $\mathbf{W}_f$ are learnable weight matrices.

**Set Attention Block (SAB)**   Set Attention Block (SAB) is defined as follows (Lee et al., 2019):

$$\mathrm{SAB}(\mathbf{X}) \stackrel{df}{=} \mathrm{MAB}(\mathbf{X}, \mathbf{X}). \tag{8}$$

**Induced Set Attention Block (ISAB)**   Induced Set Attention Block (ISAB) is defined as follows (Lee et al., 2019):

$$\mathrm{ISAB}(\mathbf{X}) \stackrel{df}{=} \mathrm{MAB}\left(\mathbf{X}, \mathrm{MAB}(\mathbf{U}, \mathbf{X})\right), \tag{9}$$

where $\mathbf{U} \in \mathbb{R}^{m \times D}$ are *inducing points*, i.e., a global weight matrix learnable by backpropagation.

**Pooling by Multi-Head Attention**   Pooling by Multi-head Attention (PMA) is defined as follows:

$$\mathrm{PMA}(\mathbf{X}) = \mathrm{MAB}(\mathbf{S}, \mathrm{rFF}(\mathbf{X})), \tag{10}$$

where $\mathbf{S} \in \mathbb{R}^{m \times D}$ is a matrix of learnable *inducing points* (or *pseudoinputs*), and $\mathrm{rFF} : \mathbb{R}^{M \times D} \to \mathbb{R}^{M \times D}$ is a row-wise linear layer. For fixed inducing points $\mathbf{S}$ in PMA, this layer is permutation-invariant (this is due to applying the attention layer, see Property 3 in Appendix C).

## C. Permutation-equivariance and permutation-invariance of attention mechanism

The row-wise self-attention function fulfills the following property:

*Property* 1. For a given matrix $\mathbf{X} \in \mathbb{R}^{M \times D}$, and two permutation matrices $\mathbf{P}_M \in \{0, 1\}^{M \times M}$ and $\mathbf{P}_D \in \{0, 1\}^{D \times D}$, the following statements hold true for the row-wise softmax function: (i) $\mathrm{softmax}(\mathbf{X}\mathbf{P}_D^\top) = \mathrm{softmax}(\mathbf{X})\mathbf{P}_D^\top$, (ii) $\mathrm{softmax}(\mathbf{P}_M\mathbf{X}) = \mathbf{P}_M\mathrm{softmax}(\mathbf{X})$.

Property 1 tells us that applying the permutation to the softmax function just reorders its columns/rows, hence, applying softmax before or after the reordering gives the same vector, merely shuffled.

---

[4]We skip scaling $\mathbf{Q}\mathbf{K}^\top$ by $1/\sqrt{D}$ to avoid unnecessary clutter.

Before we move to next properties, we recall that any permutation matrix is orthogonal, hence, $\mathbf{P}\mathbf{P}^\top = \mathbf{P}^\top\mathbf{P} = \mathbf{I}$.

It is a well-known fact that the (self-)attention mechanism is permutation-equivariant, namely, the following property holds true:

*Property* 2. For a given permutation matrix $\mathbf{P} \in \{0,1\}^{M \times M}$, the attention mechanism is permutation-equivariant, i.e., $\text{Att}\left(\mathbf{PQ}(\mathbf{X}), \mathbf{PK}(\mathbf{X}), \mathbf{PV}(\mathbf{X})\right) = \mathbf{P}\,\text{Att}\left(\mathbf{Q}(\mathbf{X}), \mathbf{K}(\mathbf{X}), \mathbf{V}(\mathbf{X})\right)$.

*Proof.* First, we notice that: $\mathbf{Q}(\mathbf{PX}) = \mathbf{PXW}_Q = \mathbf{PQ}(\mathbf{X})$, $\mathbf{K}(\mathbf{PX}) = \mathbf{PXW}_K = \mathbf{PK}(\mathbf{X})$ and $\mathbf{V}(\mathbf{PX}) = \mathbf{PXW}_V = \mathbf{PV}(\mathbf{X})$. Next, to avoid unnecessary clutter, let us skip the dependency on $\mathbf{X}$. Then:

$$
\begin{aligned}
\text{Att}\left(\mathbf{PQ}, \mathbf{PK}, \mathbf{PV}\right) &= \text{softmax}(\mathbf{PQ}(\mathbf{PK})^\top)\mathbf{PV} \\
&= \text{softmax}(\mathbf{PQK}^\top\mathbf{P}^\top)\mathbf{PV} \\
&= \mathbf{P}\,\text{softmax}(\mathbf{QK}^\top)\mathbf{P}^\top\mathbf{PV} \\
&= \mathbf{P}\,\text{softmax}(\mathbf{QK}^\top)\mathbf{V} \\
&= \mathbf{P}\,\text{Att}\left(\mathbf{Q}, \mathbf{K}, \mathbf{V}\right).
\end{aligned}
$$

$\square$

The attention mechanism becomes permutation-invariant for $\mathbf{Q}$ being a global parameter matrix only if the following property holds true:

*Property* 3. For a given permutation matrix $\mathbf{P} \in \{0,1\}^{M \times M}$, the attention mechanism with inducing points $\mathbf{Q} \in \mathbb{R}^{m \times D}$ is permutation-invariant, i.e., $\text{Att}\left(\mathbf{Q}, \mathbf{PK}(\mathbf{X}), \mathbf{PV}(\mathbf{X})\right) = \text{Att}\left(\mathbf{Q}, \mathbf{K}(\mathbf{X}), \mathbf{V}(\mathbf{X})\right)$.

*Proof.* First, we notice that: $\mathbf{K}(\mathbf{PX}) = \mathbf{PXW}_K = \mathbf{PK}(\mathbf{X})$ and $\mathbf{V}(\mathbf{PX}) = \mathbf{PXW}_V = \mathbf{PV}(\mathbf{X})$. Next, to avoid unnecessary clutter, let us skip the dependency on $\mathbf{X}$. Then:

$$
\begin{aligned}
\text{Att}\left(\mathbf{Q}, \mathbf{PK}, \mathbf{PV}\right) &= \text{softmax}(\mathbf{Q}(\mathbf{PK})^\top)\mathbf{PV} \\
&= \text{softmax}(\mathbf{QK}^\top\mathbf{P}^\top)\mathbf{PV} \\
&= \text{softmax}(\mathbf{QK}^\top)\mathbf{P}^\top\mathbf{PV} \\
&= \text{softmax}(\mathbf{QK}^\top)\mathbf{V} \\
&= \text{Att}\left(\mathbf{Q}, \mathbf{K}, \mathbf{V}\right).
\end{aligned}
$$

$\square$

However, for a latent matrix $\mathbf{Z} \in \mathbb{R}^{m \times D}$ obtained by transforming $\mathbf{X}$ in the permutation-invariant manner, an embedding matrix $\mathbf{E} \in \mathbb{R}^{V \times D}$, the inducing points determined by a set of indices $\mathcal{I}$, i.e., $\mathbf{Q} = \mathbf{E}_\mathcal{I}$, is permutation-equivariant if the indices $\mathcal{I}$ are permuted, i.e., a permutation of indices $\pi(\mathcal{I})$ induces the matrix $\mathbf{P}$, thus, $\mathbf{PQ} = \mathbf{E}_{\pi(\mathcal{I})}$. Then, the following property holds true:

*Property* 4. For a given permutation of indices $\mathcal{I}$, $\pi(\mathcal{I})$, or, equivalently, a matrix permutation $\mathbf{P} \in \{0,1\}^{|\mathcal{I}| \times |\mathcal{I}|}$, and latents $\mathbf{Z}$ calculated by a permutation-invariant function $f$, i.e., $\mathbf{Z} = f(\mathbf{PX})$, the attention mechanism with inducing points $\mathbf{Q} \in \mathbb{R}^{|\mathcal{I}| \times D}$ is permutation-equivariant, i.e., $\text{Att}\left(\mathbf{PQ}, \mathbf{K}(f(\mathbf{PX})), \mathbf{V}(f(\mathbf{PX}))\right) = \mathbf{P}\text{Att}\left(\mathbf{Q}, \mathbf{K}(\mathbf{Z}), \mathbf{V}(\mathbf{Z})\right)$.

*Proof.* First, since $f$ is permutation-invariance, we get $\mathbf{Z} = f(\mathbf{PX}) = f(\mathbf{X})$. Second, we note that $\mathbf{K}(\mathbf{PZ}) = \mathbf{PZW}_K = \mathbf{PK}(\mathbf{Z})$ and $\mathbf{V}(\mathbf{PZ}) = \mathbf{PZW}_V = \mathbf{PV}(\mathbf{Z})$. Then:

$$
\begin{aligned}
\text{Att}\left(\mathbf{PQ}, \mathbf{K}(f(\mathbf{PX})), \mathbf{V}(f(\mathbf{PX}))\right) &= \text{softmax}(\mathbf{PQK}(f(\mathbf{PX}))^\top)\mathbf{V}(f(\mathbf{PX})) \\
&= \mathbf{P}\,\text{softmax}(\mathbf{QK}(f(\mathbf{X}))^\top)\mathbf{V}(f(\mathbf{X})) \\
&= \mathbf{P}\,\text{softmax}(\mathbf{QK}(\mathbf{Z})^\top)\mathbf{V}(\mathbf{Z}) \\
&= \mathbf{P}\,\text{Att}\left(\mathbf{Q}, \mathbf{K}(\mathbf{Z}), \mathbf{V}(\mathbf{Z})\right).
\end{aligned}
$$

$\square$

# D. Related Work (extended)

Modeling a probability distribution over order-agnostic objects like sets is challenging for at least two reasons. First, a model must be permutation-equivariant, meaning, changing the order of variables changes the order of parameters as well. Second, the model must also be exchangeable. Additionally, in the case of gene expression, for various tissues, we get different subsets of genes, thus, ideally, we would like to learn a single model to *transfer* hidden dependencies (correlations) among cells from distinct tissues.

**Autoregressive Models**   A varying-size objects are typically modeled by autoregressive models (ARMs) like transformer-based LLMs for text (Vaswani et al., 2017) or WaveNet for audio (Van Den Oord et al., 2016). However, ARMs assume a fixed order of variables, otherwise, like in the case of sets, their performance can drop significantly. Recently, it has been shown that misspecifying the order in ARMs can result in a huge drop in their performance (Kim et al., 2025). There are ways of dealing with the order in ARMs (Pannatier et al., 2024), but they are not well-suited for processing objects without an explicitly defined order.

**Masked Diffusion Models**   Recently, a masked version of diffusion-based models (Ho et al., 2020) are used to generate text quite successfully (Nie et al., 2025) since they can alleviate the need of specifying the order of generation. However, as proven in (Kim et al., 2025), masked diffusion-based models are order-agnostic but at the price of learning an extremely complex task of predicting a variable value conditioned on a set of unmasked variables in arbitrary positions.

**Variational Auto-Encoders**   Another modeling approach is to define a Variational Auto-Encoder (Kingma & Welling, 2022; Rezende et al., 2014) since this framework allows defining its components in a flexible manner. In (Kim et al., 2021), a SetVAE was formulated by introducing two separate latent variables to deal with varying size of sets, namely, $\mathbf{z}_{\mathcal{I}}$ of the same dimensionality as $\mathbf{x}_{\mathcal{I}}$ such that $\mathbf{z}_i$ corresponds to $\mathbf{x}_i$, $i \in \mathcal{I}$, and an additional vector of latents $\mathbf{c} \in \mathbb{R}^{d_1}$ of a constant size $d_1$. In general, $\mathbf{x}_{\mathcal{I}}$ can be generated given $\mathbf{z}_{\mathcal{I}}$ and each $\mathbf{z}_i$ is generated given $\mathbf{c}$, i.e., $p(\mathbf{x}_{\mathcal{I}}|\mathcal{I}, \mathbf{z}_{\mathcal{I}})\, p(\mathbf{z}_{\mathcal{I}}|\mathbf{c})\, p(\mathbf{c})$, where $p(\mathbf{x}_{\mathcal{I}}|\mathcal{I}, \mathbf{z}_{\mathcal{I}}) = \prod_{i=1}^{|\mathcal{I}|} p(\mathbf{x}_i|\mathbf{z}_i)$ and $p(\mathbf{z}_{\mathcal{I}}|\mathbf{c}) = p(|\mathcal{I}|) \prod_{i=1}^{|\mathcal{I}|} p(\mathbf{z}_i|\mathbf{c})$. Then, the variational posteriors can take the following form: $q(\mathbf{z}_{\mathcal{I}}|\mathbf{x}_{\mathcal{I}}) = \delta(|\mathcal{I}|) \prod_{i=1}^{|\mathcal{I}|} q(\mathbf{z}_i|\mathbf{x}_i)$, where $\delta(\cdot)$ is Dirac's delta, and additionally we have $q(\mathbf{c}|\mathbf{x}_{\mathcal{I}})$. In (Kim et al., 2021), a few simplifications were made such that the model fits well modeling point clouds (sets of 3-D points), namely, for all $i = 1, \ldots, |\mathcal{I}|$, $p(\mathbf{z}_i|\mathbf{c}) = p(\mathbf{z}_i)$, and $q(\mathbf{z}_i|\mathbf{c}) = p(\mathbf{z}_i)$. Further, the authors of (Kim et al., 2021) suggested to define a hierarchical VAE with multiple layers of $\mathbf{z}_{\mathcal{I}}$'s and $\mathbf{c}$'s since a single layer did not result in good performance, and they replaced the conditional likelihood with Chamfer Distance as a well-suited distance for point clouds. In this paper, we find a great appeal of the VAE framework and its flexibility; however, we claim using two distinct latents and a hierarchical latent structure to be unnecessary. Instead, we suggest picking a careful parameterization to be *crucial* in obtaining high performance.

**Permutation-equivariant/invariant Parameterizations**   Deep Neural Networks are widely used as transformations of raw data and parameterizations of probability distributions. It is advocated (but also observed empirically) that modeling probability distributions requires utilizing symmetries in data (Bronstein et al., 2021). For instance, for objects whose dimensions can be shuffled without changing the underlying latent structure, we need either *permutation-invariant* or *permutation-equivariant* transformations. For a given permutation matrix $\mathbf{P}$, a function $f : \mathbb{X} \to \mathbb{Y}$ is *permutation-invariant* if $f(\mathbf{Px}) = \mathbf{y}$; on the other hand, a function $f : \mathbb{X} \to \mathbb{Y}$ is *permutation-equivariant* if $f(\mathbf{Px}) = \mathbf{Py}$.

A general *blueprint* for composing geometric deep neural networks is a composition of permutation-invariant layers and/pr permutation-equivariant layers, with nonlinearity activation functions in between (Bronstein et al., 2021). An example of such a blueprint is an architecture called DeepSets (Zaheer et al., 2017). It formulates a general permutation-equivariance layer treating all variables consistently regardless their positions. Then it applies a symmetric aggregation like averaging or other pooling operators (Kimura et al., 2024; Ilse et al., 2018; Xie & Tong, 2025) to combine these equivariant features in a permutation-invariant fashion, ensuring the order of inputs does not affect the output. The drawback of this approach is that all elements are processed separately before being aggregated with a non-learnable pooling operator. This manner of constructing permutation-invariant transformations might be highly limiting.

A potential solution to that issue is replacing static pooling with a learned attention mechanism, allowing utilizing transformer-based architectures to transform all elements jointly in an equivariant and invariant fashion. An example of a fully-transformer-based model is SetTransformer (Lee et al., 2019) that builds on the following idea. Attention mechanism is defined as $\mathrm{Att}(\mathbf{Q}, \mathbf{K}, \mathbf{V}) \stackrel{df}{=} \mathrm{softmax}(\mathbf{Q}\mathbf{K}^{\top})\mathbf{V}$, where $\mathbf{Q} \in \mathbb{R}^{m \times D}$, $\mathbf{K} \in \mathbb{R}^{M \times D}$, $\mathbf{V} \in \mathbb{R}^{M \times D}$, and $\mathrm{softmax}(\cdot)$ is the

row-wise softmax function.[5] Typically, $\mathbf{Q}$, $\mathbf{K}$ and $\mathbf{V}$ are results of linear transformations of some inputs. If all matrices are calculated based on the same input $\mathbf{X}$, we refer to it as *self-attention*. However, if $\mathbf{Q}$ is calculated using $\mathbf{X}$, and another input $\mathbf{Y}$ is used to obtain $\mathbf{K}$ and $\mathbf{V}$, then we refer to it as *cross-attention*. Let $\mathrm{Att}_K$ denote a multi-head attention with $K$ heads, i.e., a concatenation of $K$ attention layers. SetTransformer introduces a multi-head attention block (MAB) (Lee et al., 2019; Zhang et al., 2022):

$$\mathrm{MAB}(\mathbf{X}, \mathbf{Y}) \stackrel{df}{=} f(\mathbf{X}, \mathbf{Y}) + \mathrm{relu}(f(\mathbf{X}, \mathbf{Y})\mathbf{W}), \tag{11}$$

$$f(\mathbf{X}, \mathbf{Y}) \stackrel{df}{=} \mathbf{X}\mathbf{W}_f + \mathrm{Att}_K(\mathbf{Q}(\mathbf{X}), \mathbf{K}(\mathbf{Y}), \mathbf{V}(\mathbf{Y})),$$

where $f(\mathbf{X}, \mathbf{Y}) \stackrel{df}{=} \mathbf{X}\mathbf{W}_f + \mathrm{Att}_K(\mathbf{Q}(\mathbf{X}), \mathbf{K}(\mathbf{Y}), \mathbf{V}(\mathbf{Y}))$, $\mathbf{W}$ and $\mathbf{W}_f$ are learnable weight matrices. Given MAB, SetTransformer further defines the following two blocks, namely, the Set Attention Block (SAB) and Induced Set Attention Block (ISAB): $\mathrm{SAB}(\mathbf{X}) \stackrel{df}{=} \mathrm{MAB}(\mathbf{X}, \mathbf{X})$, and $\mathrm{ISAB}(\mathbf{X}) \stackrel{df}{=} \mathrm{MAB}(\mathbf{X}, \mathrm{MAB}(\mathbf{U}, \mathbf{X}))$, where $\mathbf{U} \in \mathbb{R}^{m \times D}$ are *inducing points*, i.e., a global weight matrix learnable by backpropagation. ISAB allows to change the size of the input, and similarly to SAB, it is permutation-equivariant (Lee et al., 2019). To obtain a permutation-invariant transformation, SetTransformer proposes to use another layer called Pooling by Multi-head Attention (PMA):

$$\mathrm{PMA}(\mathbf{X}) = \mathrm{MAB}(\mathbf{S}, \mathrm{rFF}(\mathbf{X})), \tag{12}$$

where $\mathbf{S} \in \mathbb{R}^{m \times D}$ is a matrix of learnable *inducing points* (or *pseudoinputs*), and $\mathrm{rFF} : \mathbb{R}^{M \times D} \to \mathbb{R}^{M \times D}$ is a row-wise linear layer. For fixed inducing points $\mathbf{S}$ in PMA, this layer is permutation-invariant (this is due to applying the attention layer, see Property 3 in Appendix C). These building blocks can be used to formulate a deep neural network for parameterizing a probabilistic model. However, we advocate for a different parameterization that applies a single multi-head attention layer in a transformer block to obtain a fixed-size output, and then a series of transformer blocks.

**Latent Diffusion Models** Latent Diffusion Models (LDMs) perform diffusion processes in learned latent spaces rather than directly in high-dimensional data spaces. Stable Diffusion (Rombach et al., 2022) pioneered this approach for text-to-image synthesis by training diffusion models in the latent space of a pre-trained VAE, dramatically reducing computational costs while maintaining generation quality. This paradigm has proven effective across diverse scientific domains: all-atom diffusion transformers (Joshi et al., 2025) generate molecules and materials with atomic-level precision, similary LaM-SLidE (Sestak et al., 2025) utilizes transformer-based LMD for molecular dynamics (among others), while La-proteina (Geffner et al., 2025) employs transformer-based partially latent flow matching for atomistic protein generation. These advances demonstrate the versatility of latent diffusion approaches for complex, high-dimensional scientific data across multiple modalities. Here, we extend this framework to single-cell transcriptomics by proposing a transformer-based LDM for this biological data type.

**Generative Models for scRNA-seq** In the context of single-cell genomics, numerous generative models have been developed for (conditional) sampling of gene expression profiles. scVI (Lopez et al., 2018) represents an early VAE-based generative model, while more recent approaches include GAN-based and diffusion-based architectures such as scGAN (Marouf et al., 2020a) and scDiffusion (Luo et al., 2024). These models operate in continuous space and therefore transform discrete gene expression data into log-normalized counts. Recently, latent diffusion frameworks have emerged, including SCLD (Wang et al., 2023), scVAEDer (Sadria & Layton, 2025), and CFGen (Palma et al., 2025a), which leverage them. Additionally, application-specific generative models have been developed for perturbational single-cell genomics, including CPA (Lotfollahi et al., 2023a), SquiDiff (He et al., 2024), CellFlow (Klein et al., 2025), and CellOT (Bunne et al., 2023), which are tailored to capture the effects of genetic and chemical perturbations on cellular states. Our approach is similar in spirit to CFGen, SCLD, and scVAEDer, but leverages transformer-based architectures for both our newly proposed VAE and the latent diffusion model.

# E. Our Approach: Additional information

## E.1. Conditional likelihood: The parameterization of Negative Binomial

We model the gene expression counts using a Negative Binomial distribution, which effectively captures the overdispersion commonly observed in single-cell RNA-seq data. The conditional likelihood for our model is specified as follows.

---

[5]We skip scaling $\mathbf{Q}\mathbf{K}^\top$ by $1/\sqrt{D}$ to avoid unnecessary clutter.

Let $\mathbf{h}(\mathbf{Z}) \in \mathbb{R}^D$ denote the output of our neural network for a given cell embeddibg $\mathbf{Z}$, where $D$ is the number of genes. We apply a softmax transformation to obtain normalized ratios:

$$p_i(\mathbf{Z}) = \frac{\exp(\mathbf{h}_i(\mathbf{Z}))}{\sum_{j=1}^{D} \exp(\mathbf{h}_j(\mathbf{Z}))} \tag{13}$$

where $i = 1, 2, \ldots, D$, and $\sum_{i=1}^{D} p_i = 1$.

To obtain the expected expression counts, we scale these probabilities by the cell-specific library size $L$, namely:

$$\eta_i(\mathbf{Z}) = L \cdot p_i(\mathbf{Z}) \tag{14}$$

where $\mu_i$ represents the mean parameter for gene $i$ in the Negative Binomial distribution.

The gene expression count $x_i$ for gene $i$ is then modeled as:

$$x_i \sim \mathrm{NB}(\eta_i(\mathbf{Z}), \alpha_i) \tag{15}$$

where NB denotes the Negative Binomial distribution parameterized by mean $\mu_i$ and dispersion $\alpha_i$. The probability mass function is given by:

$$p(x_i | \eta_i(\mathbf{Z}), \alpha_i) = \frac{\Gamma(x_i + \alpha_i^{-1})}{\Gamma(x_i + 1)\Gamma(\alpha_i^{-1})} \left( \frac{\alpha_i^{-1}}{\alpha_i^{-1} + \eta_i(\mathbf{Z})} \right)^{\alpha_i^{-1}} \left( \frac{\eta_i(\mathbf{Z})}{\alpha_i^{-1} + \eta_i(\mathbf{Z})} \right)^{x_i} \tag{16}$$

We consider two parameterizations for the dispersion:

1. **Shared dispersion**: A single parameter $\alpha$ is used for all genes, i.e., $\alpha_i = \alpha$ for all $i \in \{1, 2, \ldots, D\}$. This reduces the number of parameters and assumes homogeneous overdispersion across genes.

2. **Gene-specific dispersion**: Each gene has its own dispersion parameter, resulting in a vector $\boldsymbol{\alpha} = (\alpha_1, \alpha_2, \ldots, \alpha_D)$. This allows for heterogeneous overdispersion patterns across genes, providing greater flexibility at the cost of additional parameters.

This formulation ensures that the predicted expression values respect the constraint that total counts sum to the observed library size, while the Negative Binomial distribution appropriately models the count nature and overdispersion of the data. The softmax transformation guarantees that the neural network learns a proper distribution over genes, making the model interpretable as learning the relative expression probabilities for each cell.

## F. Datasets

**General**   In our experiments, we used the following datasets: Dentate gyrus, Tabula Muris, Human Lung Census Atlas (HLCA), Parse1M and Replogle-Nadig; see Table 6 for details.

**Experiment 1: Cell generation (benchmarks)**   In the cell generation experiment, we used three widely used datasets, namely, Dentate gyrus, Tabula Muris, and HLCA. Dentate gyrus is the smallest dataset (only 18k cell and 17k genes). Tabula Muris is a small dataset with over 245k cells and almost 20k genes. Human Lung Cell Atlas (HLCA) is the largest, having about 585k cells and almost 28k genes.

**Experiment 2: Parse1M & Replogle**   In the second experiment, we used a curated subset of 10 Million Human PBMC dataset. We carried out experiments on 2k highly variable genes (HVGs). In this data, we focused on a single donor who had 18 cell types undergone 90 cytokine perturbations as well as a control treatment. We left out for testing the cell type 'CD4 Naive' and 27 cytokine perturbations.

Next, we used the well-known Replogle-Nadig dataset which consists of four cell lines and 2024 gene-edits. We carried out experiments on 2k highly variable genes (HVGs). All 2024 gene-edits in three cell-lines ('jurkat', 'k562', 'rpe1'), along with a subset of edits from the 'hepg2' cell line were used for taining. The remaining 'hepg2' gene-edits were held out for testing. We held out 372 gene edits from hepg2 cells.

**Experiment 3: COVID-19 and Tabula Sapiens 2.0** In the fourth experiment, we used two datasets for embedding evaluation. First, we used the scRNA-seq experimental dataset of four healthy donors' lung sections infected with SARS-CoV-2 (Wu et al., 2024). Data were downloaded from CZ CELLxGENE[6]. Second, we used the Tabula Sapiens 2.0 dataset (Tabula Sapiens Consortium & Quake, 2025), a comprehensive single-cell atlas of human tissues. We focused on 6 tissues: blood, spleen, lymph node, small intestine, thymus, and liver. We filtered out cell types with fewer than 100 cells to ensure robust classification performance and used the resulting filtered dataset for multinomial logistic regression-based cell type prediction tasks.

*Table 6.* Summary of datasets used in the experiments.

| Experiment | Dataset name | No. of cells | No. of genes | No. of cell types/lines |
|:---|:---|:---:|:---:|:---:|
| 1 | Dentate gyrus | 18,213 | 17,002 | 14 cell types |
| 1 | Tabula Muris | 245,389 | 19,734 | 123 cell types |
| 1 | HLCA | 584,944 | 27,997 | 50 cell types |
| 2 | Parse1M | 1,267,690 | 2,000 (HVGs) | 18 cell types |
| 2 | Replogle-Nadig | 624,158 | 2,000 (HVGs) | 4 cell lines |
| 3 | COVID-19 | 354,026 | 27,998 | 55 cell types |
| 3 | Tabula Sapiens 2.0 | 1,482,026 | $\sim 25$k | 22 cell types |

## G. Baselines

**scVI** Single-cell Variational Inference (scVI) (Lopez et al., 2018) is VAE-based generative models designed for single-cell discrete data. Following the standard VAE framework, this model learns a Gaussian latent space that is subsequently decoded into the parameters of a discrete conditional likelihood model. For the reconstruction. For the observational data experiments, we implemented our own scVI model following default parameters from (Palma et al., 2025a). For the perturbational dataset, we used the implementation from the State (Adduri et al., 2025) reproducibility repo `https://github.com/ArcInstitute/State-reproduce`, using default parameters. We train the models for 120k steps.

**scDiffusion** A version of a latent diffusion model for single-cell gene expression data is scDiffusion (Luo et al., 2024). The scDiffusion model consists of three modules. The first module is an auto-encoderthat transforms gene expression patterns into a compact representation space, allowing dimensionality reduction and identification of complex cellular measurements. In the latent space, a denoising network is trained to reverse a diffusion process applied to the latent embeddings, turning noise into meaningful biological signal encoded in the latent space. To ensure guided generation, a third model is trained, a classifier, for incorporating cell type or other biological attributes.

**CFGen** CFGen is a current state-of-the-art latent diffusion model that builds upon scVI, training a latent flow matching model in the VAE's latent space (Palma et al., 2025a). Similar to our approach, CFGen employs a two-stage training strategy: first training the autoencoder, then training the flow matching model on the VAE-generated embeddings. While CFGen introduces additive steering through classifier-free guidance. However, since in the observational experiments we only conditioned on a single attribute, we do not think this is the source of the performance difference. Additionally, CFGen models the library size within the diffusion framework and samples from the mean and standard deviation of the library size distribution for conditional generation. We adapted this approach for sampling library size in our Negative Binomial conditional likelihood; however, unlike CFGen, we do not condition our Diffusion Transformer model on library size. We did not retrain CFGen but used the checkpoints for each observational dataset from the original repository. We followed the notebook for sampling from the model using guidance value of 1 (default).

**CPA** Compositional Perturbation Autoencoder (CPA) (Lotfollahi et al., 2023b) is a deep generative model developed to predict gene expression changes under perturbations and their combinations. CPA disentangles latent representations of basal cellular state, perturbation effects, and additional covariates such as cell type. By recombining these factors through its decoder, CPA can reconstruct observed expression profiles and generalize to unseen perturbation–covariate combinations. This compositional structure enables CPA to extrapolate beyond training data, making it particularly well-suited for evaluating out-of-distribution generalization in perturbational single-cell datasets. For each dataset (Replogle

---

[6]`https://cellxgene.cziscience.com/collections/2a9a17c9-1f61-4877-b384-b8cd5ffa4085`

and Parse1M) we trained 3 models with different seeds, but the same leave-out test set. We used the implementation from the State (Adduri et al., 2025) reproducibility repo https://github.com/ArcInstitute/State-reproduce, using default parameters. We train the models for 120k steps.

**scGPT** For each dataset (Replogle and Parse1M) we trained 3 models with different seeds, but the same leave-out test set. We used as baseline the scGPT (Cui et al., 2024) model in the 'genetic' configuration for the Replogle-Nadig dataset, and in the 'chemical' configuration for the Parse 1M dataset. We used the implementation from the State (Adduri et al., 2025) reproducibility repo https://github.com/ArcInstitute/State-reproduce, using default parameters. We train both scGPT models for 80k steps. For the 'genetic' configuration, we noticed that not all genes were present in the vocabulary (scGPT human downloaded from the original repo, following the baseline reproducibility repo from STATE), and that some genetic perturbations would be dropped. Hence, we built a feature set that consists of the union of 2000 HVG as well as the 2024 genetic perturbations, recovering 3482 genes. The 'scGPT genetic' model was therefore trained on 3482 genes, but evaluated on 1962 genes, which correspond to the subset of HVG that are part of the scGPT vocabulary. For the 'chemical' configuration, we trained on the same feature set of the rest of the baselines 2000 HVG.

**STATE-Tx** State is a first-generation virtual cell model developed to predict how single-cell gene expression profiles change in response to chemical, genetic, or cytokine perturbations (Adduri et al., 2025). The model is trained on one of the largest compendia of single-cell data to date, with observational data from nearly 170 million cells and perturbational data from over 100 million cells across diverse cell types. State comprises two interconnected modules: the State Embedding (SE) model and the State Transition (ST) model. SE embeds raw transcriptomes into a smooth, multidimensional latent space that mitigates technical noise and groups similar cell types, facilitating robust representation learning. Conditioned on these embeddings, ST employs a bidirectional transformer architecture with self-attention over sets of cells to model how perturbations induce transitions in expression space, capturing both biological and technical heterogeneity without explicit distributional assumptions. Once trained, the combined model predicts post-perturbation expression shifts from an input transcriptome and perturbation specification, achieving improved accuracy over prior computational baselines in distinguishing true perturbation effects and identifying differentially expressed genes. For each dataset (Replogle and Parse1M) we trained 3 models with different seeds, but the same leave-out test set. We used the STATE-Tx implementation from https://github.com/ArcInstitute/State-reproduce and trained for 80k steps for both datasets. We used the model configurations used in this notebook https://colab.research.google.com/drive/1Ih-KtTEsPqDQnjTh6etVv_f-gRAA86ZN for both datasets.

**CellFlow** CellFlow (Klein et al., 2025) is a conditional generative framework based on flow matching that models single-cell phenotypes under diverse perturbations. The method decomposes an experimental condition into perturbations, their covariates (such as dosage or timing), and sample-level covariates (such as donor or cell line identity). Each factor is embedded—using, for example, molecular fingerprints for small molecules or protein language model embeddings for gene perturbations—and aggregated into a single condition representation via permutation-invariant modules such as multi-head attention or DeepSets. Guided by this representation, a conditional flow-matching network learns a vector field that transports control cells to their perturbed counterparts in a low-dimensional latent space, using optimal transport couplings to align source and target populations and thereby improve training stability and biological fidelity. For each dataset (Replogle and Parse1M) we trained 3 models with different seeds, but the same leave-out test set. We used the CellFlow implementation with gene and condition embedding computed from the mean expression of the training data. We used PCA for encoding and reconstruction. We trained for 500k steps for both datasets. We used the model configurations used in this notebook https://cellflow.readthedocs.io/en/latest/notebooks/100_pbmc.html for both datasets.

**Perturb mean** The Perturbation Mean baseline provides predictions by combining cell-type-specific control means with learned perturbation effects. The model operates as follows:

For each cell type $c$ and perturbation $p$, we calculate separate means for control and perturbed populations:

$$\boldsymbol{\mu}_c^{\text{ctrl}} = \frac{1}{|\mathcal{C}_c|} \sum_{i \in \mathcal{C}_c} \mathbf{x}^{(i)}, \quad \boldsymbol{\mu}_{c,p}^{\text{pert}} = \frac{1}{|\mathcal{P}_{c,p}|} \sum_{i \in \mathcal{P}_{c,p}} \mathbf{x}^{(i)} \tag{17}$$

Here, $\mathcal{C}_c$ denotes the collection of control (ctrl) cells belonging to type $c$, while $\mathcal{P}_{c,p}$ represents perturbed cells of type $c$ that

received perturbation $p$. The cell-type-specific perturbation effect is computed as:

$$\boldsymbol{\delta}_{c,p} = \boldsymbol{\mu}_{c,p}^{\text{pert}} - \boldsymbol{\mu}_c^{\text{ctrl}} \tag{18}$$

To obtain a global perturbation effect, we average these cell-type-specific effects across all cell types where the perturbation was applied:

$$\boldsymbol{\delta}_p = \frac{1}{|\mathcal{C}_p|} \sum_{c \in \mathcal{C}_p} \boldsymbol{\delta}_{c,p}, \quad \text{where } \mathcal{C}_p = \{c \mid |\mathcal{P}_{c,p}| > 0\} \tag{19}$$

For prediction, given a test cell of type $t$ and perturbation $p$, the model generates:

$$\hat{\mathbf{x}} = \boldsymbol{\mu}_t^{\text{ctrl}} + \boldsymbol{\delta}_p, \quad \boldsymbol{\delta}_{\text{ctrl}} = \mathbf{0} \tag{20}$$

This approach ensures that control cells are predicted without modification, while perturbed cells receive a consistent global perturbation shift regardless of their specific cell type.

**Hepg2 and CD4 mean, context mean** The Context Mean baseline predicts a cell's post-perturbation profile by returning the average perturbed expression of cells of the same cell type observed in the training set. For every cell type $c$, we collect all training cells whose perturbation is not the control and form the pseudo-bulk mean:

$$\boldsymbol{\mu}_c = \frac{1}{|\mathcal{T}_c|} \sum_{i \in \mathcal{T}_c} \mathbf{x}^{(i)}, \quad \mathcal{T}_c = \left\{ i \mid \text{cell\_type}(i) = c, p^{(i)} \neq \text{ctrl} \right\}. \tag{21}$$

At inference time, for a test cell $i$ with cell type $c^{(i)}$ and perturbation label $p^{(i)}$, we predict:

$$\hat{\mathbf{x}}^{(i)} = \begin{cases} \mathbf{x}^{(i)} & \text{if } p^{(i)} = \text{ctrl}, \\ \boldsymbol{\mu}_{c^{(i)}} & \text{if } p^{(i)} \neq \text{ctrl}. \end{cases} \tag{22}$$

In other words, control cells are passed through unchanged, whereas perturbed cells inherit their cell-type mean.

## H. Implementation details

In this paper, we carried out model selection for various values of hyperparameters. In the following paragraphs, we provide further details for reproducibility.

**VAE** Table 7 summarizes the hyperparameter configurations used for the VAE encoder architectures in our experiments.

Table 8 summarizes the hyperparameter configurations used for the VAE decoder architectures in our experiments. Note that we use either the NegativeBinomial stochastic layer (SCLDM (NB) in our experiments) or the Gaussian stochastic layer (SCLDM (GAUSS) in our experiments)

**Flow Matching** Table 9 summarizes the hyperparameter configurations used for the LDM architectures in our experiments.

**scLDM-VAE Census** Table 10 summarizes the hyperparameter configurations used for the scLDM-VAE Census architectures in our experiments.

**Training details** For training, we swept over various configurations of hyperparameters, see Table 11.

*Table 7.* Hyperparameter values of VAE Encoders considered in this paper.

| VAE Encoder | |
| --- | --- |
| **Embedding layer** | |
| Embedding size | 256 |
| **Cross-Attention Block** | |
| Number of Heads | {1,4,8} |
| No. pseudoinputs | {64,128, 256} |
| Embedding size | {128,256} |
| **Transformer Blocks** | |
| Number of Blocks | {2, 4} |
| Number of Heads | 1 |
| Embedding size | 256 |
| **Gaussian Stochastic Layer** | |
| Latents per token | {8, 16, 32} |

*Table 8.* Hyperparameter values of VAE Decoders considered in this paper.

| VAE Decoder | |
| --- | --- |
| **Transformer Blocks** | |
| Number of Blocks | {2, 4} |
| Number of Heads | {1,4,8} |
| Embedding size | {128,256} |
| Normalization | `LayerNorm` |
| **Cross-Attention Block** | |
| Shared embedding layer | `True` |
| No. pseudoinputs | {64,128, 256} |
| Number of Heads | 1 |
| Embedding size | 256 |
| **NegativeBinomial Stochastic Layer** | |
| Shared $\theta$ | `False` |
| **Gaussian Stochastic Layer** | |
| Output parameters | `means` |

*Table 9.* Hyperparameter values of LDMs considered in this paper.

| LDM – Flow Matching | |
|---|---|
| **Denoising Transformer** | |
| Number of Blocks | 8 |
| Number of Heads | 8 |
| Embedding size | 256 |
| Normalization | `LayerNorm` |
| Adaptive Normalization | `True` |
| **Hyperparams** | |
| $\sigma$ | $1e^{-4}$ |
| $v$ | 0 |
| Transport | `linear` |

*Table 10.* Hyperparameter values of scLDM-VAE Census.

| Encoder-Decoder hyperparameters | |
|---|---|
| Number of Blocks | $\{8,16\}$ |
| Number of Heads | $\{8,16\}$ |
| Number of Layers | $\{8,16\}$ |
| Number of Latent Tokens | $\{256\}$ |
| Embedding size | $\{256,512,768\}$ |
| Normalization | `LayerNorm` |

*Table 11.* Hyperparameter values of training procedures considered in this paper.

| Training | |
|---|---|
| KL-weight | $\{0, 1e^{-5}\}$ |
| Optimizer | `AdamW` |
| Mini-batch size | $\{64,128,256\}$ |
| Learning rate | $\{1e^{-3}, 5e^{-4}\}$ |
| $(\beta_1, \beta_2)$ | $(0.9, 0.95)$ |
| Weight Decay | $\{0, 1e^{-7}, 1e^{-4}\}$ |
| Learning scheduler | `cosine` |

**Model Complexity** In Tables 12 and 13, we report the computational complexity and parameter counts for the models used in our experiments. Table 12 shows the FLOPs and parameter counts for both the VAE and LDM components across five different single-cell datasets: dentate gyrus, tabula muris, HLCA, Replogle, and Parse1M. Table 13 presents the FLOPs and parameter counts for three VAE models trained on the Census datasets, with model sizes ranging from 20M to 270M parameters. In Table 14, we additionally report wall-clock training time and peak GPU memory across all five datasets, measured on NVIDIA A100-80GB GPUs with Distributed Data Parallel across 8 GPUs.

*Table 12.* Model Complexity Comparison Across Datasets

| Dataset | VAE FLOPs | LDM FLOPs | VAE Params | LDM Params |
|---|---|---|---|---|
| dentate_gyrus | 5.02 GFLOPs | 6.45 GFLOPs | 3.45M | 9.77M |
| tabula_muris | 29.45 GFLOPs | 77.33 GFLOPs | 13.23M | 57.81M |
| hlca | 38.13 GFLOPs | 77.33 GFLOPs | 15.35M | 57.83M |
| replogle | 28.02 GFLOPs | 51.56 GFLOPs | 21.16M | 39.91M |
| parse1m | 28.02 GFLOPs | 51.56 GFLOPs | 21.16M | 38.93M |

*Table 13.* Model Complexity Comparison

| Model | VAE TFLOPs | VAE Params |
|---|---|---|
| VAE_Census_20M | 0.33 TFLOPs | 23.75M |
| VAE_Census_70M | 1.02 TFLOPs | 76.07M |
| VAE_Census_270M | 2.63 TFLOPs | 270.14M |

*Table 14.* Training wall-clock time and peak GPU memory for scLDM across datasets. Times on 8×A100 (Distributed Data Parallel) are reported as mean ± standard error across runs; 1×A100 times are linearly projected. Peak memory is per GPU.

| Dataset | Likelihood | Time 8×A100 (h) | Time 1×A100 (h) | Peak Mem/GPU (GB) |
|---|---|---|---|---|
| Dentate Gyrus | Gauss | $0.30 \pm 0.00$ | $2.17 \pm 0.00$ | 32.50 |
| | NB | $0.30 \pm 0.00$ | $2.17 \pm 0.00$ | 32.50 |
| Tabula Muris | Gauss | $14.61 \pm 0.19$ | 106.0 | 34.10 |
| | NB | $14.50 \pm 0.28$ | 105.2 | 33.18 |
| HLCA | Gauss | $17.01 \pm 0.11$ | 123.4 | 49.59 |
| | NB | $16.97 \pm 0.15$ | 123.1 | 44.72 |
| Replogle | Gauss | $2.91 \pm 0.02$ | 21.1 | 5.92 |
| | NB | $3.19 \pm 0.04$ | 23.1 | 5.83 |
| Parse 1M | Gauss | $5.40 \pm 0.01$ | 39.2 | 5.92 |
| | NB | $5.94 \pm 0.04$ | 43.1 | 5.83 |

# I. Evaluation

## I.1. Maximum Mean Discrepancy (MMD)

We propose to use the Maximum Mean Discrepancy (MMD) (Gretton et al., 2012). MMD is a non-parametric distance measure between probability distributions based on the notion of embedding distributions into a reproducing kernel Hilbert space (RKHS) $\mathcal{H}$. Given two distributions $P$ and $Q$ over a domain $\mathcal{X}$, the MMD is defined as:

$$\text{MMD}[\mathcal{F}, P, Q] = \sup_{f \in \mathcal{F}} \left( \mathbb{E}_{x \sim P}[f(x)] - \mathbb{E}_{y \sim Q}[f(y)] \right), \tag{23}$$

where $\mathcal{F}$ is a class of functions. When $\mathcal{F}$ is the unit ball in an RKHS $\mathcal{H}$ with kernel $k$, the MMD can be expressed as:

$$\text{MMD}^2[\mathcal{H}, P, Q] = \mathbb{E}_{x,x' \sim P}[k(x, x')] + \mathbb{E}_{y,y' \sim Q}[k(y, y')] - 2\mathbb{E}_{x \sim P, y \sim Q}[k(x, y)]. \tag{24}$$

In practice, given finite samples $X = \{x_1, \ldots, x_m\}$ drawn from $P$ and $Y = \{y_1, \ldots, y_n\}$ drawn from $Q$, we use the unbiased empirical estimate:

$$\widehat{\mathrm{MMD}}^2[X, Y] = \frac{1}{m(m-1)} \sum_{i \neq j}^{m} k(x_i, x_j) + \frac{1}{n(n-1)} \sum_{i \neq j}^{n} k(y_i, y_j) - \frac{2}{mn} \sum_{i=1}^{m} \sum_{j=1}^{n} k(x_i, y_j). \tag{25}$$

The choice of kernel $k$ determines the richness of the function class $\mathcal{F}$. Common choices include the Gaussian RBF kernel $k(x, y) = \exp(-\|x - y\|^2 / 2\sigma^2)$ with bandwidth parameter $\sigma$. The MMD is zero if and only if $P = Q$ when using a characteristic kernel, making it a powerful tool for two-sample testing and distribution matching applications.

### I.2. 2-Wasserstein Distance (W2)

The 2-Wasserstein distance provides an alternative metric for comparing probability distributions based on optimal transport theory. For distributions $P$ and $Q$ on $\mathbb{R}^d$, the 2-Wasserstein distance is defined as:

$$W_2(P, Q) = \left( \inf_{\gamma \in \Gamma(P,Q)} \int_{\mathbb{R}^d \times \mathbb{R}^d} \|\mathbf{x} - \mathbf{y}\|^2 \, d\gamma(\mathbf{x}, \mathbf{y}) \right)^{1/2}, \tag{26}$$

where $\Gamma(P, Q)$ denotes the set of all joint distributions with marginals $P$ and $Q$. For empirical distributions with equal sample sizes $n$, given samples $X = \{\mathbf{x}_1, \ldots, \mathbf{x}_n\}$ and $Y = \{\mathbf{y}_1, \ldots, \mathbf{y}_n\}$, the discrete 2-Wasserstein distance simplifies to:

$$W_2^2(X, Y) = \frac{1}{n} \min_{\pi \in \Pi_n} \sum_{i=1}^{n} \|\mathbf{x}_i - \mathbf{y}_{\pi(i)}\|^2, \tag{27}$$

where $\Pi_n$ is the set of all permutations of $\{1, \ldots, n\}$. This optimization problem can be solved efficiently using the Hungarian algorithm or entropic regularization approaches.

When both distributions are Gaussian with means $\boldsymbol{\mu}_P, \boldsymbol{\mu}_Q$ and covariances $\boldsymbol{\Sigma}_P, \boldsymbol{\Sigma}_Q$, the 2-Wasserstein distance has a closed-form expression:

$$W_2^2(P, Q) = \|\boldsymbol{\mu}_P - \boldsymbol{\mu}_Q\|^2 + \mathrm{tr}\left( \boldsymbol{\Sigma}_P + \boldsymbol{\Sigma}_Q - 2(\boldsymbol{\Sigma}_P^{1/2} \boldsymbol{\Sigma}_Q \boldsymbol{\Sigma}_P^{1/2})^{1/2} \right), \tag{28}$$

which coincides with the Frechét Distance.

Unlike MMD, the Wasserstein distance directly captures the geometry of the underlying space and provides interpretable transport plans between distributions.

### I.3. Fréchet Distance for Gene Expression Profile Evaluation

We adapt the Fréchet Inception Distance (FID) framework to evaluate the quality of synthetic gene expression profiles by replacing the Inception network's feature extraction with Principal Component Analysis (PCA). This approach provides a computationally efficient and interpretable metric for comparing distributions of real and synthetic gene expression data.

Assuming the feature representations follow multivariate Gaussian distributions:

- Real data: $\mathcal{N}(\mu_r, \Sigma_r)$;

- Synthetic data: $\mathcal{N}(\mu_s, \Sigma_s)$.

We estimate the parameters:

$$\mu_r = \frac{1}{n} \sum_{i=1}^{n} \mathbf{z}_i^r, \quad \Sigma_r = \frac{1}{n-1} \sum_{i=1}^{n} (\mathbf{z}_i^r - \mu_r)(\mathbf{z}_i^r - \mu_r)^\top \tag{29}$$

$$\mu_s = \frac{1}{m} \sum_{j=1}^{m} \mathbf{z}_j^s, \quad \Sigma_s = \frac{1}{m-1} \sum_{j=1}^{m} (\mathbf{z}_j^s - \mu_s)(\mathbf{z}_j^s - \mu_s)^\top \tag{30}$$

The Fréchet Distance between these distributions is then computed as:

$$\text{FD} = ||\mu_r - \mu_s||_2^2 + \text{Tr}(\Sigma_r + \Sigma_s - 2(\Sigma_r \Sigma_s)^{1/2}) \tag{31}$$

where $\text{Tr}(\cdot)$ denotes the matrix trace and $(\Sigma_r \Sigma_s)^{1/2}$ is the matrix square root of $\Sigma_r \Sigma_s$.

### I.3.1. INTERPRETATION

This metric captures both the difference in means (first term) and the difference in covariance structure (second term) between real and synthetic gene expression profiles in the reduced PCA space. Lower values indicate better agreement between the distributions, suggesting higher quality synthetic data. The use of PCA ensures that the comparison focuses on the most significant sources of variation in the gene expression data while reducing computational complexity from $O(d^2)$ to $O(k^2)$ where typically $k \ll d$.

### I.4. Leave-One-Out 1-NN Classifier Accuracy

The leave-one-out 1-nearest neighbor (1-NN) classifier accuracy provides a simple yet effective two-sample test for comparing real and generated distributions (Lopez-Paz & Oquab, 2016). Given real samples $\mathcal{X}_r = \{\mathbf{x}_1^r, \ldots, \mathbf{x}_n^r\}$ and generated samples $\mathcal{X}_g = \{\mathbf{x}_1^g, \ldots, \mathbf{x}_m^g\}$, we pool both sets and assign binary labels $y_i = 1$ for real samples and $y_i = 0$ for generated samples.

For each sample $\mathbf{x}_i$ in the pooled set, we find its nearest neighbor $\mathbf{x}_{\text{NN}(i)}$ (excluding itself) and predict its label as $\hat{y}_i = y_{\text{NN}(i)}$. The 1-NN accuracy is then computed as:

$$\text{1-NN} = \frac{1}{n+m} \sum_{i=1}^{n+m} \mathbf{1}[\hat{y}_i = y_i]. \tag{32}$$

When the real and generated distributions are identical, each sample's nearest neighbor is equally likely to come from either distribution, yielding an expected accuracy of 0.5. Conversely, when the distributions are perfectly separable, the accuracy approaches 1.0, as each sample's nearest neighbor will belong to the same distribution. Thus, values closer to 0.5 indicate better distributional matching, while values approaching 1.0 suggest the generative model fails to capture the true data distribution. In practice, we compute this metric in PCA space to reduce computational cost and focus on the most salient variation in the data.

### I.5. $k$-NN Precision and Recall

While aggregate metrics like MMD and Wasserstein distance capture overall distributional similarity, they conflate two distinct aspects of generative model performance: sample quality (fidelity) and sample diversity (coverage). Following (Kynkäänniemi et al., 2019), we separately measure these aspects using non-parametric manifold-based precision and recall metrics.

**Manifold Estimation.** Given feature representations $\Phi_r = \{\phi_1^r, \ldots, \phi_n^r\}$ for real data and $\Phi_g = \{\phi_1^g, \ldots, \phi_m^g\}$ for generated data (obtained via PCA projection), we estimate each manifold by surrounding each point with a hypersphere whose radius equals the distance to its $k$-th nearest neighbor within the same set. For a point $\phi$ and a reference set $\Phi$, we define the binary membership function:

$$f(\phi, \Phi) = \begin{cases} 1 & \text{if } \exists \, \phi' \in \Phi : \|\phi - \phi'\| \leq \|\phi' - \text{NN}_k(\phi', \Phi)\| \\ 0 & \text{otherwise} \end{cases}, \tag{33}$$

where $\text{NN}_k(\phi', \Phi)$ denotes the $k$-th nearest neighbor of $\phi'$ within $\Phi$.

**Precision and Recall.** Precision measures the fraction of generated samples that fall within the estimated real data manifold:

$$\text{Precision} = \frac{1}{m} \sum_{j=1}^{m} f(\phi_j^g, \Phi_r),$$
(34)

while recall measures the fraction of real samples covered by the estimated generated data manifold:

$$\text{Recall} = \frac{1}{n} \sum_{i=1}^{n} f(\phi_i^r, \Phi_g).$$
(35)

Intuitively, high precision indicates that generated samples are realistic (i.e., lie within regions of high real data density), while high recall indicates that the generator covers the full diversity of the real data distribution. A model exhibiting mode collapse would show high precision but low recall, whereas a model generating diffuse, low-quality samples might show high recall but low precision. We use $k = 3$ following Kynkäänniemi et al. (2019), which provides robust estimates without saturating the metrics.

## I.6. Pearson Correlation Coefficient (PCC)

While MMD and Wasserstein distances measure distributional differences, the Pearson correlation coefficient quantifies linear relationships between paired observations. For two random variables $X$ and $Y$, the population Pearson correlation coefficient is defined as:

$$\rho_{X,Y} = \frac{\text{Cov}(X,Y)}{\sigma_X \sigma_Y} = \frac{\mathbb{E}[(X - \mu_X)(Y - \mu_Y)]}{\sqrt{\mathbb{E}[(X - \mu_X)^2]} \sqrt{\mathbb{E}[(Y - \mu_Y)^2]}},$$
(36)

where $\mu_X, \mu_Y$ are the means and $\sigma_X, \sigma_Y$ are the standard deviations. Given paired samples $\{(x_i, y_i)\}_{i=1}^{n}$, the sample correlation coefficient is:

$$r = \frac{\sum_{i=1}^{n} (x_i - \bar{x})(y_i - \bar{y})}{\sqrt{\sum_{i=1}^{n} (x_i - \bar{x})^2} \sqrt{\sum_{i=1}^{n} (y_i - \bar{y})^2}},$$
(37)

where $\bar{x} = \frac{1}{n} \sum_{i=1}^{n} x_i$ and $\bar{y} = \frac{1}{n} \sum_{i=1}^{n} y_i$. The coefficient $r \in [-1, 1]$, with $|r| = 1$ indicating perfect linear relationship and $r = 0$ suggesting no linear correlation. For multivariate data $\mathbf{X} \in \mathbb{R}^{n \times d}$ and $\mathbf{Y} \in \mathbb{R}^{n \times d}$, one can compute the average correlation across dimensions or construct a correlation matrix. While Pearson correlation captures only linear dependencies and is sensitive to outliers, it remains widely used due to its computational efficiency and interpretability in assessing feature-wise relationships between datasets.

## I.7. Coefficient of Determination ($R^2$ Score)

To evaluate the quality of generated single-cell expression profiles, we compute three complementary $R^2$ metrics that capture different statistical properties of the gene expression distributions. Given true counts $\mathbf{X} \in \mathbb{R}^{C \times G}$ and generated counts $\hat{\mathbf{X}} \in \mathbb{R}^{C \times G}$, where $C$ is the number of cells and $G$ is the number of genes, we define:

$R^2$ **Mean ($R_\mu^2$)** measures how well the model captures the average expression level of each gene across cells:

$$\boldsymbol{\mu}_{\text{true}} = \frac{1}{C} \sum_{c=1}^{C} \mathbf{X}_{c,g}, \quad \boldsymbol{\mu}_{\text{gen}} = \frac{1}{C} \sum_{c=1}^{C} \hat{\mathbf{X}}_{c,g}$$
(38)

$$R_\mu^2 = 1 - \frac{\sum_{g=1}^{G} (\boldsymbol{\mu}_{\text{true},g} - \boldsymbol{\mu}_{\text{gen},g})^2}{\sum_{g=1}^{G} (\boldsymbol{\mu}_{\text{true},g} - \bar{\boldsymbol{\mu}}_{\text{true}})^2}$$
(39)

$R^2$ **Variance** $(R^2_{\sigma^2})$    measures how well the model captures the variability of expression for each gene:

$$\boldsymbol{\sigma}^2_{\text{true}} = \frac{1}{C} \sum_{c=1}^{C} (\mathbf{X}_{c,g} - \boldsymbol{\mu}_{\text{true},g})^2, \quad \boldsymbol{\sigma}^2_{\text{gen}} = \frac{1}{C} \sum_{c=1}^{C} (\hat{\mathbf{X}}_{c,g} - \boldsymbol{\mu}_{\text{gen},g})^2 \tag{40}$$

$$R^2_{\sigma^2} = 1 - \frac{\sum_{g=1}^{G} (\boldsymbol{\sigma}^2_{\text{true},g} - \boldsymbol{\sigma}^2_{\text{gen},g})^2}{\sum_{g=1}^{G} (\boldsymbol{\sigma}^2_{\text{true},g} - \bar{\boldsymbol{\sigma}}^2_{\text{true}})^2} \tag{41}$$

$R^2$ **Zeros** $(R^2_{\text{zeros}})$    measures how well the model captures the sparsity pattern (dropout rate) for each gene:

$$\mathbf{z}_{\text{true},g} = \sum_{c=1}^{C} \mathbb{K}[\mathbf{X}_{c,g} = 0], \quad \mathbf{z}_{\text{gen},g} = \sum_{c=1}^{C} \mathbb{K}[\hat{\mathbf{X}}_{c,g} = 0] \tag{42}$$

$$R^2_{\text{zeros}} = 1 - \frac{\sum_{g=1}^{G} (\mathbf{z}_{\text{true},g} - \mathbf{z}_{\text{gen},g})^2}{\sum_{g=1}^{G} (\mathbf{z}_{\text{true},g} - \bar{\mathbf{z}}_{\text{true}})^2} \tag{43}$$

These metrics provide a comprehensive assessment of whether the generative model accurately reproduces the first-order statistics (mean), second-order statistics (variance), and sparsity structure (zero-inflation) of the true gene expression distribution at the gene level. Values closer to 1 indicate better agreement between the generated and true distributions. The value is not bounded by -1 but can be arbitrarily ¡ 0.

### I.8. Metrics used in experiments

**Reconstruction Metrics**    In our experiments, we use the reconstruction error for the Negative Binomial distribution, PCC and Mean Squared Errors (MSE) as reconstruction metrics.

**Generation Metrics**    For evaluating generation capabilities of models, we use the MMD with the RBF kernel, the Wasserstein Distance, the Frechet Distance, the 1-NN classifier, and the Precision-Recall for generative models, all calculated to 30 principal components, unless stated otherwise (e.g. in the case of differentially expressed genes). We compute the PCA on the true data, and project the generated data using the loadings. All evaluations were run using 3-generation seeds.

**Perturbation Metrics**    For evaluating perturbation prediction capabilities of models, we used the overlap of DEG (Differentially Expressed Genes), the Spearman Correlation of Log2Fold Changes of significant DEGs, the discrimination L1 score and the Mean Absolute Error (MAE), all computed using the `cell-eval` (Adduri et al., 2025) library. All evaluations were run using 3 model outputs, generated for 3 different seeds. For scLDM, we simply sampled from the same model 3 times using different seeds. For all other models, we retrained from scratch with different seeds, which only affects initialization since the dataset split is fixed.

## J. Ablation on type of Classifier-Free Guidance

**Classifier-Free Guidance with Multiple Conditioning Variables.**    In our setting, as described in section 3.2, the diffusion model is conditioned on multiple attributes simultaneously (e.g., cell type and perturbation). We explore two alternative strategies for applying classifier-free guidance (CFG):

**(Type I: Joint conditioning).** A single conditioning token is assigned to each unique combination of attributes. The model output under this strategy is given by

$$\tilde{v}_{t,\epsilon}(\mathbf{Z}, y) = v_{t,\epsilon}(\mathbf{Z}; \text{Null}) + \omega \left[ v_{t,\epsilon}(\mathbf{Z}; y) - v_{t,\epsilon}(\mathbf{Z}; \text{Null}) \right], \tag{44}$$

where $y$ encodes the full joint condition (e.g., "CD4 Naive + IL-9" or "HepG2 + PPP6C").

**(Type II: Additive conditioning).** Instead of encoding combinations directly, we treat each conditioning variable independently. For $M$ attributes with labels $\{y^{(j)}\}_{j=1}^{M}$, the guided output is

$$\tilde{v}_{t,\epsilon}\Big(\mathbf{Z}, \{y^{(j)}\}_{j=1}^{M}\Big) = v_{t,\epsilon}(\mathbf{Z}; \text{Null}) + \sum_{j=1}^{M} \omega_j \Big[v_{t,\epsilon}(\mathbf{Z}; y^{(j)}) - v_{t,\epsilon}(\mathbf{Z}; \text{Null})\Big], \tag{45}$$

where each attribute contributes an additive adjustment relative to the unconditional prediction.

**Empirical Comparison.** We evaluate both approaches on **Parse 1M** (conditioning on cell type + cytokine perturbation) and **Replogle** (conditioning on cell type + gene knockout). As shown in Appendix K.4, the joint conditioning strategy (Type I) consistently outperforms the additive conditioning strategy (Type II) across metrics, indicating that learning a single joint embedding for each combination of attributes is more effective in capturing complex context–perturbation interactions than treating them independently.

In all experiments, we set the guidance weight $\omega$ to 1, unless specified otherwise, and we did not tune this parameter.

## K. Additional Results

### K.1. Experiment 1: Ablation experiments and additional results

In Table 15 we report an ablation study on encoding gene expression following our approach of zero padding the non-expressed genes, or utilizing the full context as input, as typically done in scVI. Our approach is superior in terms of reconstruction performance, as well as more computationally efficient.

*Table 15.* Reconstruction performance comparison of our scLDM using the zero padding strategy for encoding, or using all genes as input.

| Dataset | Model | RE $\downarrow$ | PCC $\uparrow$ | MSE $\downarrow$ |
|---|---|---|---|---|
| Dentate Gyrus | full context | 5458.6 | 0.097 | 0.252 |
| | zero padding | **5325.3** | **0.125** | **0.242** |

In Figure 4, we report another ablation study on hyperparameters of the VAE architecture: we sweep over depth (number of layers for the encoder/decoder layers), width (embedding dimensions) and number of latent tokens, over two datasets that consists of vastly different number of genes (18k for dentate gyrus versus 2k for replogle) and cells (600k for replogle and 18k for dentate gyrus). Interestingly, depth appears to negatively impact performance in the dentate gyrus dataset, whereas it does not in the Replogle dataset. For both datasets, a larger embedding dimension as well as a higher number of latent tokens improve performance.

In Figure 5, we compare the performance of our latent aggregation approach against set-based aggregation methods (max, mean, and sum pooling) across two datasets (dentate gyrus in the top row and Replogle in the bottom row).

We implemented the Set-Transformer-based aggregation in the following way: Given an input tensor $\mathbf{X} \in \mathbb{R}^{B \times G \times D}$, where $B$ is the batch size, $G$ is the number of genes, and $D$ is the embedding dimension, the module computes:

$$\mathbf{Z} = \text{MLP}(\text{Agg}(\mathbf{X})^{\top}) \tag{46}$$

where the aggregation function $\text{Agg} : \mathbb{R}^{B \times G \times D} \to \mathbb{R}^{B \times D \times 1}$ is defined as:

$$\text{Agg}(\mathbf{X})_{b,d,1} = \begin{cases} \max_{g=1}^{G} \mathbf{X}_{b,g,d} & \text{if aggregation type = "max"} \\ \frac{1}{G}\sum_{g=1}^{G} \mathbf{X}_{b,g,d} & \text{if aggregation type = "mean"} \\ \sum_{g=1}^{G} \mathbf{X}_{b,g,d} & \text{if aggregation type = "sum"} \end{cases} \tag{47}$$

After transposing the output of $\text{Agg}(\cdot)$, the MLP consists of a linear projection followed by layer normalization:

$$\text{MLP}(\mathbf{Y}) = \text{LayerNorm}(\mathbf{YW} + \mathbf{b}) \tag{48}$$

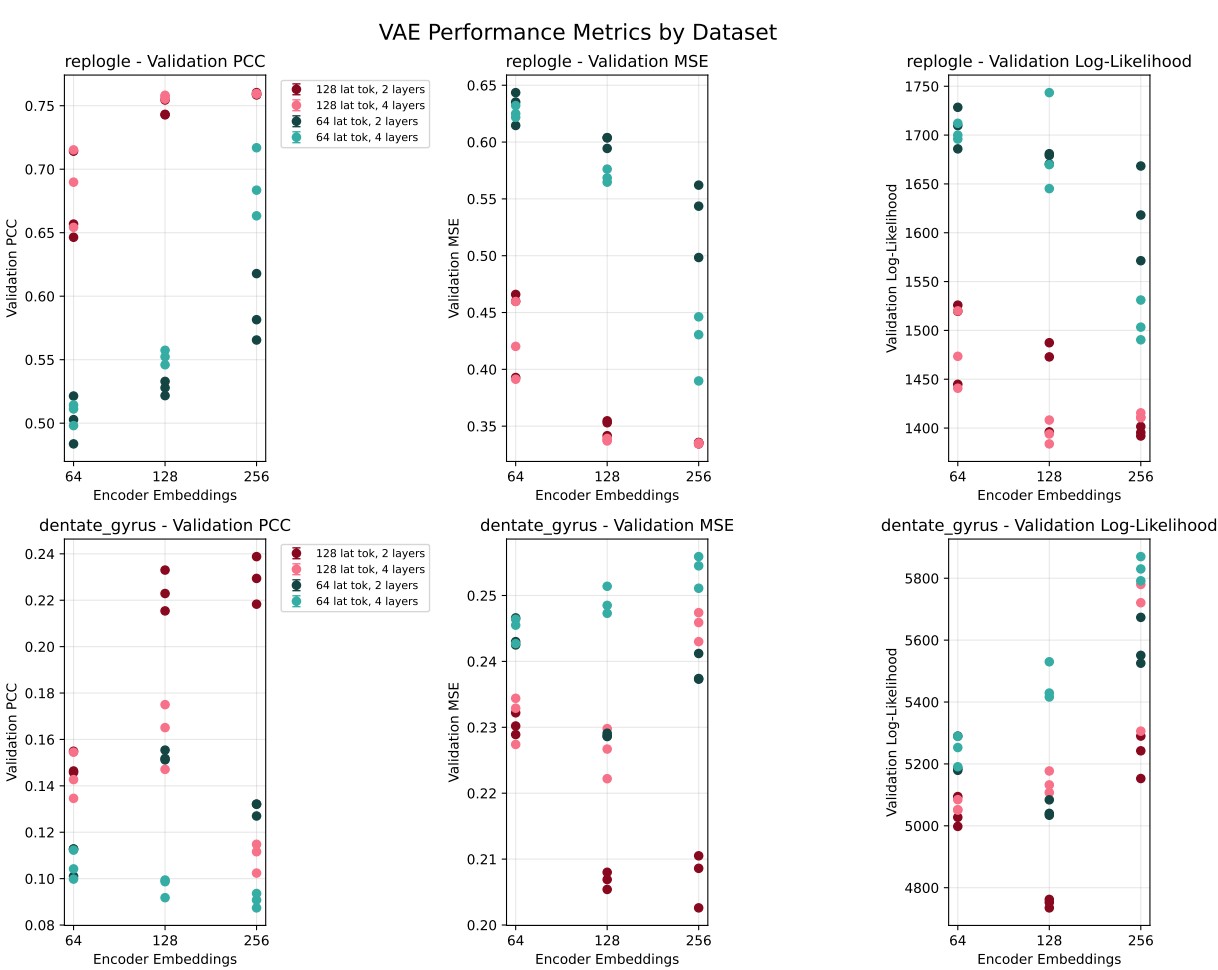

*Figure 4.* Ablations on VAE width, depth and number of latent tokens

where $\mathbf{W} \in \mathbb{R}^{1 \times D'}$ and $\mathbf{b} \in \mathbb{R}^{D'}$ are learnable parameters with $D' = \texttt{n\_embed}$. The final output $\mathbf{Z} \in \mathbb{R}^{B \times D \times D'}$.

This ensures that the dimensionality of the latent space, as well as the number of parameters for the encoder-decoder transformer layers is comparable between this implementation and the latent implementation.

We evaluate each aggregation method using three reconstruction metrics: log-likelihood, mean squared error (MSE), and Pearson correlation coefficient (PCC). Results are shown for two model configurations with embedding dimensions of 64 and 128, with error bars representing the standard deviation across three random seeds. The high errors bars for the mean and sum aggregation in the replogle datasets are due to high training instability that we observed in the training runs, making some runs to temporarily diverge.

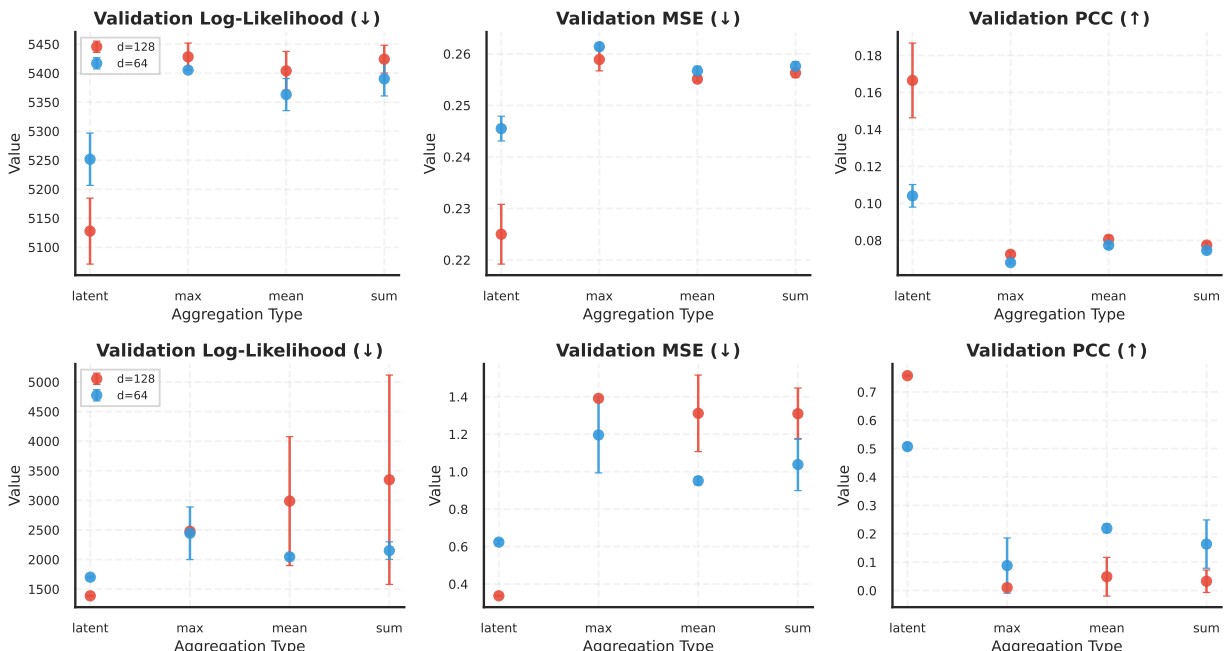

*Figure 5.* Ablations on different aggregation strategies between Set-Transformer-like max, sum and mean aggregation, versus the perceiver-style latent aggregation used in our model. Top: dentate gyrus dataset, Bottom: replogle dataset.

In Table 16, we present the model comparison on benchmark datasets, but only on differentially expressed genes. Our model is superior to the baselines across metrics and datasets. We should note that this additional evaluation is motivated by the difficulty in evaluating gnerative models, which is a long-standing issue in the field (Theis et al., 2016).

In Table 17 we present results on $R^2$ scores for all genes in the observational datasets, for unconditional and conditional settings.

In Figure 6, 7, and 8, a visualization of gene-wise variance for true and generated data on Dentate Gyrus, Tabula Muris, and HLCA, respectively, in the conditional settings. CFGen and scLDM properly recover true variance, with a slight tendency of scLDM to overestimate, while scDiffusion completely fails and underestimates the true Variance.

In Figure 9, we present a visualization of true and generated data for all models and all datasets in UMAP coordinates. In Figures 10 11 12, we present the conditional generation results in UMAP coordinates colored by the conditional class.

### K.2. Experiment 1: Interpretability results on cross-attention scores

The cross-attention encoder and decoder layers can be interpreted as pooling and unpooling operators on the gene tokens at input and output. The attention scores of the cross attention layers provide an interpretability tool to analyze how gene tokens map to the latent tokens. We analyzed the attention patterns of the encoder and decoder cross-attention layers in the dentate gyrus dataset. To this end, we computed the marker genes for each cell type annotated in the dataset. We further extracted the attention weights between the gene tokens and the latent tokens and averaged them across 2500 cells (where each cell type was sampled roughly the same proportion with respect to the original dataset). Using the genes v.

*Table 16.* Model performance comparison on unconditional and conditional cell generation benchmarks on all genes. W2, MMD$^2$ RBF, FD, 1-NN metrics calculated on 30 principal components are reported.

| Setting | Model | W2 ↓ | MMD$^2$ RBF ↓ | 1-NN → 0.5 | Prec ↑ | Rec ↑ |
|---|---|---|---|---|---|---|
| | | | Dentate Gyrus | | | |
| Uncond | scDiffusion | $8.616 \pm 0.263$ | $\mathbf{0.002} \pm 0.000$ | $0.632 \pm 0.002$ | $\mathbf{0.672} \pm 0.006$ | $0.496 \pm 0.004$ |
| | CFGen | $11.331 \pm 0.099$ | $0.009 \pm 0.000$ | $0.806 \pm 0.001$ | $0.237 \pm 0.003$ | $0.607 \pm 0.020$ |
| | scLDM (NB) | $\mathbf{7.668} \pm 0.169$ | $0.003 \pm 0.000$ | $0.634 \pm 0.002$ | $0.610 \pm 0.003$ | $0.599 \pm 0.007$ |
| | scLDM (Gauss) | $10.650 \pm 0.116$ | $0.003 \pm 0.000$ | $\mathbf{0.551} \pm 0.004$ | $0.133 \pm 0.002$ | $\mathbf{0.907} \pm 0.003$ |
| Cond | scDiffusion | $11.459 \pm 3.612$ | $0.035 \pm 0.027$ | $0.702 \pm 0.033$ | $\mathbf{0.584} \pm 0.070$ | $0.456 \pm 0.074$ |
| | CFGen | $9.420 \pm 1.734$ | $0.026 \pm 0.017$ | $0.785 \pm 0.035$ | $0.183 \pm 0.046$ | $0.701 \pm 0.042$ |
| | scLDM (NB) | $\mathbf{7.214} \pm 1.556$ | $\mathbf{0.011} \pm 0.006$ | $0.653 \pm 0.029$ | $0.546 \pm 0.073$ | $0.576 \pm 0.035$ |
| | scLDM (Gauss) | $9.447 \pm 2.205$ | $0.012 \pm 0.006$ | $\mathbf{0.562} \pm 0.008$ | $0.071 \pm 0.042$ | $\mathbf{0.912} \pm 0.033$ |
| | | | Tabula Muris | | | |
| Uncond | scDiffusion | $8.616 \pm 0.263$ | $\mathbf{0.002} \pm 0.000$ | $0.632 \pm 0.002$ | $\mathbf{0.672} \pm 0.006$ | $0.496 \pm 0.004$ |
| | CFGen | $11.331 \pm 0.099$ | $0.009 \pm 0.000$ | $0.806 \pm 0.001$ | $0.237 \pm 0.003$ | $0.607 \pm 0.020$ |
| | scLDM (NB) | $\mathbf{7.668} \pm 0.169$ | $0.003 \pm 0.000$ | $0.634 \pm 0.002$ | $0.610 \pm 0.003$ | $0.599 \pm 0.007$ |
| | scLDM (Gauss) | $10.650 \pm 0.116$ | $0.003 \pm 0.000$ | $\mathbf{0.551} \pm 0.004$ | $0.133 \pm 0.002$ | $\mathbf{0.907} \pm 0.003$ |
| Cond | scDiffusion | $11.459 \pm 3.612$ | $0.035 \pm 0.027$ | $0.702 \pm 0.033$ | $\mathbf{0.584} \pm 0.070$ | $0.456 \pm 0.074$ |
| | CFGen | $9.420 \pm 1.734$ | $0.026 \pm 0.017$ | $0.785 \pm 0.035$ | $0.183 \pm 0.046$ | $0.701 \pm 0.042$ |
| | scLDM (NB) | $\mathbf{7.214} \pm 1.556$ | $\mathbf{0.011} \pm 0.006$ | $0.653 \pm 0.029$ | $0.546 \pm 0.073$ | $0.576 \pm 0.035$ |
| | scLDM (Gauss) | $9.447 \pm 2.205$ | $0.012 \pm 0.006$ | $\mathbf{0.562} \pm 0.008$ | $0.071 \pm 0.042$ | $\mathbf{0.912} \pm 0.033$ |
| | | | HLCA | | | |
| Uncond | scDiffusion | $\mathbf{9.234} \pm 0.010$ | $0.002 \pm 0.000$ | $0.648 \pm 0.001$ | $\mathbf{0.602} \pm 0.002$ | $0.539 \pm 0.002$ |
| | CFGen | $12.651 \pm 0.031$ | $0.008 \pm 0.000$ | $0.804 \pm 0.002$ | $0.236 \pm 0.008$ | $0.607 \pm 0.007$ |
| | scLDM (NB) | $9.837 \pm 0.061$ | $0.004 \pm 0.000$ | $\mathbf{0.664} \pm 0.006$ | $0.557 \pm 0.005$ | $0.631 \pm 0.010$ |
| | scLDM (Gauss) | $10.058 \pm 0.046$ | $\mathbf{0.001} \pm 0.000$ | $0.549 \pm 0.001$ | $0.280 \pm 0.007$ | $\mathbf{0.857} \pm 0.005$ |
| Cond | scDiffusion | $10.000 \pm 2.502$ | $0.094 \pm 0.143$ | $0.755 \pm 0.048$ | $0.441 \pm 0.128$ | $0.492 \pm 0.142$ |
| | CFGen | $10.714 \pm 1.521$ | $0.087 \pm 0.123$ | $0.757 \pm 0.049$ | $0.230 \pm 0.092$ | $0.675 \pm 0.079$ |
| | scLDM (NB) | $\mathbf{9.071} \pm 1.302$ | $0.069 \pm 0.107$ | $0.661 \pm 0.054$ | $\mathbf{0.507} \pm 0.100$ | $0.625 \pm 0.086$ |
| | scLDM (Gauss) | $9.359 \pm 1.326$ | $\mathbf{0.057} \pm 0.111$ | $\mathbf{0.564} \pm 0.037$ | $0.219 \pm 0.142$ | $\mathbf{0.856} \pm 0.050$ |

*Table 17.* Model performance on $R^2$ metrics across datasets for the conditional models.

| Dataset | Model | $R^2$ Mean ↑ | $R^2$ Var ↑ | $R^2$ Zeros ↑ |
|---|---|---|---|---|
| Dentate Gyrus | cfgen | $1.00 \pm 0.00$ | $0.99 \pm 0.00$ | $1.00 \pm 0.00$ |
| | scDiffusion | $0.75 \pm 0.00$ | $-2.33 \pm 0.06$ | $< -10$ |
| | scLDM (NB) | $0.99 \pm 0.00$ | $0.99 \pm 0.00$ | $0.97 \pm 0.00$ |
| | scLDM (Gauss) | $1.00 \pm 0.00$ | $< -10$ | $< -10$ |
| HLCA | cfgen | $1.00 \pm 0.00$ | $0.99 \pm 0.00$ | $1.00 \pm 0.00$ |
| | scDiffusion | $0.69 \pm 0.00$ | $-1.42 \pm 0.03$ | $-6.92 \pm 0.02$ |
| | scLDM (NB) | $1.00 \pm 0.00$ | $0.99 \pm 0.00$ | $0.96 \pm 0.00$ |
| | scLDM (Gauss) | $1.00 \pm 0.00$ | $< -10$ | $0.00 \pm 0.00$ |
| Tabula Muris | cfgen | $1.00 \pm 0.00$ | $0.99 \pm 0.00$ | $0.99 \pm 0.00$ |
| | scDiffusion | $0.69 \pm 0.00$ | $-0.63 \pm 0.05$ | $-7.67 \pm 0.08$ |
| | scLDM (NB) | $1.00 \pm 0.00$ | $0.99 \pm 0.00$ | $0.97 \pm 0.00$ |
| | scLDM (Gauss) | $1.00 \pm 0.00$ | $< -10$ | $< -10$ |

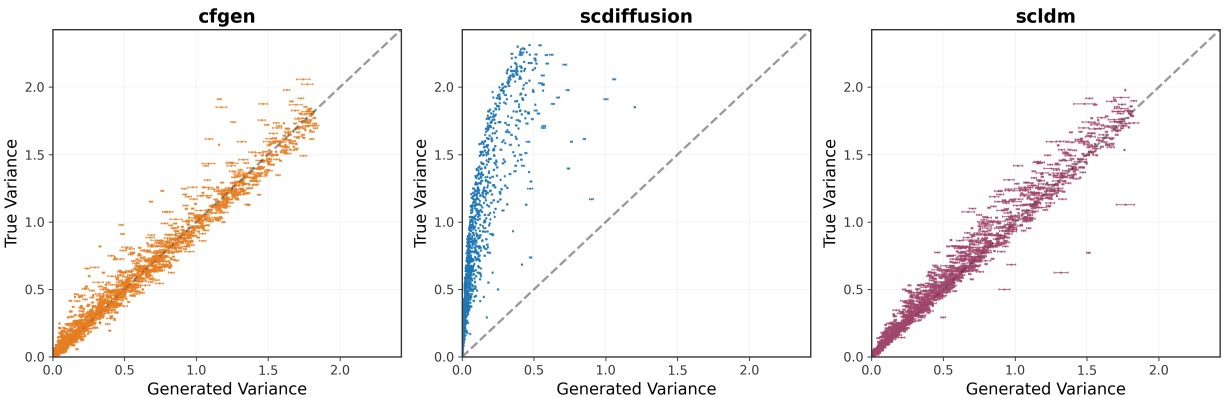

*Figure 6.* Visualization of the gene-wise variance for true and generated data for CFGen (left), scDiffusion (middle) our model (right), for the conditional generation settings on Dentate Gyrus. The error bars represent the standard errors over 3 seeds.

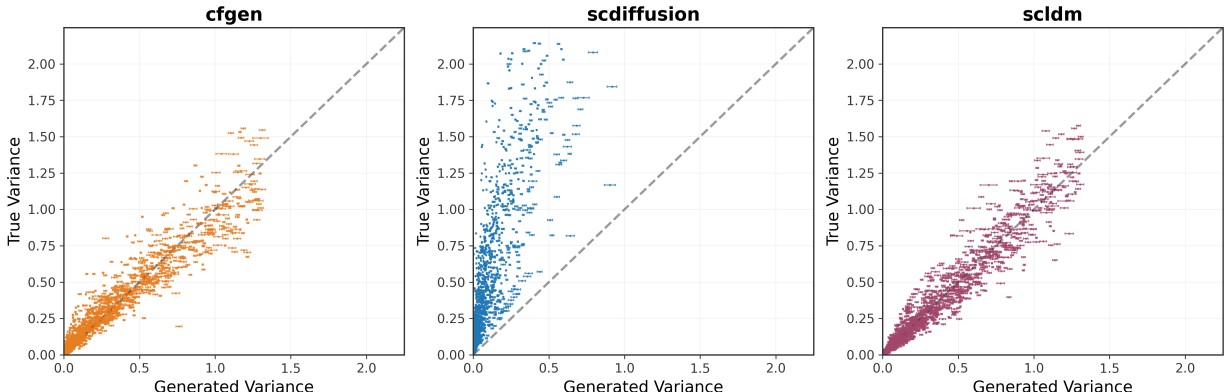

*Figure 7.* Visualization of the gene-wise variance for true and generated data for CFGen (left), scDiffusion (middle) our model (right), for the conditional generation settings on Tabula Muris. The error bars represent the standard errors over 3 seeds.

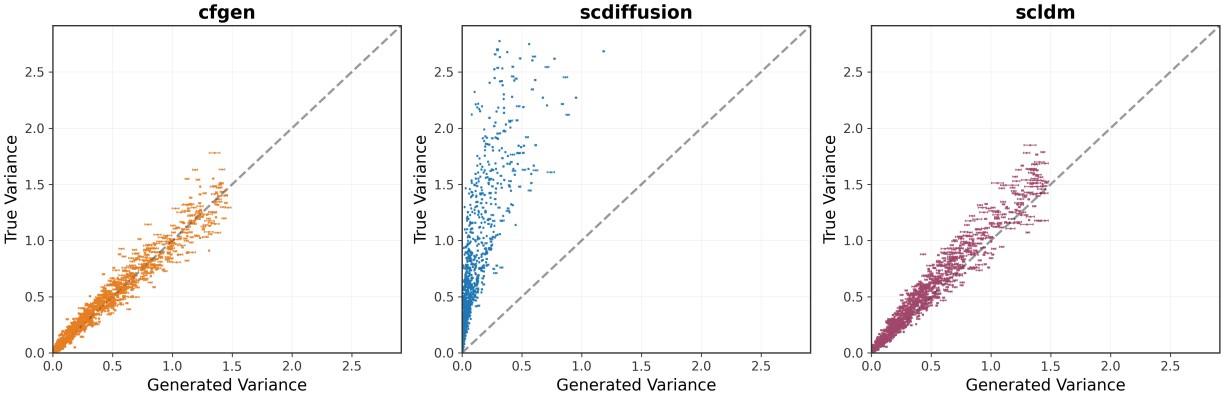

*Figure 8.* Visualization of the gene-wise variance for true and generated data for CFGen (left), scDiffusion (middle) our model (right), for the conditional generation settings on HLCA. The error bars represent the standard errors over 3 seeds.

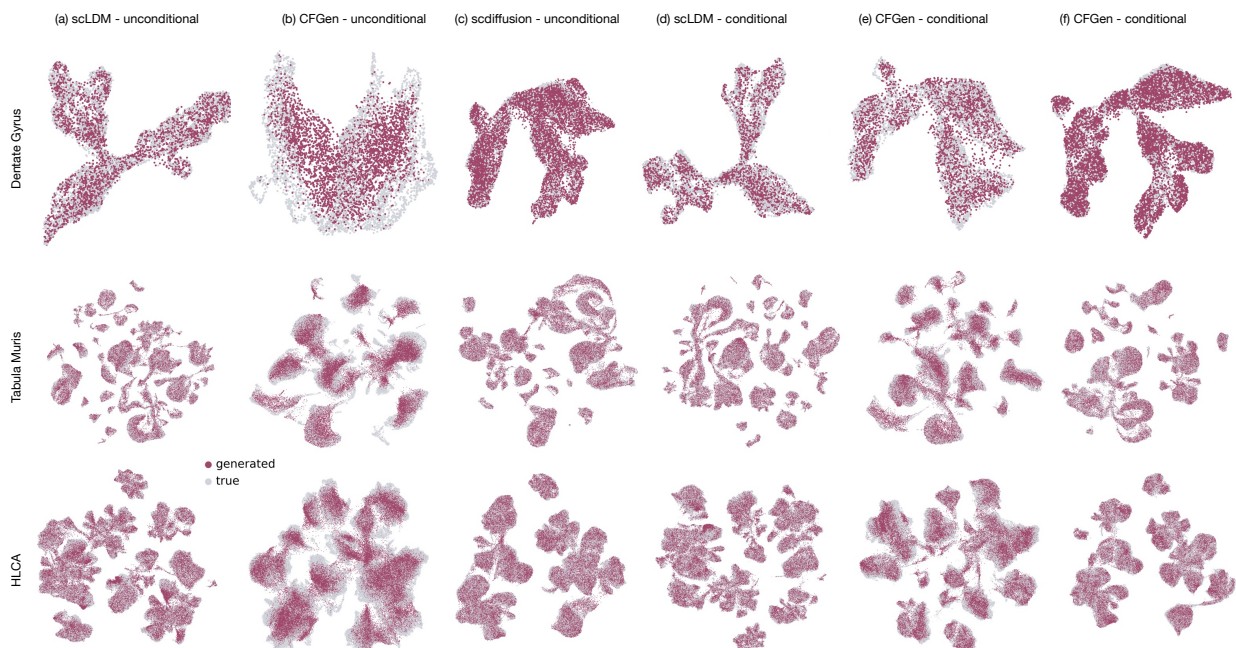

*Figure 9.* Visualization of the generation results for all datasets and models for conditional and unconditional generations. True and Generated gene expression is embedded in UMAP coordinates jointly, upon normalization, following standard Scanpy pipeline.

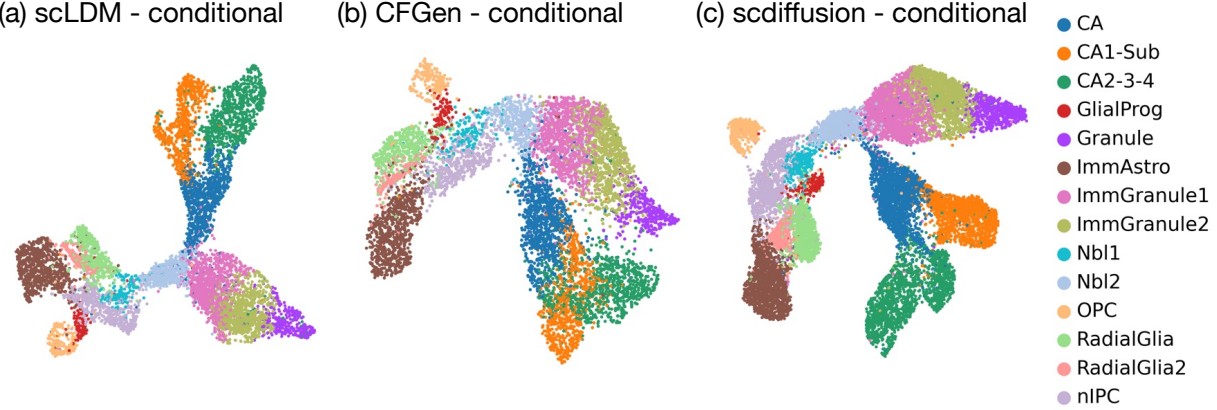

*Figure 10.* Visualization of the conditional generation results for the dentate gyrus dataset and all models, colored by the conditional label (clusters).

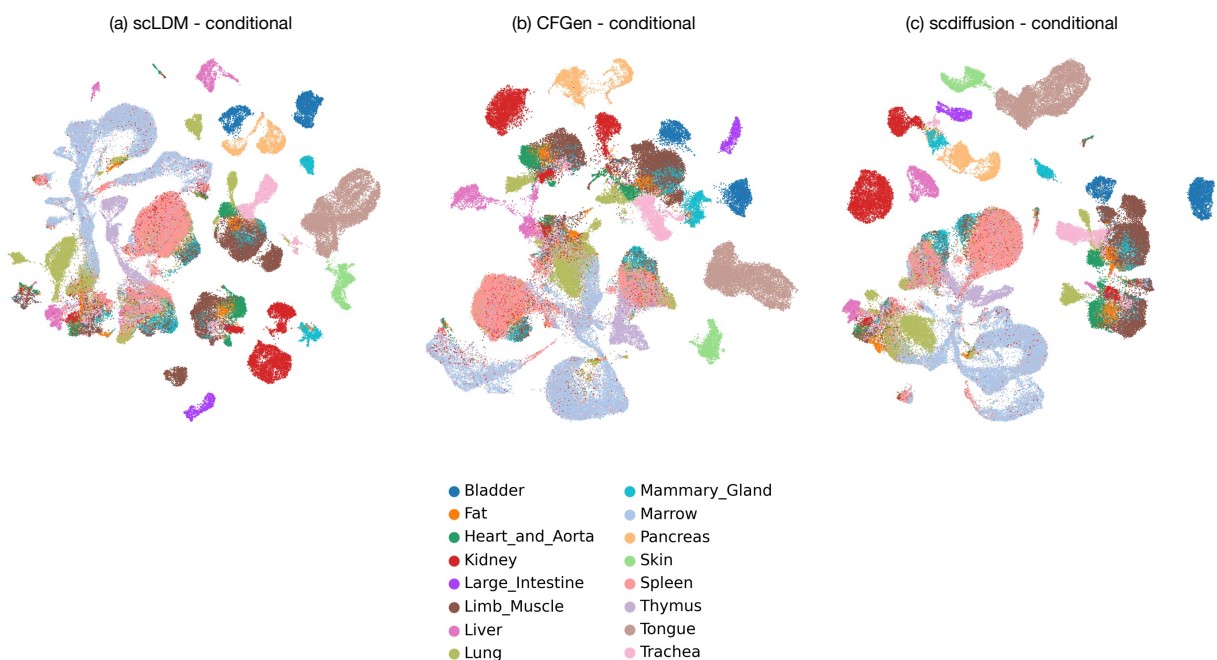

*Figure 11.* Visualization of the conditional generation results for the tabula muris dataset and all models, colored by the conditional label (tissue).

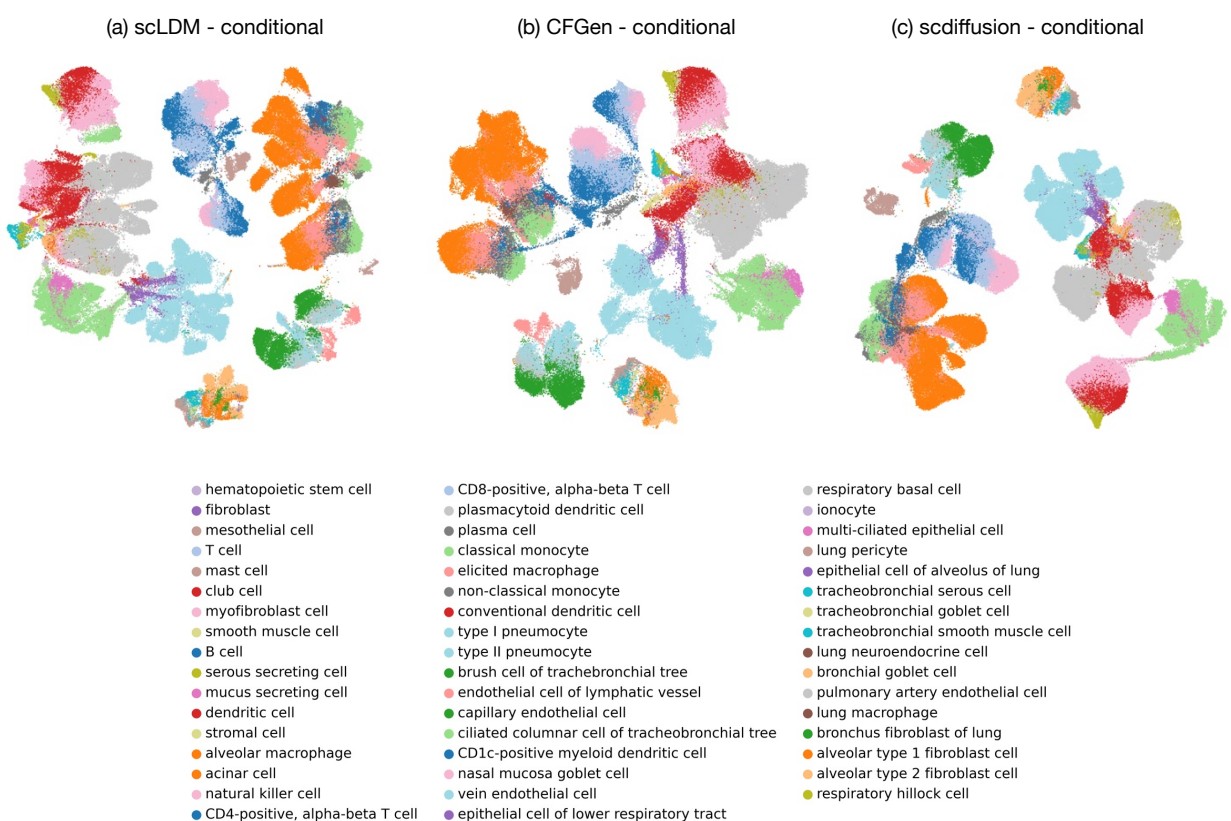

*Figure 12.* Visualization of the conditional generation results for the hlca dataset and all models, colored by the conditional label (cell type).

latent tokens average attention matrix, we computed enrichment scores for the cell type markers for each latent token using decoupler (Badia-I-Mompel et al., 2022). We visualized the enrichment score between each latent token and cell-type marker genes in Figures 13, 14, 15, where each column (latent tokens) and rows (marker genes gene set) was clustered using hierarchical clustering. We can observe that the latent tokens do show a selective enrichment for marker genes of specific cell types.

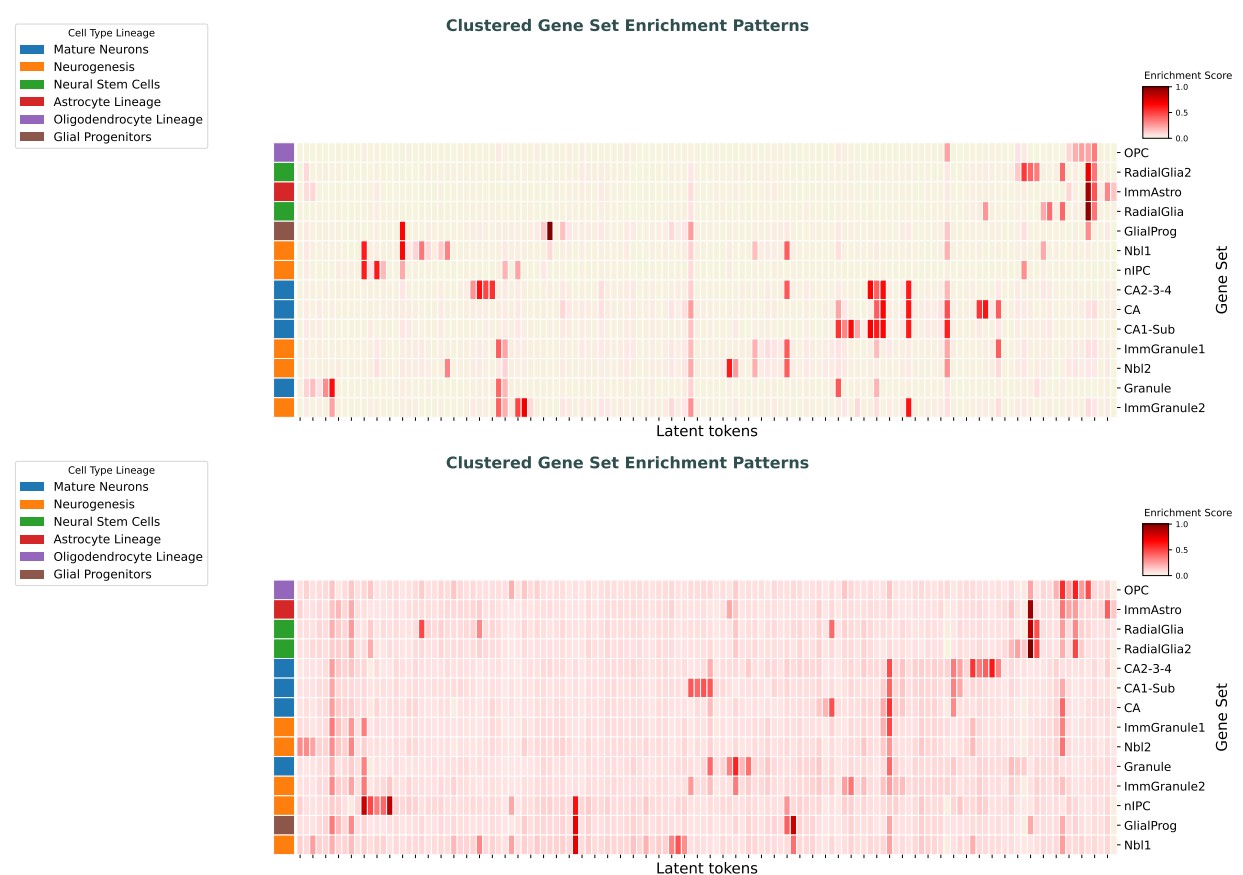

Figure 13. Enrichment scores for marker genes of cell-types in the dentate gyrus dataset of cross-attention for both the cross-attention encoder (top) and decoder (bottom)layers.

### K.3. Experiment 2: Reconstruction capabilities on perturbation datasets

The results presented in Table 18 demonstrate that our proposed approach significantly outperforms the scVI baseline across all evaluated metrics on the Parse1M and Replogle dataset. Most notably, our method achieves a substantially lower reconstruction error (RE) of about 310 nats compared to scVI's 432 nats, indicating better reconstructive capabilities. Furthermore, our approach yields a remarkable improvement in Pearson correlation coefficient (PCC), achieving 0.887 versus scVI's mediocre 0.351, which suggests that our model captures the underlying biological relationships much more effectively. The mean squared error (MSE) is also greatly reduced from 0.701 to 0.188, representing an approximately 73% reduction in reconstruction error. These consistent improvements across multiple evaluation criteria provide strong evidence that our method offers substantial advantages over scVI and indicate its great potential in analyzing biological data.

We further report gene-level $R^2$ scores for all perturbation datasets and models in Table 19. Our model shows competitivbe performance for the $R^2$ mean dramatic improved performance for the $R^2$ variance in the perturbation conditional generation task.

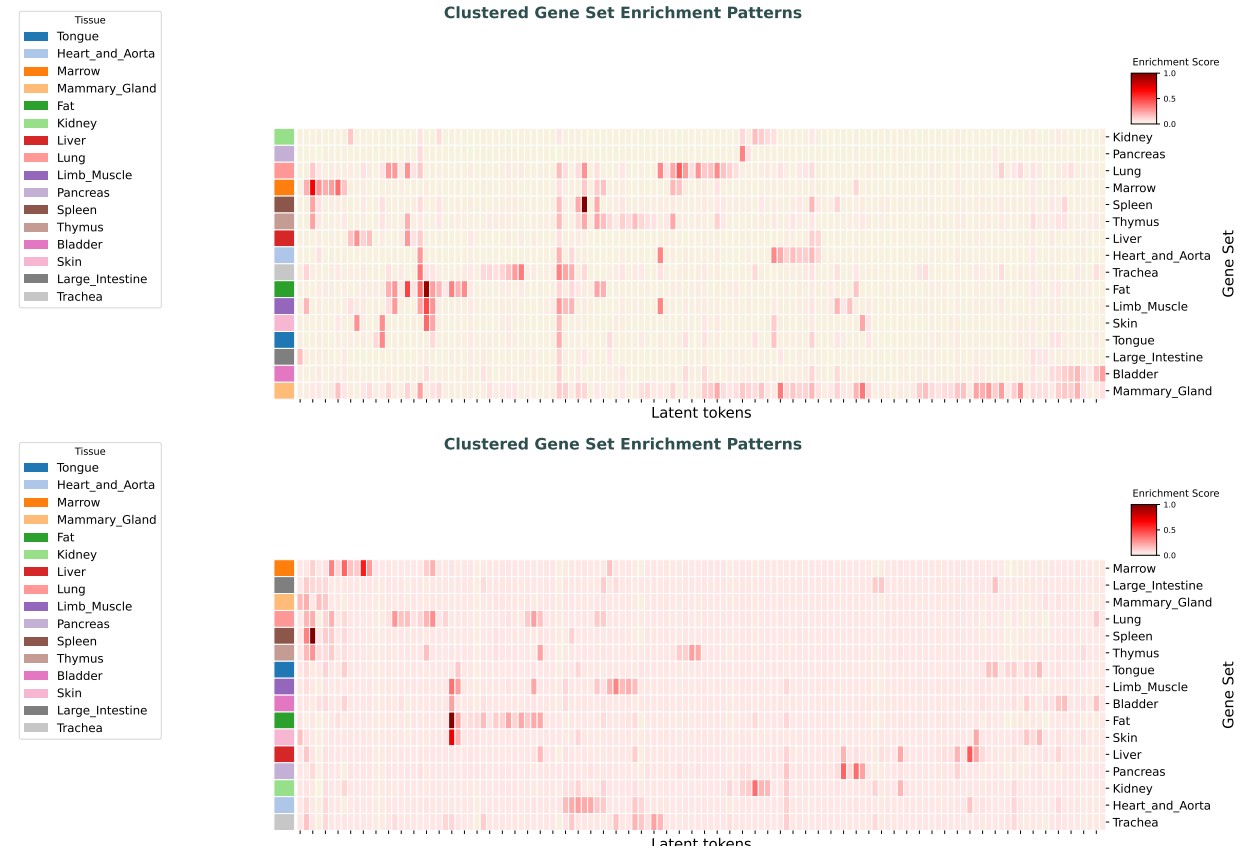

*Figure 14.* Enrichment scores for marker genes of cell-types in the tabula muris dataset of cross-attention for both the cross-attention encoder (top) and decoder (bottom) layers.

*Table 18.* Model performance comparison on cell reconstruction task.

| Dataset | Model | RE ↓ | PCC ↑ | MSE ↓ |
|---------|-------|------|-------|-------|
| Parse 1M | scVI | $432.41 \pm 0.08$ | $0.351 \pm 0.000$ | $0.701 \pm 0.001$ |
| | scLDM | $\mathbf{150.47} \pm 0.02$ | $\mathbf{0.887} \pm 0.000$ | $\mathbf{0.162} \pm 0.000$ |
| Replogle | scVI | $2144.86 \pm 0.35$ | $0.166 \pm 0.000$ | $0.703 \pm 0.001$ |
| | scLDM | $\mathbf{1602.66} \pm 1.38$ | $\mathbf{0.709} \pm 0.001$ | $\mathbf{0.290} \pm 0.001$ |

*Table 19.* Model performance on $R^2$ metrics across datasets.

| Dataset | Model | $R^2$ **Mean** ↑ | $R^2$ **Var** ↑ | $R^2$ **Zeros** ↑ |
|---------|-------|------------------|------------------|-------------------|
| Parse 1M | Cellflow | $1.00 \pm 0.00$ | $0.39 \pm 0.00$ | $< -10$ |
| | CPA | $1.00 \pm 0.00$ | $< -10$ | $< -10$ |
| | scGPT | $-1.17 \pm 0.02$ | $< -10$ | $< -10$ |
| | scLDM (NB, $\omega$=1) | $0.98 \pm 0.00$ | $0.98 \pm 0.00$ | $0.94 \pm 0.00$ |
| | scLDM (Gauss, $\omega$=1) | $1.00 \pm 0.00$ | $0.09 \pm 0.00$ | $0.00 \pm 0.00$ |
| | scVI | $-0.27 \pm 0.06$ | $< -10$ | $< -10$ |
| | STATE | $-3.40 \pm 0.09$ | $< -10$ | $0.90 \pm 0.00$ |
| Replogle | Cellflow | $0.99 \pm 0.00$ | $< -10$ | $0.00 \pm 0.00$ |
| | CPA | $1.00 \pm 0.00$ | $< -10$ | $0.00 \pm 0.00$ |
| | scGPT | $< -10$ | $< -10$ | $0.00 \pm 0.00$ |
| | scLDM (NB, $\omega$=1) | $0.99 \pm 0.00$ | $0.81 \pm 0.00$ | $0.83 \pm 0.00$ |
| | scLDM (Gauss, $\omega$=1) | $1.00 \pm 0.00$ | $< -10$ | $0.00 \pm 0.00$ |
| | scVI | $0.91 \pm 0.00$ | $< -10$ | $0.00 \pm 0.00$ |
| | STATE | $0.51 \pm 0.02$ | $-8.81 \pm 4.35$ | $0.80 \pm 0.05$ |

**Clustered Gene Set Enrichment Patterns**

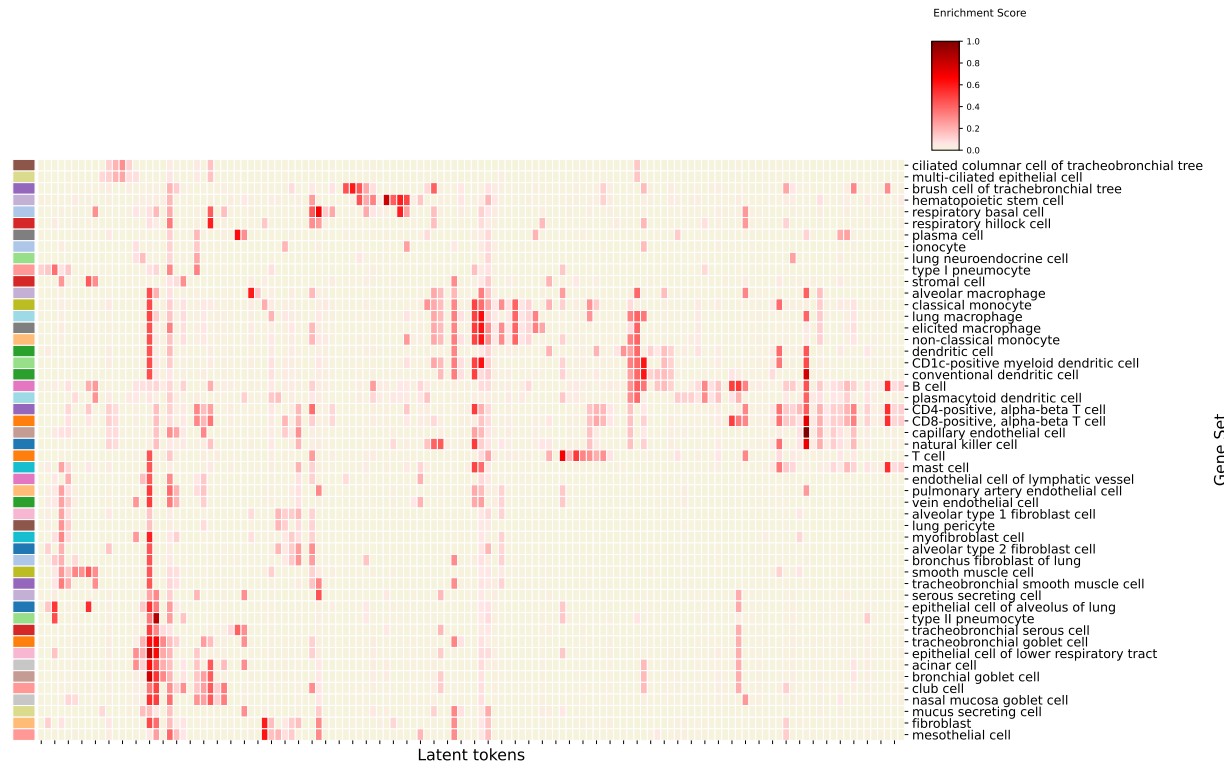

**Clustered Gene Set Enrichment Patterns**

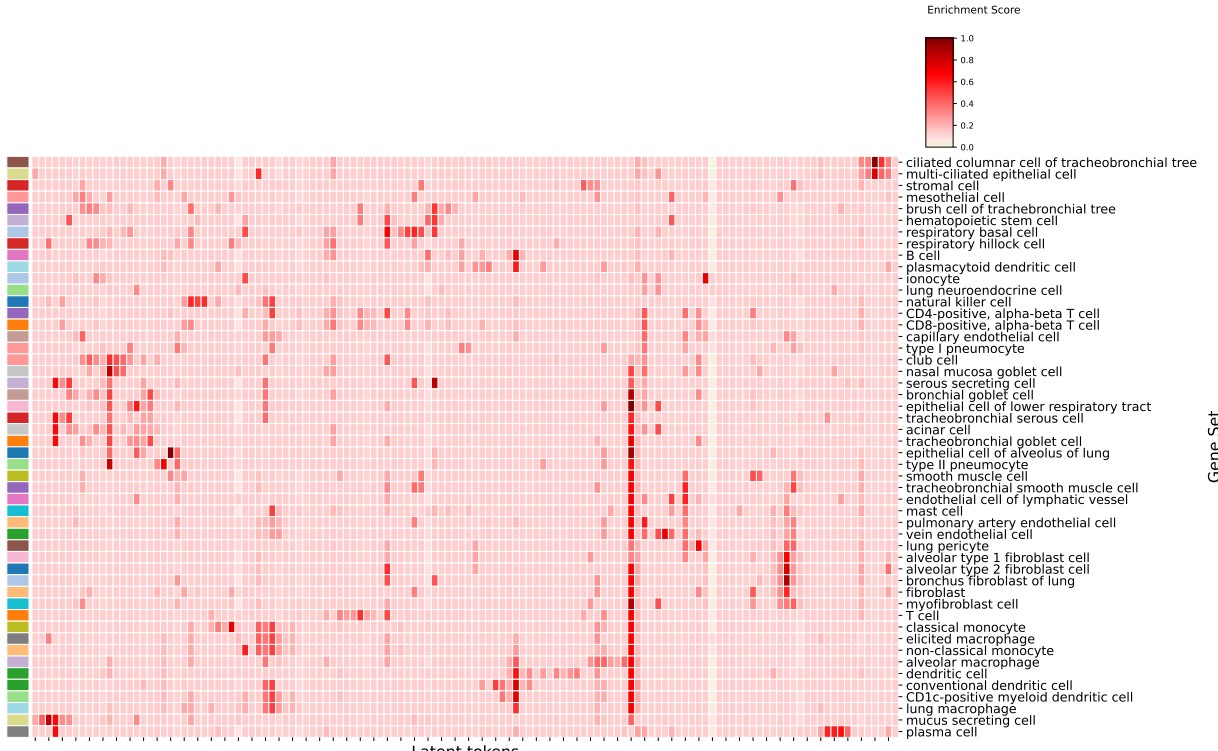

*Figure 15.* Enrichment scores for marker genes of cell-types in the hlca dataset of cross-attention for both the cross-attention encoder (top) and decoder (bottom) layers.

### K.4. Experiment 2: A comparison between *additive* and *joint* conditioning in classifier-free guidance

Table 22 compares the performance of our scLDM model using two different classifier-free guidance approaches for conditional cell generation: the additive steering method proposed by Palma et al. (2025a) and our joint attribute steering method. Across all metrics (Wasserstein-2 distance, MMD$^2$ RBF, and Fréchet Distance) and both datasets (Parse 1M and Replogle), the joint approach consistently outperforms the additive approach, demonstrating substantial improvements in generation quality.

*Table 20.* Model performance comparison on conditional cell generation on Parse1M and Replogle. For these results scLDM was trained using the classifier-free guidance approach proposed in (Palma et al., 2025a) (additive) and ours (joint).

| Dataset | Model | W2 ↓ | MMD$^2$ RBF ↓ | FD ↓ |
|---------|-------|------|---------------|------|
| Parse 1M | scLDM (additive) | $15.850 \pm 0.073$ | $0.129 \pm 0.004$ | $109.196 \pm 2.933$ |
| | scLDM (joint) | $\mathbf{12.455} \pm 0.001$ | $\mathbf{0.027} \pm 0.000$ | $\mathbf{18.145} \pm 0.068$ |
| Replogle | scLDM (additive) | $18.538 \pm 0.058$ | $0.451 \pm 0.003$ | $255.510 \pm 2.163$ |
| | scLDM (joint) | $\mathbf{11.288} \pm 0.011$ | $\mathbf{0.200} \pm 0.001$ | $\mathbf{53.555} \pm 0.210$ |

### K.5. Experiment 2: A comparison of guidance weights in classifier-free guidance

We further evaluated whether adding a difference guidance weight would improve the results in terms of generation metrics in the perturbational datasets. We report results in Table 21.

*Table 21.* Effect of classifier-free guidance weight $\omega$ on conditional cell generation performance. Lower guidance weights yield better distributional metrics.

| Dataset | Model | W2 ↓ | MMD$^2$ RBF ↓ | FD ↓ |
|---------|-------|------|---------------|------|
| Parse 1M | scLDM (NB, $\omega = 1$) | $\mathbf{12.457} \pm 0.045$ | $\mathbf{0.027} \pm 0.002$ | $\mathbf{18.136} \pm 0.903$ |
| | scLDM (NB, $\omega = 5$) | $12.902 \pm 0.087$ | $0.071 \pm 0.004$ | $43.363 \pm 2.246$ |
| | scLDM (NB, $\omega = 10$) | $13.638 \pm 0.111$ | $0.122 \pm 0.006$ | $69.769 \pm 3.363$ |
| Replogle | scLDM (NB, $\omega = 1$) | $\mathbf{11.292} \pm 0.033$ | $\mathbf{0.200} \pm 0.002$ | $\mathbf{53.750} \pm 0.666$ |
| | scLDM (NB, $\omega = 5$) | $12.900 \pm 0.069$ | $0.320 \pm 0.004$ | $105.365 \pm 1.935$ |
| | scLDM (NB, $\omega = 10$) | $14.911 \pm 0.091$ | $0.436 \pm 0.005$ | $166.877 \pm 3.036$ |

### K.6. Experiment 2: Perturbation prediction metrics

**Perturbation results using pertrubation metrics from the `cell-eval` package.** We further evaluated our models and all baselines on the generated results for the test set perturbations using the `cell-eval` package [7] (Figure 16). Our model, although not explicitly designed for perturbation prediction, is competitive across various metrics compared to the baselines on both datasets considered.

### K.7. Experiment 2: Generation metrics on Differentially Expressed Genes

Table 22 presents generation metrics for the same datasets and models as in Table 3, but evaluated specifically on Differentially Expressed Genes (DEGs). To identify DEGs, we applied Scanpy's `tl.rank_genes_groups` method using the Wilcoxon rank-sum test for each perturbation. For perturbations with more than 10 identified DEGs, we computed generation metrics directly in the gene space rather than in PCA-reduced space. This evaluation offers insight into the model's performance on the most biologically relevant genes affected by each perturbation.

### K.8. Experiment 3

For the last experiment, we trained three VAEs for our approach (scLDM-VAE): with 20M parameters, 70M parameters, and 270M parameters. Further, we evaluated the resulting models using embeddings on a downstream task (classification) for two out-of-distribution datasets (COVID-19 and Tabula Sapiens 2.0).

First, we evaluated these three versions of our model using reconstruction metrics on the dataset they were trained on,

---

[7] https://github.com/ArcInstitute/cell-eval

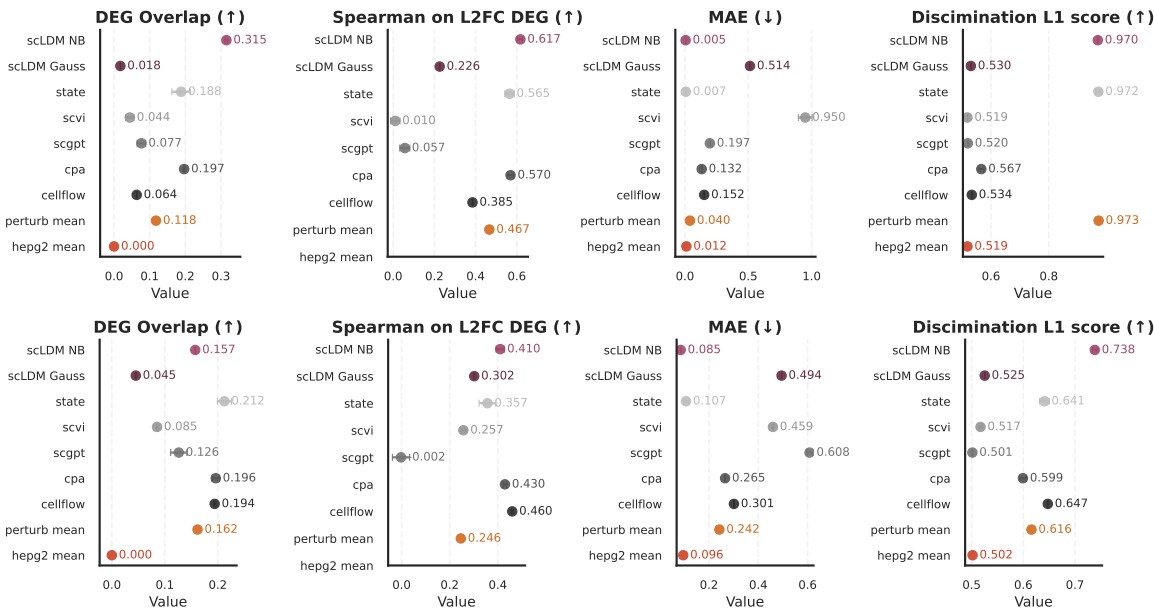

*Figure 16.* Evaluation metrics from the `cell-eval` package for the Parse 1M dataset (top) and Replogle dataset (bottom)

*Table 22.* Model performance comparison on conditional cell generation on Parse1M and Replogle on differentially expressed genes.

| Dataset | Model | W2 ↓ | MMD$^2$ RBF ↓ | FD ↓ | 1-NN → 0.5 | Precision ↑ | Recall ↑ |
|---|---|---|---|---|---|---|---|
| | scVI | $26.131 \pm 0.721$ | $1.112 \pm 0.013$ | $703.413 \pm 39.084$ | $0.874 \pm 0.013$ | $0.664 \pm 0.043$ | $0.000 \pm 0.000$ |
| | CPA | $\underline{18.168} \pm 0.426$ | $1.378 \pm 0.015$ | $342.468 \pm 16.009$ | $0.670 \pm 0.020$ | $\mathbf{0.996} \pm 0.003$ | $0.000 \pm 0.000$ |
| | Cellflow | $\mathbf{17.990} \pm 0.516$ | $0.036 \pm 0.002$ | $114.441 \pm 10.376$ | $\underline{0.548} \pm 0.007$ | $0.854 \pm 0.011$ | $0.027 \pm 0.005$ |
| Parse 1M | scGPT | $25.853 \pm 0.513$ | $1.813 \pm 0.025$ | $687.085 \pm 27.790$ | $0.999 \pm 0.000$ | $0.504 \pm 0.054$ | $0.000 \pm 0.000$ |
| | STATE | $21.304 \pm 0.506$ | $0.576 \pm 0.005$ | $346.857 \pm 16.616$ | $0.872 \pm 0.012$ | $\underline{0.904} \pm 0.016$ | $0.000 \pm 0.000$ |
| | scLDM (NB, $\omega$=1) | $20.149 \pm 0.619$ | $\mathbf{0.007} \pm 0.001$ | $\mathbf{18.059} \pm 1.760$ | $0.573 \pm 0.005$ | $0.333 \pm 0.009$ | $\underline{0.345} \pm 0.014$ |
| | scLDM (Gauss, $\omega$=1) | $21.813 \pm 0.697$ | $\underline{0.009} \pm 0.001$ | $\underline{24.231} \pm 2.308$ | $\mathbf{0.519} \pm 0.003$ | $0.078 \pm 0.007$ | $\mathbf{0.788} \pm 0.008$ |
| | scVI | $12.587 \pm 0.180$ | $0.440 \pm 0.003$ | $184.379 \pm 4.090$ | $0.812 \pm 0.004$ | $0.608 \pm 0.007$ | $0.001 \pm 0.000$ |
| | CPA | $\underline{9.871} \pm 0.131$ | $0.756 \pm 0.006$ | $114.196 \pm 2.715$ | $0.769 \pm 0.004$ | $\mathbf{0.965} \pm 0.004$ | $0.000 \pm 0.000$ |
| | Cellflow | $\mathbf{9.311} \pm 0.131$ | $0.338 \pm 0.004$ | $\mathbf{81.365} \pm 2.468$ | $0.651 \pm 0.004$ | $\underline{0.959} \pm 0.003$ | $0.033 \pm 0.002$ |
| Replogle | scGPT | $27.690 \pm 0.342$ | $3.043 \pm 0.011$ | $892.139 \pm 20.073$ | $1.000 \pm 0.000$ | $0.028 \pm 0.005$ | $0.000 \pm 0.000$ |
| | STATE | $15.390 \pm 0.203$ | $0.549 \pm 0.006$ | $236.127 \pm 5.047$ | $0.971 \pm 0.001$ | $0.177 \pm 0.007$ | $0.007 \pm 0.001$ |
| | scLDM (NB, $\omega$=1) | $11.668 \pm 0.183$ | $\mathbf{0.174} \pm 0.002$ | $\underline{100.909} \pm 3.973$ | $\underline{0.617} \pm 0.002$ | $0.356 \pm 0.009$ | $\underline{0.547} \pm 0.005$ |
| | scLDM (Gauss, $\omega$=1) | $13.846 \pm 0.239$ | $\underline{0.195} \pm 0.002$ | $163.443 \pm 7.340$ | $\mathbf{0.539} \pm 0.002$ | $0.224 \pm 0.009$ | $\mathbf{0.899} \pm 0.003$ |

namely, Human Census Data from CellxGene[8]. Looking at Table 23, we can see a clear relationship between model size and reconstruction performance for the scLDM-VAE models on the CellxGene dataset. As the number of parameters increases from 20M to 270M, all three metrics show substantial improvement: reconstruction error (RE) decreases, Pearson correlation coefficient (PCC) increases, and mean squared error (MSE) drops. These results demonstrate that scaling up the scLDM-VAE architecture yields consistent performance gains across all reconstruction metrics, with the 270M parameter model achieving approximately 17% lower reconstruction error and 18% higher correlation compared to the smallest 20M model.

*Table 23.* Reconstruction performance comparison of our scLDM-VAEs with varying number of parameters: 20M, 70M, and 270M.

| Dataset | Model | RE $\downarrow$ | PCC $\uparrow$ | MSE $\downarrow$ |
|---|---|---|---|---|
| CellxGene Census | scLDM-VAE (20M) | 1742.7 | 0.661 | 0.137 |
| | scLDM-VAE (70M) | 1552.7 | 0.739 | 0.106 |
| | scLDM-VAE (270M) | **1441.7** | **0.783** | **0.091** |

Table 24 presents a comprehensive performance comparison of various models on the COVID-19 dataset, averaged across all donors. Our scLDM model with 270M parameters achieves the best performance across all metrics (ROC AUC, PR AUC, F1 Score, Recall, and Precision), demonstrating consistent improvements over both transformer-based baselines (TranscriptFormer, scGPT, Geneformer, UCE) and traditional VAE approaches (scVI, AIDO.Cell).

*Table 24.* COVID-19 Model Performance Summary (Averaged Across All Donors). **Bold** indicates the best performing model.

| Model | ROC AUC | PR AUC | F1 Score | Recall | Precision |
|---|---|---|---|---|---|
| scLDM (270M) | **0.909**± 6e-04 | **0.877**± 0.001 | **0.820**± 0.001 | **0.836**± 0.001 | **0.806**± 0.001 |
| TranscriptFormer | 0.905± 4e-04 | 0.874± 9e-04 | 0.814± 0.002 | 0.829± 0.003 | 0.801± 0.001 |
| scLDM (70M) | 0.905± 5e-04 | 0.872 ± 0.001 | 0.815 ± 0.001 | 0.83 ± 0.002 | 0.801 ± 0.001 |
| scLDM (20M) | 0.902± 5e-04 | 0.869 ± 0.001 | 0.811 ± 0.001 | 0.827 ± 0.001 | 0.797 ± 0.002 |
| UCE | 0.876± 5e-04 | 0.834± 0.002 | 0.775± 8e-04 | 0.781± 0.001 | 0.771± 0.002 |
| scGPT | 0.876± 4e-04 | 0.831± 0.001 | 0.779± 9e-04 | 0.793± 0.002 | 0.766± 0.001 |
| Geneformer | 0.866± 6e-04 | 0.815± 0.001 | 0.768± 0.001 | 0.781± 0.003 | 0.757± 0.001 |
| AIDO.Cell | 0.821± 7e-04 | 0.753± 9e-04 | 0.717± 8e-04 | 0.729± 0.002 | 0.708± 0.001 |
| scVI | 0.800± 7e-04 | 0.709± 0.001 | 0.675± 0.001 | 0.680± 0.002 | 0.680± 0.001 |

Figure 17 visualizes the receiver operating characteristic (ROC) and precision-recall (PR) curves for all models on the COVID-19 classification task. The curves further illustrate the superior discriminative performance of scLDM variants, with the 270M parameter model achieving the highest area under both curves, consistent with the quantitative results in Table 24.

Table 25 summarizes model performance on the Tabula Sapiens 2.0 dataset, averaged across all tissues. Notably, the smallest scLDM variant (20M parameters) achieves the highest F1 score (0.804), slightly outperforming both larger scLDM models and all baseline methods, suggesting that model scale may have diminishing returns on this particular cell type classification task.

---

[8]https://cellxgene.cziscience.com/

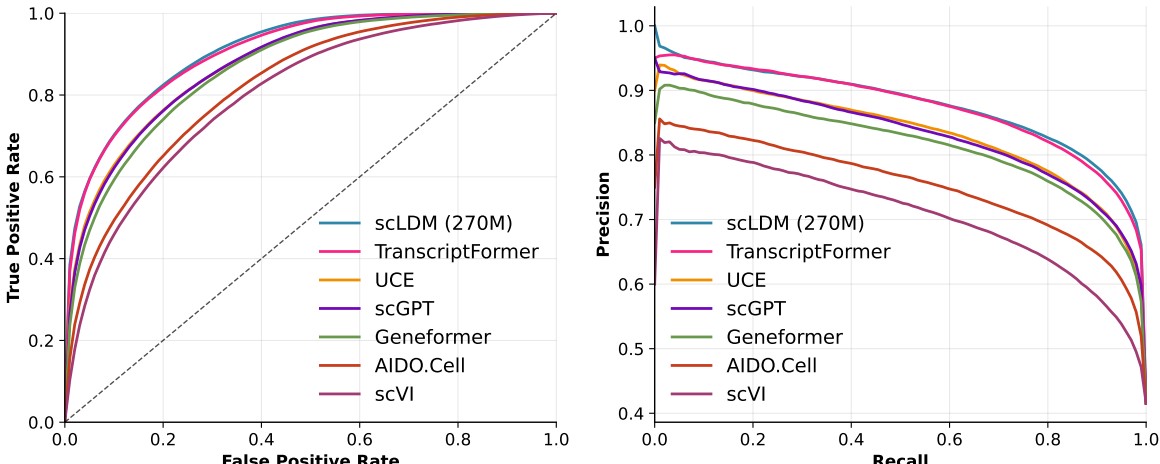

*Figure 17.* Precision-recall and receiver operator curves for COVID-19 data.

*Table 25.* Tabula Sapiens 2.0 model performance summary (averaged across all tissues)

| Model | F1 Score | Recall | Precision |
|---|---|---|---|
| scLDM-20M | **0.804** ± 0.002 | **0.805** ± 0.002 | **0.812** ± 0.002 |
| scLDM-270M | 0.802 ± 0.002 | 0.803 ± 0.002 | 0.811 ± 0.002 |
| scLDM-70M | 0.802 ± 0.002 | 0.802 ± 0.002 | 0.810 ± 0.002 |
| scGPT | 0.800 ± 0.002 | 0.802 ± 0.002 | 0.806 ± 0.002 |
| scVI | 0.799 ± 0.002 | 0.794 ± 0.002 | 0.814 ± 0.003 |
| TranscriptFormer | 0.799 ± 0.002 | 0.800 ± 0.002 | 0.802 ± 0.002 |
| UCE | 0.796 ± 0.002 | 0.797 ± 0.001 | 0.801 ± 0.003 |
| Geneformer | 0.777 ± 0.002 | 0.776 ± 0.002 | 0.786 ± 0.003 |
| AIDO.Cell | 0.724 ± 0.002 | 0.715 ± 0.002 | 0.748 ± 0.003 |

