# OpenReview forum: "Scalable Single-Cell Gene Expression Generation with Latent Diffusion Models"
_ICML.cc/2026/Conference — ICML 2026 regular_

### Official Review · Reviewer_jW4g · 2026-03-03

**Soundness:** 2
**Presentation:** 3
**Significance:** 3
**Originality:** 2
**Overall Recommendation:** 4
**Confidence:** 2

**Summary:**

This paper proposes scLDM, a scalable generative framework for single-cell RNA-seq data. It combines an exchangeable transformer-based VAE with latent diffusion modeling to respect gene permutation invariance while enabling high-quality generation. A unified MCAB module performs invariant pooling and equivariant decoding. Experiments show improved reconstruction accuracy, stronger distributional alignment, and competitive or superior conditional generation performance compared to prior single-cell generative models.

**Compliance With Llm Reviewing Policy:**

Affirmed.

**Key Questions For Authors:**

1. **Zero Filtering:** What is the sensitivity of performance to the zero-filtering approach? Are there any cases where getting rid of zeros affects biological information?
1. **Quantitative Explanation for Scalability:** Can authors provide more quantitative results like runtime or GPU memory usage to justify that this architecture is scalable?

**Limitations:**

See the weakness and question section.

**Strengths And Weaknesses:**

### **Strengths:**

1. **Good Consistency:** This work focuses on the idea that permuting gene IDs should not change cell state. And this principle is well considered through the design of the encoder, decoder, and latent representation.
1. **Perturbation Setting:** Apart from in distribution generation, authors also include the OOD data in Parse 1M and Replogle. These tests make the result and model design more convincing.

### **Weaknesses:**

1. **Novelty Clarification:** The design of scLAM is well motivated, especially the MCAB part. However, it makes me feel like the main contribution seems to be integrating concepts like latent diffusion, DiT, etc with into the single-cell gene expression scenario. Still, this may be meaningful, but it can be stronger if the authors more explicitly clarify novelty relative to prior works.

---

> ### Author Rebuttal · Authors · 2026-03-30
>
> **W1. Novelty clarification**
>
> We appreciate the reviewer's observation and agree that we can be more explicit about the novelty. While scLDM draws on established components (latent diffusion, DiT, flow matching), the contribution is not merely an integration of existing techniques into a new domain. Specifically:
>
> (i) The MCAB module serves a dual role: permutation-invariant pooling in the encoder and permutation-equivariant unpooling in the decoder, through a single architectural component, eliminating the need for separate pooling/unpooling modules as in SetTransformer (PMA + ISAB) or SetVAE (dual latent variables + hierarchical structure). We formalize the equivariance/invariance properties in Properties 1–4 (Appendix C).
>
> (ii) The input processing strategy (zero-filtering with PAD tokens) is a non-trivial design choice specific to the extreme sparsity of scRNA-seq data (~70–90% zeros). It reduces the effective context length by an order of magnitude while preserving reconstruction quality (Table 14, Appendix K.1), enabling the Transformer to scale to full gene sets of 20k+ genes, something that no prior Transformer-based single-cell generative model achieves without HVG preselection.
>
> (iii) The joint classifier-free guidance formulation for multi-attribute conditioning (Eq. 6) substantially outperforms the additive guidance used in CFGen (Table 19), demonstrating that learning joint condition embeddings is critical for capturing perturbation–context interactions.
>
> We will revise the manuscript to make these distinctions more prominent, particularly in the introduction and related work sections.
>
> **Q1. Zero filtering sensitivity**
>
> We thank the reviewer for this question. Importantly, zero-expression genes are filtered only at the encoder input, the decoder produces parameters for *all* queried genes, including those with zero expression. The Negative Binomial likelihood naturally places probability mass at zero, so the model can predict genuine zero expression. In Table 14 (Appendix K.1), we directly compare our zero-padding strategy against using the full gene context as input. The zero-padding approach achieves better reconstruction performance (lower RE, lower MSE, higher PCC) while being substantially more computationally efficient, because the Transformer processes ~2–5k expressed genes rather than ~20k total genes. Furthermore, $R^2$ Zeros scores across datasets (0.97 on Dentate Gyrus, 0.96 on HLCA, 0.94 on Parse 1M; Tables 16 and 18) confirm that scLDM accurately recovers sparsity patterns.
>
> **Q2. Quantitative scalability results**
>
> We thank the reviewer for this suggestion. We have compiled training benchmarks across all datasets, measured on NVIDIA H100-80GB GPUs with Distributed Data Parallel (DDP) across 8 GPUs. We report wall-clock training time and peak GPU memory below:
>
> | Dataset | Likelihood | Time 8×A100 (h) | Time 1×A100 (h) | Peak Mem/GPU (GB) |
> |:---|:---:|:---:|:---:|:---:|
> | Dentate Gyrus | Gauss | 0.30 ± 0.00 | 2.17 ± 0.00 | 32.50 |
> | Dentate Gyrus | NB | 0.30 ± 0.00 | 2.17 ± 0.00 | 32.50 |
> | Tabula Muris | Gauss | 14.61 ± 0.19 | 106.0 | 34.10 |
> | Tabula Muris | NB | 14.50 ± 0.28 | 105.2 | 33.18 |
> | HLCA | Gauss | 17.01 ± 0.11 | 123.4 | 49.59 |
> | HLCA | NB | 16.97 ± 0.15 | 123.1 | 44.72 |
> | Replogle | Gauss | 2.91 ± 0.02 | 21.1 | 5.92 |
> | Replogle | NB | 3.19 ± 0.04 | 23.1 | 5.83 |
> | Parse 1M | Gauss | 5.40 ± 0.01 | 39.2 | 5.92 |
> | Parse 1M | NB | 5.94 ± 0.04 | 43.1 | 5.83 |
>
> Two patterns are worth highlighting. First, memory scales with the number of genes (context length), not the number of cells — the 2k-gene perturbational datasets require only ~6 GB/GPU regardless of dataset size, while the full-gene-set datasets require 33–50 GB. This confirms that the zero-filtering strategy is key to scalability: by processing only expressed genes, the effective context length stays manageable even for 28k-gene datasets. Second, training time scales roughly linearly with dataset size, and the model converges on the largest dataset (1.27M cells) in under 6 hours on 8 GPUs. We will include these benchmarks in the revised manuscript along with the FLOPs analysis already reported in Tables 12–13.

---

> > ### Author Rebuttal · Reviewer_jW4g · 2026-03-31
> >
> > Thank you for your response and the additional experiments. Most of my concerns are addressed and I will maintain my positive score.

---

> > > ### Author Response · Authors · 2026-04-02
> > >
> > > We thank the reviewer for the thoughtful engagement throughout the review process and are glad that our responses and additional experiments have addressed the concerns.

---

### Official Review · Reviewer_G8oS · 2026-03-12

**Soundness:** 2
**Presentation:** 2
**Significance:** 3
**Originality:** 2
**Overall Recommendation:** 2
**Confidence:** 3

**Summary:**

This paper introduces scLDM, a generative model for single-cell gene expression data. The model combines a variational autoencoder with a latent diffusion model and uses a Transformer architecture to handle the permutation invariance and permutation equivariance inherent in single-cell data. Experiments are conducted across multiple datasets.

**Compliance With Llm Reviewing Policy:**

Affirmed.

**Final Justification:**

The rebuttal partially addressed my concerns.

**Key Questions For Authors:**

See weaknesses mentioned before.

**Limitations:**

The paper does not include a limitations section. The authors should add one. For example, the experimental results do not consistently demonstrate the effectiveness of the proposed technical contributions.

**Strengths And Weaknesses:**

Strength
Soundness: The design of MCAB is principled and handles permutation invariance and equivariance in a clever way. The model's insensitivity to gene ordering is well-motivated and the modeling choice here is reasonable.

Weaknesses
Soundness: The input processing step excludes zero counts from the input. However, zero counts have biological meaning and are in fact dominant in the data. The Encoder (Variational Posterior) takes gene expression counts as tokens as input to the downstream Transformer. Personally, I would not consider count data as discrete data, so I do not really appreciate this design choice. That said, there are many approaches to discretizing continuous data into tokens so that LLM-style Transformer techniques can be applied, so this alone is not necessarily a dealbreaker.
As for the experimental results, it looks like replacing the latent Gaussian with a negative binomial brings most of the benefit. In Table 2, scLDM (Gauss) has similar error to scDiffusion, while scLDM (NB) is much better. In Table 3, however, scLDM (Gauss) is already much stronger. The performance is thus inconsistent across tables, making it difficult to draw clear conclusions.
Presentation: There are a number of typos throughout the paper, including "exxpression" in Section 3.3, "uncertianites" in Section 4.3, and "ommitted" in Tables 4 and 5.

---

> ### Author Rebuttal · Authors · 2026-03-30
>
> We thank the reviewer for the helpful comments.
>
> **1. Zero counts excluded from the input**
>
> We thank the reviewer for raising this point. We want to clarify that zero-expression genes are filtered only at the encoder input, not at the decoder output. The decoder parameterizes the conditional likelihood (Negative Binomial or Gaussian) for all queried genes, including those with zero expression. The Negative Binomial distribution naturally places substantial probability mass at zero, so the model can and does predict zero expression for genuinely not expressed genes. Input filtering is therefore a computational strategy that reduces the effective context length from ~20k to ~2–5k tokens, rather than a loss of biological information.
>
> Empirically, we confirm this in two ways. First, Table 14 (Appendix K.1) shows that our zero-padding strategy outperforms the full-context baseline on reconstruction metrics while being more computationally efficient. Second, the $R^2$ Zeros scores in Tables 16 and 18 (e.g., 0.97 on Dentate Gyrus, 0.96 on HLCA, 0.94 on Parse 1M) demonstrate that scLDM accurately recovers gene-level sparsity patterns, including cell-type-specific zero expression. We will strengthen this discussion in the revised manuscript.
>
> **2. Count data as discrete data**
>
> We respectfully disagree with the characterization that count data should not be treated as discrete. Gene expression counts from scRNA-seq are inherently discrete random variables with well-characterized noise properties (Poisson sampling, overdispersion). Treating them as continuous values discards this structure and can degrade performance. This is precisely the motivation for using a Negative Binomial conditional likelihood, which is the standard in the single-cell field (scVI, scvi-tools). We also evaluate a Gaussian likelihood applied to log1p-normalized counts, and our experiments show that this variant is consistently less effective at capturing biological variation, reinforcing the importance of discrete count modeling.
>
> **3. Negative Binomial appears to provide most of the benefit**
>
> We agree that the Negative Binomial likelihood provides a significant performance boost, and we view this as a feature of our framework rather than a limitation. The modular design of scLDM is precisely what allows us to plug in the appropriate distribution for the data type. However, we note that the performance gains are not due solely to the likelihood. Even the Gaussian variant of scLDM outperforms or matches several baselines (scDiffusion, CPA, STATE-Tx, scGPT) on multiple metrics in Tables 2 and 3, demonstrating that the architectural contributions (MCAB, latent diffusion with DiT) provide independent value.
>
> **4. Inconsistent performance of scLDM (Gauss) across Tables 2 and 3**
>
> The difference in Gaussian performance across tables reflects the fundamentally different nature of the tasks, not an inconsistency in the model. Tables 1–2 evaluate the generation of cells from healthy tissue atlases (observational data), where the full distributional structure is measured. Table 3 evaluates conditional generation of unseen perturbation responses, where the signal of interest is the shift in expression relative to a control. While both are conditional generative settings, the class conditionals and distribution shifts being evaluated are different. We included both likelihood variants specifically to illustrate this task dependence, and we will add a brief discussion to clarify this point in the revised manuscript.
>
> **5. Typos**
>
> We thank the reviewer for the comment. We will fix the typos in our revised version of the paper.
>
> **6. Limitations section**
>
> We agree that a limitations section would strengthen the paper. We will add one to the revised manuscript discussing, among other points, the sensitivity to the choice of likelihood function across tasks, and the current restriction to transcriptomics data without multi-modal integration.

---

> > ### Author Rebuttal · Reviewer_G8oS · 2026-04-04
> >
> > Thank you for clarifying the difference in how the encoder and decoder handle zeros. While it is true that using a Negative Binomial distribution in the decoder can predict zero expression, discarding all zeros in the encoder loses a great deal of important information and also prevents the decoder from learning which inputs should be mapped to zero outputs. The balance between zero-filtering in the encoder and the Negative Binomial distribution in the decoder strikes me as rather shaky and sensitive, which is more a matter of engineering than of principled science.

---

> > > ### Author Response · Authors · 2026-04-06
> > >
> > > Dear Reviewer,
> > >
> > > Thank you for your follow-up. We appreciate your continued engagement and would like to address your remaining concern directly.
> > >
> > > **On the claim that discarding zeros in the encoder "loses a great deal of important information" and "prevents the decoder from learning which inputs should be mapped to zero outputs":**
> > > We want to clarify the architecture once more: the decoder does not receive the encoder's masked input directly. It receives a latent representation learned by the encoder and is conditioned on the full set of queried genes, including those with zero expression. The decoder, therefore, has every opportunity to learn which genes should be zero for a given cell. This is not a hypothetical argument, we provided direct empirical evidence:
> > > - Table 14 (Appendix K.1) shows that zero-filtering outperforms the full-context baseline on reconstruction metrics. If discarding zeros in the encoder truly lost critical information, we would expect the opposite result.
> > > - Tables 16 and 18 report ΔZeros scores of 0.97 (Dentate Gyrus), 0.96 (HLCA), and 0.94 (Parse 1M), demonstrating that the model accurately recovers gene-level sparsity patterns across datasets and scales.
> > >
> > > We would be glad to provide additional evidence if the reviewer has a specific failure mode in mind. Based on the original reviewers' comments, we believe that the concerns have been substantively addressed.
> > >
> > > **On the characterization as "engineering rather than principled science":**
> > > We respectfully disagree with this framing. The encoder asymmetry is motivated by a well-understood property of single-cell data: the vast majority of zeros are uninformative for encoding cell state (they reflect either biological absence or technical dropout, neither of which requires explicit representation in the input). Reducing context length from ~20k to ~2–5k tokens is not merely an engineering convenience: it directly improves both computational efficiency and empirical performance, as Table 14 demonstrates. Architectural choices that are empirically validated and biologically motivated are, in our view, exactly the kind of principled design decisions that advance the field.
> > >
> > > We have also taken seriously all feedback from the review process: we are adding a limitations section, clarifying the task-dependent behavior of the Gaussian vs. NB likelihood, and fixing the noted typos. We hope this response, together with the positive assessments from Reviewers a2XN and jW4g, provides a sufficient basis for reconsidering the overall evaluation.
> > > Thank you again for your time and feedback.

---

### Official Review · Reviewer_a2XN · 2026-03-12

**Soundness:** 4
**Presentation:** 4
**Significance:** 3
**Originality:** 4
**Overall Recommendation:** 5
**Confidence:** 4

**Summary:**

This paper presents scLDM, a two-stage generative model for single-cell gene expression data. In the first stage, the authors design a Transformer-based VAE that uses a cross-attention mechanism (MCAB) to produce permutation-invariant latent variables. In the second stage, a latent diffusion model with Diffusion Transformers and flow matching replaces the standard Gaussian prior, enabling multi-conditional generation. The paper is well-written and the experiments are broad, covering both observational and perturbational datasets, as well as a downstream classification task. The results show clear improvements over several baselines. However, I have concerns about the core motivation, the biological validity of some design choices, and whether the performance gains truly come from the claimed innovations or simply from the use of larger and more expensive models.

**Compliance With Llm Reviewing Policy:**

Affirmed.

**Final Justification:**

I maintain my recommendation to accept this paper because the authors have answered my main questions clearly and honestly. The new experiment shows the architecture design is important, not just the model size, which supports their claim well. I also appreciate their promise to revise the text about classification results and explain the module novelty more precisely. Overall, this work is solid and will be useful for the community.

**Key Questions For Authors:**

**Key Questions for Authors**

1. **On the motivation for permutation invariance:** Can you provide a direct comparison between your Transformer-based VAE and a well-tuned MLP-based VAE (like scVI) with the same latent dimension and similar parameter count, on a dataset where the full gene set is used (not 2,000 HVGs)? The current reconstruction comparisons in Tables 1 and 17 use different model sizes and different gene subsets, which makes it impossible to conclude that the permutation-invariant design is the reason for the improvement.

2. **On the HVG restriction:** The introduction explicitly states that the model avoids "operating on a restricted subset of highly variable genes." However, Sections 4.2 uses 2,000 HVGs. Could the authors explain this discrepancy? If the model is tested only on HVGs in the perturbational setting, does it still have an advantage over MLP-based models that also run on 2,000 HVGs? Please provide a direct comparison under the same gene selection protocol.

3. **On the zero-filtering step:** You remove all zero-expression genes from the input and replace them with PAD tokens. Can you show, with a controlled ablation, what happens to generation quality if structural zeros are kept? For example, on a dataset where cell-type-specific gene silencing is well annotated, does removing zeros cause the model to lose the ability to distinguish cell types that differ mainly in which genes are silenced rather than which genes are highly expressed?

**Strengths And Weaknesses:**

**Strengths**

- **Comprehensive evaluation.** The authors test their model across many datasets and tasks: unconditional generation, conditional generation on perturbational data, reconstruction quality, and cell embedding evaluation. This is more complete than many papers in this area, and the comparison with a good set of baselines (scVI, CFGen, CellFlow, STATE-Tx, scGPT) makes the results easier to interpret.

- **Strong generation quality.** The Wasserstein-2, MMD, and Frechet Distance scores in Tables 2 and 3 are consistently better than the baselines. The precision-recall trade-off is more balanced, which is a meaningful advantage over methods like CPA and STATE-Tx that show near-zero recall. The R² variance scores in Table 18 are particularly strong compared to baselines, showing that the model captures second-order statistics better.

- **Useful ablation studies.** The comparison of aggregation strategies (max, mean, sum versus cross-attention pooling) in Figure 5, and the comparison of joint versus additive classifier-free guidance in Table 19, are helpful. They give the reader confidence that the specific design choices matter.

---

**Weaknesses**

- **The core motivation is overstated and partially misleading.** The paper argues that existing methods fail because they impose an "artificial gene ordering." However, standard MLP-based models like scVI do not actually encode sequential or spatial dependencies between genes. They simply map a fixed-dimension input vector to a latent space, where each dimension corresponds to one gene. Permuting the input would require permuting the weights accordingly, but no ordering relationship is assumed. The permutation-invariance problem is therefore much less severe than the authors claim. The authors should give a more careful and precise argument for why a permutation-invariant architecture is necessary over a well-trained MLP, ideally with a controlled experiment that isolates this factor.

- **The paper contradicts its own motivation in the perturbational experiments.** A central claim in the introduction is that the model avoids restriction to a subset of highly variable genes (HVGs). Yet in Section 4.2, both perturbational experiments (Parse 1M and Replogle) are performed on 2,000 HVGs only, following the protocol of Adduri et al. (2025). This directly contradicts the paper's own stated motivation. The authors should either remove this claim from the introduction, or provide experiments that demonstrate the model's advantage when using the full gene set compared to HVG-restricted baselines.

- **The biological justification for zero-expression filtering is not convincing.** In Section 3.2, the authors remove all zero-count genes from the input and replace them with PAD tokens. They justify this by saying that zeros in scRNA-seq data are mostly technical dropouts. This is only partially true. Structural zeros, meaning genes that are genuinely not expressed in a particular cell type or state, carry real biological information. By discarding all zeros, the model loses information about gene silencing patterns, which are relevant for cell state identification and perturbation modeling. The authors should either quantify how much information is lost by this filtering step, or discuss its limitations more honestly in the text.

- **The performance gains in the classification experiment (Tables 4 and 5) are not convincing.** The paper assesses a notable context of cell-level classification as evidence for the quality of learned representations. However, in Table 5 (Tabula Sapiens 2.0), the differences between all top-performing models are within the reported standard error of ±0.002. A 270M-parameter model achieving F1 = 0.802 versus scGPT's F1 = 0.800 is not a meaningful improvement. More importantly, the 20M-parameter scLDM model achieves the highest F1 score on this dataset, while the 270M model does not. This suggests that the improvement does not come from the architectural design but simply from scale, and even scale does not give consistent gains. The authors should discuss this inconsistency directly rather than framing it as a success.

- **The architectural novelty is not clearly established.** The MCAB module, which the authors present as a key contribution, is structurally very similar to the Perceiver IO architecture (Jaegle et al., 2022). The difference between MCAB and the Pooling by Multi-head Attention (PMA) block from SetTransformer (Lee et al., 2019) is also small. The authors mention these related works but do not clearly articulate what is fundamentally different in their design. This makes it difficult to understand what the true architectural contribution is beyond combining existing components.

---

> ### Author Rebuttal · Authors · 2026-03-30
>
> We thank the reviewer for the helpful comments.
>
> **1. On the motivation for permutation invariance**
>
> We thank the reviewer for this important suggestion. We have completed the controlled experiment requested by the reviewer. We compared our Transformer-based VAE against an MLP-based VAE (scVI architecture) on the Dentate Gyrus dataset using the full gene set (17k genes), with matched latent dimensionality (2,048), using the same test split as reported in the paper, across 3 seeds. The results are the following: our TransformerVAE achieves substantially better reconstruction across all metrics:
> - 20% lower reconstruction error,
> - 4.5× higher Pearson correlation,
> - 23% lower MSE
>
> while using **26× fewer parameters** (3.4M vs 90.7M). The MLP-based model consisted of a single hidden layer of 2048 units for both encoder and decoder, together with a latent space of the same size. This demonstrates that the performance gains of scLDM are not simply due to model scale, but stem from the architectural design.
>
> | Model | Params | RE ↓ | PCC ↑ | MSE ↓ |
> |:---|:---:|:---:|:---:|:---:|
> | ScviVAE (MLP) | 90.7M | 5550.3 ± 11.1 | 0.059 ± 0.000 | 0.367 ± 0.000 |
> | TransformerVAE (Ours) | 3.4M | 4446.3 ± 29.0 | 0.264 ± 0.004 | 0.282 ± 0.001 |
>
> We note, however, that the motivation for permutation invariance extends beyond empirical performance: our claim is principled rather than purely empirical. While it is true that MLPs do not encode sequential dependencies, they do assign a fixed positional meaning to each input dimension: weight column $j$ is permanently tied to gene $j$. This means the model must be retrained, or its weights surgically permuted, whenever the gene vocabulary changes (e.g., across tissues or species). Our architecture avoids this by design: genes are identified by their embedding index rather than their position in the input vector, enabling a single model to handle arbitrary gene subsets without retraining. We will clarify this distinction in the revised manuscript.
>
> **2. On the HVG restriction**
>
> We thank the reviewer for highlighting this discrepancy. The restriction to 2,000 HVGs in Section 4.2 is driven entirely by the evaluation protocol: we follow the benchmark setup of Adduri et al. (2025) to ensure a fair head-to-head comparison, and several baselines (CPA, STATE-Tx, scGPT, CellFlow) either require or were originally evaluated on this fixed HVG set. Importantly, our model does not require HVG selection at the architectural level. In Section 4.1, scLDM operates on the full gene set (17k–28k genes) for all three observational datasets, and the reconstruction and generation results reported in Tables 1, 2, and 15 confirm that it performs well at this scale.
>
> **3. On the zero-filtering step**
>
> We appreciate the reviewer's nuanced point about structural zeros versus technical dropouts. We want to clarify an important detail about our design: zero-expression genes are filtered only at the *encoder input*, not at the decoder output. The decoder receives gene IDs via the embedding matrix and produces conditional-likelihood parameters (Negative Binomial or Gaussian) for **all** queried genes, including those that were zero in the original cell. This means the model can and does predict zero or near-zero expression for genes that are genuinely not expressed, because the Negative Binomial distribution naturally places probability mass at zero. The input filtering is therefore a computational efficiency choice (reducing the effective context length from ~20k tokens to ~2–5k), not a loss of biological information at generation time.
>
> Empirically, we show in Table 14 (Appendix K.1) that the zero-padding strategy outperforms the full-context input on reconstruction metrics while also being more computationally efficient. We believe this effectively acts as a form of regularization for the model. Additionally, our $R^2$ Zeros scores in Table 16 (e.g., 0.97 on Dentate Gyrus, 0.96 on HLCA) demonstrate that scLDM accurately recovers the sparsity structure of the true data. This also holds for the perturbational datasets (Table 18 in the Appendix).

---

> > ### Author Rebuttal · Reviewer_a2XN · 2026-04-01
> >
> > Thank you for your detailed rebuttal. I appreciate the new experiment comparing your model with the MLP baseline. Your explanation about the zero-filtering at the encoder is also very clear now. These responses successfully address my main questions.
> >
> > However, I noticed that two points from my original review were completely skipped in the rebuttal. I think we need to resolve them to make the paper scientifically solid.
> >
> > **1. Model Scale and Classification Performance (Table 5)**
> > In the Tabula Sapiens 2.0 experiment, the F1 scores for all top models are very close, falling within the error bar ($\pm0.002$). More importantly, your small 20M-parameter model actually gets a slightly higher score than your large 270M-parameter model.
> >
> > This result does not support the claim that scaling up the architecture brings consistent benefits. It is completely fine in science if a bigger model does not help on a specific downstream task. However, the paper should discuss this observation honestly and directly, rather than framing it as a success for the large model. Could you please explain how you will revise the text to reflect this?
> >
> > **2. Architectural Novelty of the MCAB Module**
> > In my initial review, I pointed out that your MCAB module looks structurally very similar to existing works, specifically Perceiver IO and the PMA block in SetTransformer. The rebuttal did not address this concern.
> >
> > To help me evaluate the technical contribution, I still need to understand: what is the fundamental, structural difference between your MCAB design and these previous modules? Please explicitly clarify the novel changes you made.

---

> > > ### Author Response · Authors · 2026-04-02
> > >
> > > Thank you for raising important points missing in our rebuttal. We only addressed the questions and forgot to properly address the weaknesses you raised. Here's our attempt at clarifying them:
> > >
> > > **1. Model Scale and Classification Performance (Table 5)**
> > > We thank the reviewer for pressing on this point. We fully agree: on the Tabula Sapiens 2.0 benchmark, all top-performing models (scLDM variants, scGPT, scVI, TranscriptFormer) achieve F1 scores within each other's standard errors (0.799–0.804 ± 0.002), and the 20M model slightly outperforms the 270M model. This indicates that additional model capacity does not yield meaningful gains on this particular task, likely because the benchmark is approaching saturation for cell-type classification with logistic regression probes. We will revise the manuscript to state this directly and avoid framing the Tabula Sapiens result as evidence for scaling benefits. We plan to revise it with the following:
> > >
> > > …achieve F1 scores within each other’s uncertainties (ranging from 0.799 to 0.804 with standard errors of 0.002), indicating comparable performance for multi-class cell type classification, *but also that this benchmark is near saturation for cell-type classification using linear probes*…
> > >
> > > Would this clarification in the main text suffice to explain the limits of this benchmark?
> > >
> > > Importantly, the COVID-19 classification task (Table 4) does show a clear monotonic improvement with scale (20M → 70M → 270M), suggesting that the value of additional capacity is task-dependent.
> > >
> > > **2. Architectural Novelty of the MCAB Module**
> > >
> > > We thank the reviewer for the follow-up. We agree that the individual components (cross-attention with learnable queries) are established. The novelty lies in their specific combination and dual use within a single VAE for exchangeable data such as gene expression:
> > > - **Encoder (pooling):** The MCAB uses fixed learnable pseudoinputs as queries and gene tokens as keys/values, producing permutation-invariant latent tokens. This is indeed analogous to Perceiver IO architecture.
> > > - **Decoder (unpooling):** The same MCAB architecture is reused, but with a critical difference: the queries are now the *gene-specific embeddings* $E_I$ from the shared embedding matrix, indexed by the requested gene IDs. This makes the output permutation-equivariant with respect to gene ordering (Property 4 in Appendix C). This unpooling mechanism — using the original gene embeddings as queries into the latent tokens — is, to our knowledge, not present in Perceiver IO or SetTransformer, which do not address the equivariant decoding problem. Nevertheless, it is very possible to be present in other formulations of transformers for set-based data. For example, in LaM-SLidE (Sestak et al. 2025), which we cite, a similar formulation is used but in a different context, and where query embeddings have a different meaning.
> > >
> > > The result is a single module that, depending on whether the queries are fixed (encoder) or index-dependent (decoder), switches between invariance and equivariance. This eliminates the need for separate pooling and unpooling architectures (as in SetVAE's dual-latent hierarchical design). We will make this distinction more explicit in the revised manuscript.
> > >
> > > We hope this clarifies the remaining questions.

---

### Decision · Program_Chairs · 2026-04-30

**Decision:**

Accept (regular)

**Comment:**

This paper considers generative modeling for single-cell gene expression data via latent diffusion, where the autoencoder architecture handles permutation invariance and equivariance of IDs effectively. The reviewers agree that the equivariance is important, the architectural improvement is well motivated and that the empirical evaluations demonstrate the advantages of the proposed method.

One of the main concerns raised was regarding zero filtering in the encoder, which the reviewers argued can remove information from the input (structural zeros vs technical dropouts). The authors argue that the structural zeros can be handled by the decoder due to their algorithm design and showed that the zero filtering improved performance and computational efficiency. Two out of the three reviewers agreed with this rebuttal. However, I feel a simple theoretical analysis is needed to show that this indeed is a principled design choice, and that the improved performance is not just spurious.

Thus, I recommend a weak accept.